# A Theoretical Justification for Asymmetric Actor-Critic Algorithms

**Gaspard Lambrechts** [1] [*] **Damien Ernst** [1] **Aditya Mahajan** [2]

## Abstract

In reinforcement learning for partially observable environments, many successful algorithms have been developed within the asymmetric learning paradigm. This paradigm leverages additional state information available at training time for faster learning. Although the proposed learning objectives are usually theoretically sound, these methods still lack a precise theoretical justification for their potential benefits. We propose such a justification for asymmetric actor-critic algorithms with linear function approximators by adapting a finite-time convergence analysis to this setting. The resulting finite-time bound reveals that the asymmetric critic eliminates error terms arising from aliasing in the agent state.

## 1. Introduction

Reinforcement learning (RL) is an appealing framework for solving decision making problems, notably because it makes very few assumptions about the problem at hand. In its purest form, the promise of an RL algorithm is to learn an optimal behavior from interaction with an environment whose dynamics are unknown. More formally, an RL algorithm aims to learn a policy – which is defined as a mapping from observations to actions – from interaction samples, in order to maximize a reward signal. While RL has obtained empirical successes for a plethora of challenging problems ranging from games to robotics (Mnih et al., 2015; Schrittwieser et al., 2020; Levine et al., 2015; Akkaya et al., 2019), most of these achievements have assumed full state observability. A more realistic assumption is partial state observability, where only a partial observation of the state of the environment is available for taking actions. In this setting, the optimal action generally depends on the complete history of past observations and actions. Tradi-

tional RL approaches have thus been adapted by considering history-dependent policies, usually with a recurrent neural network to process histories (Bakker, 2001; Wierstra et al., 2007; Hausknecht & Stone, 2015; Heess et al., 2015; Zhang et al., 2016; Zhu et al., 2017). Given the difficulty of learning effective history-dependent policies, various auxiliary representation learning objectives have been proposed to compress the history into useful representations (Igl et al., 2018; Buesing et al., 2018; Guo et al., 2018; Gregor et al., 2019; Han et al., 2019; Guo et al., 2020; Lee et al., 2020; Subramanian et al., 2022; Ni et al., 2024). Such methods usually seek to learn history representations that encode the belief, defined as the posterior distributions over the states given the history, which is a sufficient statistic of the history for optimal control.

While these methods are theoretically able to learn optimal history-dependent policies, they usually learn solely from the partial state observations, which can be restrictive. Indeed, assuming the same partial observability at training time and execution time can be too pessimistic for many environments, notably for those that are simulated. This motivated the asymmetric learning paradigm, where additional state information available at training time is leveraged during the process of learning a history-dependent policy. Although the optimal policies obtained by asymmetric learning are theoretically equivalent to those learned by symmetric learning, the promise of asymmetric learning is to improve the convergence speed. Early approaches proposed to imitate a privileged policy conditioned on the state (Choudhury et al., 2018), or to use an asymmetric critic conditioned on the state (Pinto et al., 2018). These heuristic methods initially lacked a theoretical framework, and a recent line of work has focused on proposing theoretically grounded asymmetric learning objectives. First, imitation learning of a privileged policy was known to be suboptimal, and it was addressed by constraining the privileged policy so that its imitation results in an optimal policy for the partially observable environment (Warrington et al., 2021). Similarly, asymmetric actor-critic approaches were proven to provide biased gradients, and an unbiased actor-critic approach was proposed by introducing the history-state value function (Baisero & Amato, 2022). In model-based RL, several works proposed world model objectives that are proved to provide sufficient statistics of the history, by leveraging

[*]Work done at McGill University and Mila Québec. [1]Montefiore Institute, University of Liège [2]Department of Electrical and Computer Engineering, McGill University. Correspondence to: Gaspard Lambrechts <gaspard.lambrechts@uliege.be>.

*Proceedings of the 42nd International Conference on Machine Learning*, Vancouver, Canada. PMLR 267, 2025. Copyright 2025 by the author(s).

the state (Avalos et al., 2024) or arbitrary state information (Lambrechts et al., 2024). Finally, asymmetric representation learning approaches were proposed to learn sufficient statistics from state samples (Wang et al., 2023; Sinha & Mahajan, 2023). It is worth noting that many recent successful applications of RL have greatly benefited from asymmetric learning, usually through an asymmetric critic (Degrave et al., 2022; Kaufmann et al., 2023; Vasco et al., 2024).

Despite these methods being theoretically grounded, in the sense that policies satisfying these objectives are optimal policies, they still lack a theoretical justification for their potential benefit. In particular, there is no theoretical justification for the improved convergence speed of asymmetric learning. In this work, we propose such a justification for an asymmetric actor-critic algorithm, using agent-state policies and linear function approximators. Agent-state policies rely on an internal state, which is updated recurrently based on successive actions and observations, from which the next action is selected. This agent state can introduce aliasing, a phenomenon in which an agent state may correspond to two different beliefs. Our argument relies on the comparaison of two analogous finite-time bounds: one for a symmetric natural actor-critic algorithm (Cayci et al., 2024), and its adaptation to the asymmetric setting that we derive in this paper. This comparison reveals that asymmetric learning eliminates error terms arising from aliasing in the agent state in symmetric learning. These aliasing terms are given by the difference between the true belief (i.e., the posterior distribution over the states given the history) and the approximate belief (i.e., the posterior distribution over the states given the agent state). This suggests that asymmetric learning may be particularly useful when aliasing is high.

A recent related work proposed a model-based asymmetric actor-critic algorithm relying on belief approximation, and proved its sample efficiency (Cai et al., 2024). It also considered agent-state policies, and studied the finite-time performance by providing a probably approximately correct (PAC) bound, instead of an expectation bound as here. While the algorithm was restricted to finite horizon and discrete spaces, notably for implementing count-based exploration strategies, it tackled the online exploration setting and its performance bound did not present a concentrability coefficient. This related analysis thus provides a promising framework for future works in a more challenging setting. However, it did not study the existing asymmetric actor-critic algorithm, and did not provides a direct comparison with symmetric learning. In contrast, we focus on providing comparable bounds for the existing model-free asymmetric actor-critic algorithm and its symmetric counterpart.

In Section 2, we formalize the environments, policies, and Q-functions that are considered. In Section 3, we introduce the asymmetric and symmetric actor-critic algorithms that

are studied. In Section 4, we provide the finite-time bounds for the asymmetric and symmetric actor-critic algorithms. Finally, in Section 5, we conclude by summarizing the contributions and providing avenues for future works.

## 2. Background

In Subsection 2.1, we introduce the decision processes and agent-state policies that are considered. Then, we introduce the asymmetric and symmetric Q-function for such policies, in Subsection 2.2 and Subsection 2.3, respectively.

### 2.1. Partially Observable Markov Decision Process

A partially observable Markov decision process (POMDP) is a tuple $\mathcal{P} = (\mathcal{S}, \mathcal{A}, \mathcal{O}, P, T, R, O, \gamma)$, with discrete state space $\mathcal{S}$, discrete action space $\mathcal{A}$, and discrete observation space $\mathcal{O}$. The initial state distribution $P$ gives the probability $P(s_0)$ of $s_0 \in \mathcal{S}$ being the initial state of the decision process. The dynamics are described by the transition distribution $T$ that gives the probability $T(s_{t+1}|s_t, a_t)$ of $s_{t+1} \in \mathcal{S}$ being the state resulting from action $a_t \in \mathcal{A}$ in state $s_t \in \mathcal{S}$. The reward function $R$ gives the immediate reward $r_t = R(s_t, a_t, s_{t+1})$ of the reward $r_t \in [0, 1]$ resulting from this transition. The observation distribution $O$ gives the probability $O(o_t|s_t)$ to get observation $o_t \in \mathcal{O}$ in state $s_t \in \mathcal{S}$. Finally, the discount factor $\gamma \in [0, 1)$ weights the relative importance of future rewards. Taking a sequence of $t$ actions in the POMDP conditions its execution and provides the history $h_t = (o_0, a_0, \ldots, o_t) \in \mathcal{H}$, where $\mathcal{H}$ is the set of histories of arbitrary length. In general, the optimal policy in a POMDP depends on the complete history.

However, in practice it is infeasible to learn a policy conditioned on the full history, since the latter grows unboundedly with time. We consider an agent-state policy $\pi \in \Pi_{\mathcal{M}}$ that uses an agent-state process $\mathcal{M} = (\mathcal{Z}, U)$, in order to take actions (Dong et al., 2022; Sinha & Mahajan, 2024). More formally, we consider a discrete agent state space $\mathcal{Z}$, and an update distribution $U$ that gives the probability $U(z_{t+1}|z_t, a_t, o_{t+1})$ of $z_{t+1} \in \mathcal{Z}$ being the state resulting from action $a_t \in \mathcal{A}$ and observation $o_{t+1} \in \mathcal{O}$ in agent state $z_t \in \mathcal{Z}$. Note that the update distribution $U$ also describe the initial agent state distribution with $z_{-1} \notin \mathcal{Z}$ the null agent state and $a_{-1} \notin \mathcal{A}$ the null action. Some examples of agent states that are often used are a sliding window of past observations, or a belief filter. Aliasing may occur when the agent state does not summarize all information from the history about the state of the environment, see Appendix A for an example. Given the agent state $z_t$, the policy $\pi$ samples actions according to $a_t \sim \pi(\cdot|z_t)$. An agent-state policy $\pi^* \in \Pi_{\mathcal{M}}$ is said to be optimal for an agent-state process $\mathcal{M}$ if it maximizes the expected discounted sum of rewards: $\pi^* \in \arg\max_{\pi \in \Pi_{\mathcal{M}}} J(\pi)$ with $J(\pi) = \mathbb{E}^{\pi}[\sum_{t=0}^{\infty} \gamma^t R_t]$.

In the following, we denote by $S_t$, $O_t$, $Z_t$, $A_t$ and $R_t$ the random variables induced by the POMDP $\mathcal{P}$. Given a POMDP $\mathcal{P}$ and an agent-state process $\mathcal{M}$, the initial environment-agent state distribution $P$ is given by,

$$P(s_0, z_0) = P(s_0) \sum_{o_0 \in \mathcal{O}} O(o_0|s_0) U(z_0|z_{-1}, a_{-1}, o_0). \quad (1)$$

Furthermore, given an agent-state policy $\pi \in \Pi_{\mathcal{M}}$, we define the discounted visitation distribution as,

$$d^\pi(s, z) = (1 - \gamma) \sum_{s_0, z_0} P(s_0, z_0) \quad (2)$$
$$\times \sum_{t=0}^{\infty} \gamma^k \Pr(S_t = s, Z_t = z|S_0 = s_0, Z_0 = z_0).$$

Finally, we define the visitation distribution $m$ steps from the discounted visitation distribution as,

$$d_m^\pi(s, z) = \sum_{s_0, z_0} d^\pi(s_0, z_0) \quad (3)$$
$$\times \Pr(S_m = s, Z_m = z|S_0 = s_0, Z_0 = z_0).$$

In the following, we define the various value functions for the policies that we defined. Note that we use calligraphic letters $\mathcal{Q}^\pi$, $\mathcal{V}^\pi$ and $\mathcal{A}^\pi$ for the asymmetric functions, and regular letters $Q^\pi$, $V^\pi$ and $A^\pi$ for the symmetric ones.

## 2.2. Asymmetric Q-function

Similarly to the asymmetric Q-function of Baisero & Amato (2022), which is conditioned on $(s, h, a)$, we define an asymmetric Q-function that we condition on $(s, z, a)$, where $z$ is the agent state resulting from history $h$. The asymmetric Q-function $\mathcal{Q}^\pi$ of an agent-state policy $\pi \in \Pi_{\mathcal{M}}$ is defined as the expected discounted sum of rewards, starting from environment state $s$, agent state $z$, and action $a$, and using policy $\pi$ afterwards,

$$\mathcal{Q}^\pi(s, z, a) = \mathbb{E}^\pi \left[ \sum_{t=0}^{\infty} \gamma^t R_t \middle| S_0 = s, Z_0 = z, A_0 = a \right]. \quad (4)$$

The asymmetric value function $\mathcal{V}^\pi$ of an agent-state policy $\pi \in \Pi_{\mathcal{M}}$ is defined as $\mathcal{V}^\pi(s, z) = \sum_{a \in \mathcal{A}} \pi(a|z) \mathcal{Q}^\pi(s, z, a)$. We also define the asymmetric advantage function $\mathcal{A}^\pi(s, z, a) = \mathcal{Q}^\pi(s, z, a) - \mathcal{V}^\pi(s, z)$.

Let us define the $m$-step asymmetric Bellman operator as,

$$\widetilde{\mathcal{Q}}^\pi(s, z, a) = \mathbb{E}^\pi \left[ \sum_{t=0}^{m-1} \gamma^t R_t + \gamma^m \widetilde{\mathcal{Q}}^\pi(S_m, Z_m, A_m) \middle| S_0 = s, Z_0 = z, A_0 = a \right]. \quad (5)$$

Since this $m$-step asymmetric Bellman operator is $\gamma^m$-contractive, equation (5) has a unique fixed point $\widetilde{\mathcal{Q}}^\pi$. Notice that, when using an agent-state policy, the environment

state and agent state $(S_t, Z_t)$ are Markovian. Therefore, it can be shown that the fixed point $\widetilde{\mathcal{Q}}^\pi$ is the same as the asymmetric Q-function $\mathcal{Q}^\pi$.

## 2.3. Symmetric Q-function

The symmetric Q-function $Q^\pi$ of an agent-state policy $\pi \in \Pi_{\mathcal{M}}$ in a POMDP $\mathcal{P}$ is defined as the expected discounted sum of rewards, starting from agent state $z$ and action $a$, and using policy $\pi$ afterwards,

$$Q^\pi(z, a) = \mathbb{E}^\pi \left[ \sum_{t=0}^{\infty} \gamma^t R_t \middle| Z_0 = z, A_0 = a \right]. \quad (6)$$

The symmetric value function $V^\pi$ of an agent-state policy $\pi \in \Pi_{\mathcal{M}}$ is defined as $V^\pi(z) = \sum_{a \in \mathcal{A}} \pi(a|z) Q^\pi(z, a)$. We also define the symmetric advantage function $A^\pi(z, a) = Q^\pi(z, a) - V^\pi(z)$.

Let us define the $m$-step symmetric Bellman operator as,

$$\widetilde{Q}^\pi(z, a) = \mathbb{E}^\pi \left[ \sum_{t=0}^{m-1} \gamma^t R_t + \gamma^m \widetilde{Q}^\pi(Z_m, A_m) \middle| Z_0 = z, A_0 = a \right]. \quad (7)$$

It can be verified that the $m$-step symmetric Bellman operator is $\gamma^m$-contractive. Therefore, equation (7) has a unique fixed point $\widetilde{Q}^\pi$. However, because the agent state is not necessarily Markovian, in general $Q^\pi \neq \widetilde{Q}^\pi$.

# 3. Natural Actor-Critic Algorithms

In this section, we present the asymmetric and symmetric natural actor-critic algorithms, which make use of an actor, or policy, and a critic, or Q-function. The asymmetric variant will use an asymmetric critic, learned using asymmetric temporal difference learning, while the symmetric variant will use a symmetric critic, learned using symmetric temporal difference learning. These temporal difference learning algorithms are presented in in Subsection 3.1 and Subsection 3.2, respectively. Then, Subsection 3.3 presents the complete natural actor-critic algorithm that uses a temporal difference learning algorithm as a subroutine.

For any Euclidean space $\mathcal{X}$, let $\mathcal{B}_2(0, B)$ be the $\ell_2$-ball centered at the origin with radius $B > 0$, and let $\Gamma_{\mathcal{C}} : \mathcal{X} \to \mathcal{C}$ be a projection operator into the closed and convex set $\mathcal{C} \subseteq \mathcal{X}$ in $\ell_2$-norm: $\Gamma_{\mathcal{C}}(x) \in \arg\min_{c \in \mathcal{C}} \|c - x\|_2^2 \subseteq \mathcal{C}, \forall x \in \mathcal{X}$. Finally, let us define the $\mu$-weighted $\ell_2$-norm, for any probability measures $\mu \in \Delta(\mathcal{X})$ as,

$$\|f\|_\mu = \sqrt{\sum_{x \in \mathcal{X}} \mu(x) |f(x)|^2}. \quad (8)$$

In the algorithms, we implicitly assume to be able to directly sample from the discounted visitation measure $d^\pi$. When it is unrealistic, it is still possible to sample from $d^\pi$ by sampling an initial timestep $t_0 \sim \text{Geom}(1-\gamma)$ from a geometric distribution with success rate $1-\gamma$, and then taking $t_0 - 1$ actions in the POMDP. The resulting sample $(s_{t_0}, z_{t_0})$ follows the distribution $d^\pi$.

### 3.1. Asymmetric Critic

Suppose we are given features $\phi\colon \mathcal{S} \times \mathcal{Z} \times \mathcal{A} \to \mathbb{R}^{d_\phi}$. Without loss of generality, we assume $\sup_{s,z,a}\|\phi(s,z,a)\|_2 \leq 1$. Given a weight vector $\beta \in \mathbb{R}^{d_\phi}$, let $\widehat{\mathcal{Q}}^\pi_\beta$ denote the linear approximation of the asymmetric Q-function $\mathcal{Q}^\pi$ that uses features $\phi$ with weight $\beta$,

$$\widehat{\mathcal{Q}}^\pi_\beta(s,z,a) = \langle \beta, \phi(s,z,a) \rangle. \tag{9}$$

Given an arbitrary projection radius $B > 0$, we define the hypothesis space as,

$$\mathcal{F}^B_\phi = \{(s,z,a) \mapsto \langle \beta, \phi(s,z,a) \rangle : \beta \in \mathcal{B}_2(0,B)\}. \tag{10}$$

We denote the optimal parameter of the asymmetric critic approximation by $\beta^\pi_* \in \arg\min_{\beta \in \mathcal{B}_2(0,B)} \|\langle \beta, \phi(\cdot) \rangle - \mathcal{Q}^\pi(\cdot)\|_d$, and denote the corresponding approximation by $\widehat{\mathcal{Q}}^\pi_*(\cdot) = \langle \beta^\pi_*, \phi(\cdot) \rangle$. The corresponding error is,

$$\varepsilon_{\text{app}} = \min_{f \in \mathcal{F}^B_\phi} \left\| f - \mathcal{Q}^\pi \right\|_d = \left\| \widehat{\mathcal{Q}}^\pi_* - \mathcal{Q}^\pi \right\|_d, \tag{11}$$

with $d(s,z,a) = d^\pi(s,z)\pi(a|z)$ the sampling distribution.

In Algorithm 1, we present the $m$-step temporal difference learning algorithm for approximating the asymmetric Q-function $\mathcal{Q}^\pi$ of an arbitrary agent-state policy $\pi \in \Pi_\mathcal{M}$. At each step $k$, the algorithm obtains one sample $(s_{k,0}, z_{k,0}) \sim d^\pi$ from the discounted visitation distribution. Then, $m$ actions are selected according to policy $\pi$ to provide samples $(a_{k,t}, r_{k,t}, s_{k,t+1}, o_{k,t+1}, z_{k,t+1})$ for $0 \leq t < m$. Next, the temporal difference $\delta_k$ and semi-gradient $g_k$ are computed, based on a last action $a_{k,m} \sim \pi(\cdot|z_{k,m})$,

$$\delta_k = \sum_{i=0}^{m-1} \gamma^i r_{k,i} + \gamma^m \widehat{\mathcal{Q}}^\pi_{\beta_k}(s_{k,m}, z_{k,m}, a_{k,m})$$
$$- \widehat{\mathcal{Q}}^\pi_{\beta_k}(s_{k,0}, z_{k,0}, a_{k,0}), \tag{12}$$
$$g_k = \delta_k \nabla_\beta \widehat{\mathcal{Q}}^\pi_{\beta_k}(s_{k,0}, z_{k,0}, a_{k,0}). \tag{13}$$

Then, the semi-gradient update is performed with $\beta^-_{k+1} = \beta_k + \alpha g_k$ and the parameters are projected onto the ball of radius $B$: $\beta_{k+1} = \Gamma_{\mathcal{B}_2(0,B)}(\beta^-_{k+1})$. At the end, the algorithm computes the average parameter $\bar{\beta} = \frac{1}{K}\sum_{k=0}^{K-1}\beta_k$ and returns the average approximation $\overline{\mathcal{Q}}^\pi = \widehat{\mathcal{Q}}^\pi_{\bar\beta}$.

---

**Algorithm 1** $m$-step temporal difference learning algorithm

**input:** policy $\pi \in \Pi_\mathcal{M}$, bootstrap timestep $m$, step size $\alpha$, number of updates $K$, projection radius $B$.
**for** $k = 0 \ldots K-1$ **do**
  Initialize $(s_{k,0}, z_{k,0}) \sim d^\pi$.
  **for** $i = 0 \ldots, m-1$ **do**
    Select action $a_{k,i} \sim \pi(\cdot|z_{k,i})$.
    Get environment state $s_{k,i+1} \sim T(\cdot|s_{k,i}, a_{k,i})$.
    Get reward $r_{k,i} = R(s_{k,i}, a_{k,i}, s_{k,i+1})$.
    Get observation $o_{k,i+1} \sim O(\cdot|s_{k,i+1})$.
    Update agent state $z_{k,i+1} \sim U(\cdot|z_{k,i}, a_{k,i}, o_{k,i+1})$.
  **end for**
  Sample last action $a_{k,m} \sim \pi(\cdot|z_{k,m})$.
  Compute semi-gradient $g_k$ according to equation (13) or equation (17).
  Update $\beta_{k+1} = \Gamma_{\mathcal{B}_2(0,B)}(\beta_k + \alpha g_k)$.
**end for**
**return:** average estimate $\overline{\mathcal{Q}}^\pi(\cdot) = \widehat{\mathcal{Q}}^\pi_{\bar\beta}(\cdot) = \langle \bar{\beta}, \phi(\cdot) \rangle$ or $\overline{Q}^\pi(\cdot) = \widehat{Q}^\pi_{\bar\beta}(\cdot) = \langle \bar{\beta}, \chi(\cdot) \rangle$ with $\bar{\beta} = \frac{1}{K}\sum_{k=0}^{K-1}\beta_k$.

---

### 3.2. Symmetric Critic

Similarly, we suppose that we are given features $\chi\colon \mathcal{Z} \times \mathcal{A} \to \mathbb{R}^{d_\chi}$. Without loss of generality, we assume $\sup_{z,a}\|\chi(z,a)\|_2 \leq 1$. Given a weight vector $\beta \in \mathbb{R}^{d_\chi}$, let $\widehat{Q}^\pi_\beta$ denote the linear approximation of the symmetric Q-function $Q^\pi$ that uses features $\chi$ with weight $\beta$,

$$\widehat{Q}^\pi_\beta(z,a) = \langle \beta, \chi(z,a) \rangle. \tag{14}$$

The corresponding hypothesis space for an arbitrary projection radius $B > 0$ is denoted with $\mathcal{F}^B_\chi$. The optimal parameter is also denoted by $\beta^\pi_* \in \arg\min_{\beta \in \mathcal{B}_2(0,B)} \|\langle \beta, \chi(\cdot) \rangle - Q^\pi(\cdot)\|_d$, the corresponding optimal approximation is $\widehat{Q}^\pi_* = \langle \beta^\pi_*, \chi(\cdot) \rangle$, and the corresponding error is,

$$\varepsilon_{\text{app}} = \min_{f \in \mathcal{F}^B_\chi} \left\| f - Q^\pi \right\|_d = \left\| \widehat{Q}^\pi_* - Q^\pi \right\|_d, \tag{15}$$

with $d(z,a) = \sum_{s \in \mathcal{S}} d^\pi(s,z)\pi(a|z)$ the sampling distribution.

Algorithm 1 also presents the $m$-step temporal difference learning algorithm for approximating the symmetric Q-function. The latter is identical to that of the asymmetric Q-function except that states are not exploited, such that the temporal difference $\delta_k$ and semi-gradient $g_k$ are given by,

$$\delta_k = \sum_{i=0}^{m-1} \gamma^i r_{k,i} + \gamma^m \widehat{Q}^\pi_{\beta_k}(z_{k,m}, a_{k,m})$$
$$- \widehat{Q}^\pi_{\beta_k}(z_{k,0}, a_{k,0}), \tag{16}$$
$$g_k = \delta_k \nabla_\beta \widehat{Q}^\pi_{\beta_k}(z_{k,0}, a_{k,0}). \tag{17}$$

At the end, the algorithm returns the average symmetric approximation $\overline{Q}^\pi = \widehat{Q}^\pi_{\bar\beta}$. Note that this symmetric critic

approximation and temporal difference learning algorithm corresponds to the one proposed by Cayci et al. (2024).

### 3.3. Natural Actor-Critic Algorithms

For both the asymmetric and symmetric actor-critic algorithms, we consider a log-linear agent-state policy $\pi_\theta \in \Pi_\mathcal{M}$. More precisely, the policy uses features $\psi\colon \mathcal{Z} \times \mathcal{A} \to \mathbb{R}^{d_\psi}$, with $\sup_{z,a} \|\psi(z,a)\|_2 \leq 1$ without loss of generality, and a softmax readout,

$$\pi_\theta(a_t|z_t) = \frac{\exp(\langle \theta, \psi(z_t, a_t) \rangle)}{\sum_{a \in \mathcal{A}} \exp(\langle \theta, \psi(z_t, a) \rangle)}. \qquad (18)$$

In this work, we consider natural policy gradients, which are less sensitive to policy parametrization (Kakade, 2001). Instead of computing the policy gradient in the original metric space, the idea is to compute the policy gradient on a statistical manifold, defined by the expected Fisher information metric. The natural policy gradient is thus given by the standard policy gradient multiplied by a preconditioner Fisher information matrix. Natural policy gradients are at the core of many effective modern policy-gradient methods (Schulman et al., 2015).

The natural policy gradient of policy $\pi_\theta \in \Pi_\mathcal{M}$ is defined as follows (Kakade, 2001),

$$w_*^{\pi_\theta} = (1 - \gamma) F_{\pi_\theta}^\dagger \nabla_\theta J(\pi_\theta), \qquad (19)$$

where $F_{\pi_\theta}^\dagger$ is the pseudoinverse of the Fisher information matrix, which is defined as the outer product of the score of the policy,

$$F_{\pi_\theta} = \mathbb{E}^{d^{\pi_\theta}} [\nabla_\theta \log \pi_\theta(A|Z) \otimes \nabla_\theta \log \pi_\theta(A|Z)]. \qquad (20)$$

As shown in Theorem 1, the natural policy gradient $w_*^{\pi_\theta}$ is the minimizer of the asymmetric objective (22).

**Theorem 1** (Asymmetric natural policy gradient). For any POMDP $\mathcal{P}$ and any agent-state policy $\pi_\theta \in \Pi_\mathcal{M}$, we have,

$$w_*^{\pi_\theta} = (1 - \gamma) F_{\pi_\theta}^\dagger \nabla_\theta J(\pi_\theta) \in \underset{w \in \mathbb{R}^{d_\psi}}{\arg\min} \mathcal{L}(w), \qquad (21)$$

with,

$$\mathcal{L}(w) = \mathbb{E}^{d^{\pi_\theta}} \left[ (\langle \nabla_\theta \log \pi_\theta(A|Z), w \rangle - \mathcal{A}^{\pi_\theta}(S, Z, A))^2 \right]. \qquad (22)$$

The proof is given in Appendix B. In practice, since the asymmetric advantage function is unknown, the algorithm estimates the natural policy gradient by stochastic gradient descent of $\mathcal{L}(\omega)$ using the approximation $\bar{\mathcal{A}}^{\pi_\theta}(S, Z, A) = \bar{\mathcal{Q}}^{\pi_\theta}(S, Z, A) - \bar{\mathcal{V}}^{\pi_\theta}(S, Z)$ with $\bar{\mathcal{V}}^{\pi_\theta} = \sum_{a \in \mathcal{A}} \pi_\theta(a|Z) \bar{\mathcal{Q}}(S, Z, a)$.

Our natural actor-critic algorithm generalizes the one of Cayci et al. (2024) to the asymmetric setting and is detailed in Algorithm 2. For each policy gradient step $0 \leq t < T$, the natural policy gradient $w_*^{\pi_t}$ is first estimated using $N$ steps of stochastic gradient descent. At each natural policy gradient estimation step $0 \leq n < N$, the algorithm samples an initial state $(s_{t,n}, z_{t,n}) \sim d^{\pi_t}$ from the discounted distribution $d^{\pi_t}$ and an action $a_{t,n} \sim \pi_t(\cdot|z_{t,n})$ according to the policy $\pi_t = \pi_{\theta_t}$. Then, the gradient $v_{t,n}$ of the natural policy gradient estimate $w_{t,n}$ is computed with,

$$v_{t,n} = \nabla_w \big( \langle \nabla_\theta \log \pi_\theta(a_{t,n}|z_{t,n}), w_{t,n} \rangle \\ - \bar{\mathcal{A}}^{\pi_\theta}(s_{t,n}, z_{t,n}, a_{t,n}) \big)^2, \qquad (23)$$

The gradient step is performed with $w_{t,n+1}^- = w_{t,n} - \zeta v_{t,n}$ and the parameters are projected onto the ball of radius $B$: $w_{t,n+1} = \Gamma_{\mathcal{B}_2(0,B)}(w_{t,n+1}^-)$. Finally, the algorithm computes the average parameter $\bar{w}_t = \frac{1}{N} \sum_{n=0}^{N-1} w_{t,n}$ and performs the policy gradient step: $\theta_{t+1} = \theta_t + \eta \bar{w}_t$. After all policy gradient steps, the final policy is returned.

---

**Algorithm 2** Natural actor-critic algorithm

**input:** number of updates $T$, number of steps $N$, step sizes $\zeta, \eta$, projection radius $B$.
Initialize $\theta_0 = 0$.
**for** $t = 0 \ldots T - 1$ **do**
    Obtain $\bar{\mathcal{Q}}^{\pi_t}$ or $\bar{Q}^{\pi_t}$ using Algorithm 1.
    Initialize $w_{t,0} = 0$.
    **for** $n = 0 \ldots N - 1$ **do**
        Initialize $(s_{t,n}, z_{t,n}) \sim d^{\pi_t}$.
        Sample $a_{t,n} \sim \pi_{\theta_t}(\cdot|z_{t,n})$.
        Compute the gradient $v_{t,n}$ of the policy gradient using equation (23) or equation (26).
        Update $w_{t,n+1}^- = w_{t,n} - \zeta v_{t,n}$.
        Project $w_{t,n+1} = \Gamma_{\mathcal{B}_2(0,B)}(w_{t,n+1}^-)$.
    **end for**
    Update $\theta_{t+1} = \theta_t + \eta \frac{1}{N} \sum_{n=0}^{N-1} w_{t,n}$.
**end for**
**return:** final policy $\pi_T = \pi_{\theta_T}$.

---

As shown in Theorem 2, the natural policy gradient $w_*^{\pi_\theta}$ is also the minimizer of the symmetric objective (25).

**Theorem 2** (Symmetric natural policy gradient). For any POMDP $\mathcal{P}$ and any agent-state policy $\pi_\theta \in \Pi_\mathcal{M}$, we have,

$$w_*^{\pi_\theta} = (1 - \gamma) F_{\pi_\theta}^\dagger \nabla_\theta J(\pi_\theta) \in \underset{w \in \mathbb{R}^{d_\psi}}{\arg\min} L(w), \qquad (24)$$

with,

$$L(w) = \mathbb{E}^{d^{\pi_\theta}} \left[ (\langle \nabla_\theta \log \pi_\theta(A|Z), w \rangle - A^{\pi_\theta}(Z, A))^2 \right]. \qquad (25)$$

The proof is given in Appendix B. As in the asymmetric case, the symmetric advantage function is unknown, and the algorithm estimates the natural gradient by stochastic gradient descent of equation (25) using the approximation $\overline{A}^{\pi_\theta}(Z, A) = \overline{Q}^{\pi_\theta}(Z, A) - \overline{V}^{\pi_\theta}(Z)$ with $\overline{V}^{\pi_\theta} = \sum_{a \in \mathcal{A}} \pi_\theta(a|Z) \overline{Q}^{\pi_\theta}(Z, a)$.

Algorithm 2 also presents the symmetric natural actor-critic algorithm, initially proposed by Cayci et al. (2024). The latter is similar to the asymmetric algorithm except that it uses the symmetric advantage function, such that the gradient of the policy gradient is given by,

$$v_{t,n} = \nabla_w \big( \langle \nabla_\theta \log \pi_\theta(a_{t,n}|z_{t,n}), w_{t,n} \rangle - \overline{A}^{\pi_\theta}(z_{t,n}, a_{t,n}) \big)^2. \quad (26)$$

While Theorem 1 and Theorem 2 show that $w_*^{\pi_\theta}$ is the minimizer of both the asymmetric and the symmetric objectives, the next section establishes the benefit of using the asymmetric loss. More precisely, asymmetric learning is shown to improve the estimation of the critic and thus the advantage function, which in turn results in a better estimation of the natural policy gradient.

## 4. Finite-Time Analysis

In this section, we give the finite-time bounds of the previous algorithms in both the asymmetric and symmetric cases. The bounds of the asymmetric and symmetric temporal difference learning algorithms are presented in Subsection 4.1 and Subsection 4.2, respectively. In Subsection 4.3, the bounds of the asymmetric and symmetric natural actor-critic algorithms are given.

We use $\|\mu - \nu\|_{\mathrm{TV}}$ to denote the total variation between two probability measures $\mu, \nu \in \Delta(\mathcal{X})$ over a discrete space $\mathcal{X}$,

$$\|\mu - \nu\|_{\mathrm{TV}} = \sup_{A \subseteq \mathcal{X}} |\mu(A) - \nu(A)| \quad (27)$$

$$= \frac{1}{2} \sum_{x \in \mathcal{X}} |\mu(x) - \nu(x)|. \quad (28)$$

### 4.1. Finite-Time Bound for the Asymmetric Critic

Our main result is to establish the following finite-time bound for the Q-function approximation resulting from the asymmetric temporal difference learning algorithm detailed in Algorithm 1.

**Theorem 3** (Finite-time bound for asymmetric $m$-step temporal difference learning). For any agent-state policy $\pi \in \Pi_\mathcal{M}$, and any $m \in \mathbb{N}$, we have for Algorithm 1 with $\alpha = \frac{1}{\sqrt{K}}$ and arbitrary $B > 0$,

$$\sqrt{\mathbb{E}\left[\left\|\mathcal{Q}^\pi - \overline{\mathcal{Q}}^\pi\right\|_d^2\right]} \le \varepsilon_{\mathrm{td}} + \varepsilon_{\mathrm{app}} + \varepsilon_{\mathrm{shift}}, \quad (29)$$

where the temporal difference learning, function approximation, and distribution shift terms are given by,

$$\varepsilon_{\mathrm{td}} = \sqrt{\frac{4B^2 + \left(\frac{1}{1-\gamma} + 2B\right)^2}{2\sqrt{K}(1 - \gamma^m)}} \quad (30)$$

$$\varepsilon_{\mathrm{app}} = \frac{1 + \gamma^m}{1 - \gamma^m} \min_{f \in \mathcal{F}_\phi^B} \|f - \mathcal{Q}^\pi\|_d \quad (31)$$

$$\varepsilon_{\mathrm{shift}} = \left(B + \frac{1}{1-\gamma}\right) \sqrt{\frac{2\gamma^m}{1 - \gamma^m}} \sqrt{\|d_m - d\|_{\mathrm{TV}}}, \quad (32)$$

with $d(s, z, a) = d^\pi(s, z)\pi(a|z)$ the sampling distribution, and $d_m(s, z, a) = d_m^\pi(s, z)\pi(a|z)$ the bootstrapping distribution.

The proof is given in Appendix C, and adapts the proof of Cayci et al. (2024) to the asymmetric setting. The first term $\varepsilon_{\mathrm{td}}$ is the usual temporal difference error term, decreasing in $K^{-1/4}$. The second term $\varepsilon_{\mathrm{app}}$ results from the use of linear function approximators. The third term $\varepsilon_{\mathrm{shift}}$ arises from the distribution shift between the sampling distribution $d^\pi \otimes \pi$ (i.e., the discounted visitation measure) and the bootstrapping distribution $d_m^\pi \otimes \pi$ (i.e., the distribution $m$ steps from the discounted visitation measure). It is a consequence of not assuming the existence of a stationary distribution nor assuming to sample from the stationary distribution.

### 4.2. Finite-Time Bound for the Symmetric Critic

Given a history $h_t = (o_0, a_0, \ldots, o_t)$, the belief is defined as,

$$b_t(s_t|h_t) = \Pr(S_t = s_t | H_t = h_t). \quad (33)$$

Given an agent state $z_t$, the approximate belief is defined as,

$$\hat{b}_t(s_t|z_t) = \Pr(S_t = s_t | Z_t = z_t). \quad (34)$$

We obtain the following finite-time bound for the Q-function approximation resulting from the symmetric temporal difference learning algorithm detailed in Algorithm 1.

**Theorem 4** (Finite-time bound for symmetric $m$-step temporal difference learning (Cayci et al., 2024)). For any agent-state policy $\pi \in \Pi_\mathcal{M}$, and any $m \in \mathbb{N}$, we have for Algorithm 1 with $\alpha = \frac{1}{\sqrt{K}}$, and arbitrary $B > 0$,

$$\sqrt{\mathbb{E}\left[\left\|Q^\pi - \overline{Q}^\pi\right\|_d^2\right]} \le \varepsilon_{\mathrm{td}} + \varepsilon_{\mathrm{app}} + \varepsilon_{\mathrm{shift}} + \varepsilon_{\mathrm{alias}}, \quad (35)$$

where the temporal difference learning, function approximation, distribution shift, and aliasing terms are given by,

$$\varepsilon_{\mathrm{td}} = \sqrt{\frac{4B^2 + \left(\frac{1}{1-\gamma} + 2B\right)^2}{2\sqrt{K}(1 - \gamma^m)}} \quad (36)$$

$$\varepsilon_{\mathrm{app}} = \frac{1+\gamma^m}{1-\gamma^m} \min_{f \in \mathcal{F}_\chi^B} \|f - Q^\pi\|_d \tag{37}$$

$$\varepsilon_{\mathrm{shift}} = \left(B + \frac{1}{1-\gamma}\right) \sqrt{\frac{2\gamma^m}{1-\gamma^m}} \sqrt{\|d_m - d\|_{\mathrm{TV}}} \tag{38}$$

$$\varepsilon_{\mathrm{alias}} = \frac{2}{1-\gamma} \left\| \mathbb{E}^\pi \left[ \sum_{k=0}^\infty \gamma^{km} \left\| \hat{b}_{km} - b_{km} \right\|_{\mathrm{TV}} \bigg| Z_0 = \cdot \right] \right\|_d, \tag{39}$$

with $d(z,a) = \sum_{s \in \mathcal{S}} d^\pi(s,z)\pi(a|z)$ the sampling distribution, and $d_m(z,a) = \sum_{s \in \mathcal{S}} d_m^\pi(s,z)\pi(a|z)$ the bootstrapping distribution.

The first three terms are identical or analogous to the asymmetric case. The fourth term $\varepsilon_{\mathrm{alias}}$ results from the difference between the fixed point $\widetilde{Q}^\pi$ of the symmetric Bellman operator (7) and the true Q-function $Q^\pi$.

We note some minor differences with respect to the original result of Cayci et al. (2024) that appear to be typos and minor mistakes in the original proof.[1] We provide the corrected proof in Appendix D.

The results of Theorem 3 and Theorem 4 can be straightforwardly generalized to any other sampling distribution. However, obtaining bounds in term of $d^\pi \otimes \pi$ is useful for bounding the performance of the actor-critic algorithm.

### 4.3. Finite-Time Bound for the Natural Actor-Critic

Following Cayci et al. (2024), we assume that there exists a concentrability coefficient $\overline{C}_\infty < \infty$ such that $\sup_{0 \le t < T} \mathbb{E}[C_t] \le \overline{C}_\infty$ with,

$$C_t = \sup_{s,z,a} \left| \frac{d^{\pi^*}(s,z)\pi^*(a|z)}{d^{\pi_{\theta_t}}(s,z)\pi_{\theta_t}(a|z)} \right|. \tag{40}$$

Roughly speaking, this assumption means that all successive policies should visit every agent states and actions visited by the optimal policy with nonzero probability. It motivates the log-linear policy parametrization in equation (18) and the initialization to the maximum entropy policy in Algorithm 2. We obtain the following finite-time bound for the suboptimality of the policy resulting from Algorithm 2.

**Theorem 5** (Finite-time bound for asymmetric and symmetric natural actor-critic algorithm). *For any agent-state process* $\mathcal{M} = (\mathcal{Z}, U)$, *we have for Algorithm 2 with* $\alpha = \frac{1}{\sqrt{K}}$, $\zeta = \frac{B\sqrt{1-\gamma}}{\sqrt{2N}}$, $\eta = \frac{1}{\sqrt{T}}$ *and arbitrary* $B > 0$,

$$(1-\gamma) \min_{0 \le t < T} \mathbb{E}\left[J(\pi^*) - J(\pi_t)\right] \le \varepsilon_{\mathrm{nac}} + 2\varepsilon_{\mathrm{inf}}$$

$$+ \overline{C}_\infty \left( \varepsilon_{\mathrm{actor}} + 2\varepsilon_{\mathrm{grad}} + 2\sqrt{6}\frac{1}{T}\sum_{t=0}^{T-1} \varepsilon_{\mathrm{critic}}^{\pi_t} \right), \tag{41}$$

---

[1] The authors notably wrongly bound the distance $\|\widehat{Q}_*^\pi - \widetilde{Q}^\pi\|_d$ by $\varepsilon_{\mathrm{app}}$ at one point, which nevertheless yields a similar result.

where the different terms may differ for asymmetric and symmetric critics,

$$\varepsilon_{\mathrm{nac}} = \frac{B^2 + 2\log|\mathcal{A}|}{2\sqrt{T}} \tag{42}$$

$$\varepsilon_{\mathrm{actor}} = \sqrt{\frac{(2-\gamma)B}{(1-\gamma)\sqrt{N}}} \tag{43}$$

$$\varepsilon_{\mathrm{inf,asym}} = 0 \tag{44}$$

$$\varepsilon_{\mathrm{inf,sym}} = \mathbb{E}^{\pi^*}\left[ \sum_{k=0}^\infty \gamma^k \left\| \hat{b}_k - b_k \right\|_{\mathrm{TV}} \right] \tag{45}$$

$$\varepsilon_{\mathrm{grad,asym}} = \sup_{0 \le t < T} \sqrt{\min_w \mathcal{L}_t(w)} \tag{46}$$

$$\varepsilon_{\mathrm{grad,sym}} = \sup_{0 \le t < T} \sqrt{\min_w L_t(w)}, \tag{47}$$

and $\varepsilon_{\mathrm{critic}}^{\pi_t}$ is given in Theorem 3 and Theorem 4.

The first term $\varepsilon_{\mathrm{nac}}$ is the usual natural actor-critic term decreasing in $T^{-1/2}$ (Agarwal et al., 2021). The second term $\varepsilon_{\mathrm{inf}}$ is the inference error resulting from use of an agent state in a POMDP (Cayci et al., 2024). This term is zero for the asymmetric algorithm. The third term $\varepsilon_{\mathrm{actor}}$ is the error resulting from the estimation of the natural policy gradient by stochastic gradient descent. The fourth term $\varepsilon_{\mathrm{grad}}$ is the error resulting from the use of a linear function approximator with features $\nabla_\theta \log \pi_t(a|z)$ for the natural policy gradient. Finally, the fifth term $\frac{1}{T}\sum_{t=0}^{T-1} \varepsilon_{\mathrm{critic}}^{\pi_t}$ is the error arising from the successive critic approximations. Inside of each $\varepsilon_{\mathrm{critic}}^{\pi_t}$ terms, the aliasing term is thus zero for the asymmetric algorithm. The proof, generalizing that of Cayci et al. (2024) to the asymmetric setting, is available in Appendix E.

### 4.4. Discussion

As can be seen from Theorem 3 and Theorem 4, compared to the symmetric temporal difference learning algorithm, the asymmetric one eliminates a term arising from aliasing in the agent state, in the sense of equation (39). In other words, even for an aliased agent-state process, leveraging the state to learn the asymmetric Q-function instead of the symmetric Q-function does not suffer from aliasing, while still providing a valid critic for the policy gradient algorithm. That said, these bounds are given in expectation, and future works may want to study the variance of the error of such Q-function approximations.

From Theorem 5, we notice that the inference term (45) in the suboptimality bound vanishes in the asymmetric setting. Moreover, the average error $\frac{1}{T}\sum_{t=0}^{T-1} \varepsilon_{\mathrm{critic}}^{\pi_t}$ made in the evaluation of all policies $\pi_0, \ldots, \pi_{t-1}$ appears in the finite-time bound that we obtain for the suboptimality of the policy. Thus, the suboptimality bound for the actor also improves in the asymmetric setting by eliminating the aliasing terms with respect to the symmetric setting.

By diving into the proof of Theorem 5 at equations (236) and (237), we understand that the Q-function error impacts the suboptimality bound through the estimation of the natural policy gradient (19). Indeed, this error term in the suboptimality bound directly results from the error on the advantage function estimation used in the target of the natural policy gradient estimation loss of equations (23) and (26). This advantage function estimation is derived from the estimation of the Q-function, such that the error on the latter directly impacts the error on the former, as detailed in equations (236) and (237). This improvement in the average critic error unfortunately comes at the expense of a different residual error $\varepsilon_{\mathrm{grad}}$ on the natural policy gradient loss. Indeed, as can be seen in equation (47), we obtain a residual error $\varepsilon_{\mathrm{grad,asym}}$ using the best approximation of the asymmetric advantage $\mathcal{A}^{\pi_t}(s,z,a)$, instead of a residual error $\varepsilon_{\mathrm{grad,sym}}$ using the best approximation of the symmetric critic $A^{\pi_t}(z,a)$. Since both natural policy gradients are obtained through a linear regression with features $\nabla_\theta \log \pi_t(a|z)$, it is clear than the asymmetric residual error may be higher than the symmetric residual error, even in the tabular case.

We conclude that the effectiveness of asymmetric actor-critic algorithms notably results from a better approximation of the Q-function by eliminating the aliasing bias, which in turn provides a better estimate of the policy gradient.

## 5. Conclusion

In this work, we extended the unbiased asymmetric actor-critic algorithm to agent-state policies. Then, we adapted a finite-time analysis for natural actor-critic to the asymmetric setting. This analysis highlighted that on the contrary to symmetric learning, asymmetric learning is less sensitive to aliasing in the agent state. While this analysis assumed a fixed agent-state process, we argue that it is useful to interpret the causes of effectiveness of asymmetric learning with learnable agent-state processes. Indeed, aliasing can be present in the agent-state process throughout learning, and in particular at initialization. Moreover, it should be noted that this analysis can be straightforwardly generalized to learnable agent-state processes by extending the action space to select future agent states. More formally, we would extend the action space to $\mathcal{A}^+ = \mathcal{A} \times \Delta(\mathcal{Z})$ with $a_t^+ = (a_t, a_t^z)$, the agent state space to $\mathcal{Z}^+ = \mathcal{Z} \times \mathcal{O}$ with $z_t^+ = (z_t, z_t^o)$, and the agent-state process to $U(z_{t+1}^+|z_t^+, a_t, o_{t+1}) \propto \exp(a_t^{z_{t+1}}) \delta_{z_{t+1}^o, o_{t+1}}$. This alternative to backpropagation through time would nevertheless still not reflect the common setting of recurrent actor-critic algorithms. We consider this as a future work that could build on recent advances in finite-time bound for recurrent actor-critic algorithms (Cayci & Eryilmaz, 2024a;b). Alternatively, generalizing this analysis to nonlinear approximators may include recurrent neural networks, which can

be seen as nonlinear approximators with a sliding window as agent state. Our analysis also motivates future work studying other asymmetric learning approaches that consider representation losses to reduce the aliasing bias (Sinha & Mahajan, 2023; Lambrechts et al., 2022; 2024).

## Acknowledgements

Gaspard Lambrechts acknowledges the financial support of the *Wallonia-Brussels Federation* for his FRIA grant.

## Impact Statement

This paper presents work whose goal is to advance the field of machine learning. There are many potential societal consequences of our work, none which we feel must be specifically highlighted here.

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

## A. Agent State Aliasing

In this section, we provide an example of aliased agent state, and discuss the corresponding aliasing bias. For this purpose, we introduce a slightly modified version of the Tiger POMDP (Kaelbling et al., 1998), see Figure 1. In this POMDP, there are two doors: one opening on a room with a treasure on the left, and another opening on a room with a tiger on the right. There are four states for this POMDP: being in the treasure room (Treasure), being in the tiger room (Tiger), being in front of the treasure door (Left) or being in front of the tiger door (Right). The rooms are labeled outside (Left or Right), but inside it is completely dark (Dark), such that we do not observe in which room we are. When outside of the rooms, the agent can switch to the other door (Swap) or it can open the door and enter the room (Enter). Once in a room (Treasure or Tiger), the agent stays locked forever, and gets a positive reward (+1) if it is in the treasure room (Treasure) whatever the action taken (Swap or Enter). We consider the agent state to be

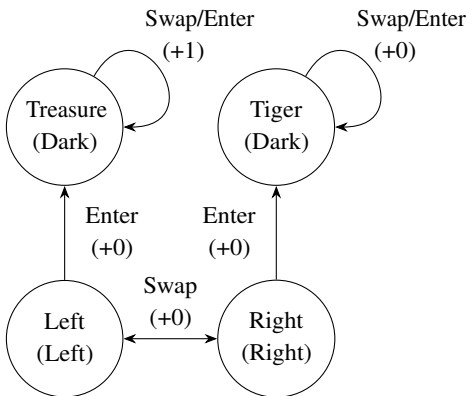

Figure 1: Aliased Tiger POMDP.

simply the last observation (Left, Right, or Dark). Notice that the optimal agent-state policy conditioned on this agent state is also an optimal history-dependent policy. In other words, the current observation is a sufficient statistic for optimal control in this POMDP. We consider a uniform initial distributions over the four states.

For a given agent state (Dark), there exist two different underlying states (Treasure or Tiger). We call this phenomenon aliasing. Now, let us consider a simple policy $\pi$ that always takes the same action (Enter). It is clear that the symmetric value function defined according to equation (6) is given by $V^\pi(z=\text{Dark}) = \frac{1}{2(1-\gamma)}$, $V^\pi(z=\text{Left}) = \frac{\gamma}{1-\gamma}$, and $V^\pi(z=\text{Right}) = 0$. However, when considering the unique fixed point of the aliased Bellman operator of equation (7) with $m=1$, we have instead $\widetilde{V}^\pi(z=\text{Dark}) = \frac{1}{2(1-\gamma)}$, $\widetilde{V}^\pi(z=\text{Left}) = \frac{\gamma}{2(1-\gamma)}$, and $\widetilde{V}^\pi(z=\text{Right}) = \frac{\gamma}{2(1-\gamma)}$. We refer to the distance between $V^\pi$ and $\widetilde{V}^\pi$, or similarly $Q^\pi$ and $\widetilde{Q}^\pi$, as the aliasing bias. In the analysis of this paper, this distance appears as the weighted $\ell_2$-norm $\|Q^\pi - \widetilde{Q}^\pi\|_d$ where $d(s,z,a) = d^\pi(s,z)\pi(a|z)$. In the analysis, we also define the aliasing term $\varepsilon_{\text{alias}}$ as an upper bound on this aliasing bias, see Lemma D.1 for a detailed definition.

## B. Proof of the Natural Policy Gradients

In this section, we prove that the natural policy gradient is the minimizer of analogous asymmetric and symmetric losses.

### B.1. Proof of the Asymmetric Natural Policy Gradient

In this section, we prove that the natural policy gradient is the minimizer of an asymmetric loss.

**Theorem 1** (Asymmetric natural policy gradient). For any POMDP $\mathcal{P}$ and any agent-state policy $\pi_\theta \in \Pi_\mathcal{M}$, we have,

$$w_*^{\pi_\theta} = (1-\gamma)F_{\pi_\theta}^\dagger \nabla_\theta J(\pi_\theta) \in \underset{w \in \mathbb{R}^{d_\psi}}{\arg\min}\, \mathcal{L}(w), \tag{21}$$

with,

$$\mathcal{L}(w) = \mathbb{E}^{d^{\pi_\theta}}\Big[ \big( \langle \nabla_\theta \log \pi_\theta(A|Z), w \rangle - \mathcal{A}^{\pi_\theta}(S,Z,A) \big)^2 \Big]. \tag{22}$$

*Proof.* Let us note that,

$$\nabla_w \mathcal{L}(w) = 2\mathbb{E}^{d^{\pi_\theta}}\big[ \nabla_\theta \log \pi_\theta(A|Z)\, (\langle \nabla_\theta \log \pi_\theta(A|Z), w \rangle - \mathcal{A}^{\pi_\theta}(S,Z,A)) \big]. \tag{48}$$

Therefore, for any $w_*^{\pi_\theta} \in \mathbb{R}^{d_\psi}$ minimizing $\mathcal{L}(w)$, we have $\nabla_w \mathcal{L}(w) = 0$, such that,

$$\mathbb{E}^{d^{\pi_\theta}}\big[ \nabla_\theta \log \pi_\theta(A|Z) \mathcal{A}^{\pi_\theta}(S,Z,A) \big] = \mathbb{E}^{d^{\pi_\theta}}\big[ \nabla_\theta \log \pi_\theta(A|Z) \langle \nabla_\theta \log \pi_\theta(A|Z), w_*^{\pi_\theta} \rangle \big] \tag{49}$$

$$= \mathbb{E}^{d^{\pi_\theta}}\big[ (\nabla_\theta \log \pi_\theta(A|Z) \otimes \nabla_\theta \log \pi_\theta(A|Z)) w_*^{\pi_\theta} \big] \tag{50}$$

$$= \mathbb{E}^{d^{\pi_\theta}}\big[ \nabla_\theta \log \pi_\theta(A|Z) \otimes \nabla_\theta \log \pi_\theta(A|Z) \big] w_*^{\pi_\theta} \tag{51}$$

$$= F_{\pi_\theta} w_*^{\pi_\theta}. \tag{52}$$

which follows from the definition of the Fisher information matrix $F_{\pi_\theta}$ in equation (20). Now, let us define the policy $\pi_\theta^+(A|S,Z) = \pi_\theta(A|Z)$, which ignores the state $S$. From there, we have,

$$F_{\pi_\theta} w_*^{\pi_\theta} = \mathbb{E}^{d^{\pi_\theta}} \left[ \nabla_\theta \log \pi_\theta(A|Z) \mathcal{A}(S,Z,A) \right] \tag{53}$$

$$= \mathbb{E}^{d^{\pi_\theta^+}} \left[ \nabla_\theta \log \pi_\theta^+(A|S,Z) \mathcal{A}(S,Z,A) \right] \tag{54}$$

$$= \mathbb{E}^{d^{\pi_\theta^+}} \left[ \nabla_\theta \log \pi_\theta^+(A|S,Z) (\mathcal{A}(S,Z,A) + \mathcal{V}(S,Z) - \mathcal{V}(S,Z)) \right] \tag{55}$$

$$= \mathbb{E}^{d^{\pi_\theta^+}} \left[ \nabla_\theta \log \pi_\theta^+(A|S,Z) \mathcal{Q}(S,Z,A) \right] - \mathbb{E}^{d^{\pi_\theta^+}} \left[ \nabla_\theta \log \pi_\theta^+(A|S,Z) \mathcal{V}(S,Z) \right] \tag{56}$$

$$= \mathbb{E}^{d^{\pi_\theta^+}} \left[ \nabla_\theta \log \pi_\theta^+(A|S,Z) \mathcal{Q}(S,Z,A) \right] - \mathbb{E}^{d^{\pi_\theta^+}} \left[ \mathcal{V}(S,Z) \sum_{a \in \mathcal{A}} \pi_\theta^+(a|S,Z) \nabla_\theta \log \pi_\theta^+(a|S,Z) \right] \tag{57}$$

$$= \mathbb{E}^{d^{\pi_\theta^+}} \left[ \nabla_\theta \log \pi_\theta^+(A|S,Z) \mathcal{Q}(S,Z,A) \right] - \mathbb{E}^{d^{\pi_\theta^+}} \left[ \mathcal{V}(S,Z) \sum_{a \in \mathcal{A}} \nabla_\theta \pi_\theta^+(a|S,Z) \right] \tag{58}$$

$$= \mathbb{E}^{d^{\pi_\theta^+}} \left[ \nabla_\theta \log \pi_\theta^+(A|S,Z) \mathcal{Q}(S,Z,A) \right] - \mathbb{E}^{d^{\pi_\theta^+}} \left[ \mathcal{V}(S,Z) \nabla_\theta \sum_{a \in \mathcal{A}} \pi_\theta^+(a|S,Z) \right] \tag{59}$$

$$= \mathbb{E}^{d^{\pi_\theta^+}} \left[ \nabla_\theta \log \pi_\theta^+(A|S,Z) \mathcal{Q}(S,Z,A) \right] - \mathbb{E}^{d^{\pi_\theta^+}} \left[ \mathcal{V}(S,Z) \nabla_\theta 1 \right] \tag{60}$$

$$= \mathbb{E}^{d^{\pi_\theta^+}} \left[ \nabla_\theta \log \pi_\theta^+(A|S,Z) \mathcal{Q}(S,Z,A) \right]. \tag{61}$$

Using the policy gradient theorem (Sutton et al., 1999) and equation (61),

$$F_{\pi_\theta} w_*^{\pi_\theta} = (1-\gamma) \nabla_\theta J(\pi_\theta^+), \tag{62}$$

From there, we obtain using the definition of $\pi_\theta^+$,

$$F_{\pi_\theta} w_*^{\pi_\theta} = (1-\gamma) \nabla_\theta J(\pi_\theta^+) \tag{63}$$

$$= (1-\gamma) \nabla_\theta J(\pi_\theta). \tag{64}$$

This concludes the proof. $\qquad \square$

### B.2. Proof of the Symmetric Natural Policy Gradient

In this section, we prove that the natural policy gradient is the minimizer of an asymmetric loss.

**Theorem 2** (Symmetric natural policy gradient). For any POMDP $\mathcal{P}$ and any agent-state policy $\pi_\theta \in \Pi_\mathcal{M}$, we have,

$$w_*^{\pi_\theta} = (1-\gamma) F_{\pi_\theta}^\dagger \nabla_\theta J(\pi_\theta) \in \underset{w \in \mathbb{R}^{d_\psi}}{\arg\min} L(w), \tag{24}$$

with,

$$L(w) = \mathbb{E}^{d^{\pi_\theta}} \left[ \left( \langle \nabla_\theta \log \pi_\theta(A|Z), w \rangle - A^{\pi_\theta}(Z,A) \right)^2 \right]. \tag{25}$$

*Proof.* Similarly to the asymmetric setting, for any $w_*^{\pi_\theta}$ minimizing $L(w)$, we have $\nabla_w L(w) = 0$, such that,

$$\mathbb{E}^{d^{\pi_\theta}} \left[ \nabla_\theta \log \pi_\theta(A|Z) A(Z,A) \right] = \mathbb{E}^{d^{\pi_\theta}} \left[ \nabla_\theta \log \pi_\theta(A|Z) \langle \nabla_\theta \log \pi_\theta(A|Z) w_*^{\pi_\theta} \rangle \right] \tag{65}$$

$$= \mathbb{E}^{d^{\pi_\theta}} \left[ (\nabla_\theta \log \pi_\theta(A|Z) \otimes \nabla_\theta \log \pi_\theta(A|Z)) w_*^{\pi_\theta} \right] \tag{66}$$

$$= \mathbb{E}^{d^{\pi_\theta}} \left[ \nabla_\theta \log \pi_\theta(A|Z) \otimes \nabla_\theta \log \pi_\theta(A|Z) \right] w_*^{\pi_\theta} \tag{67}$$

$$= F_{\pi_\theta} w_*^{\pi_\theta}, \tag{68}$$

which follows from the definition of the Fisher information matrix $F_{\pi_\theta}$ in equation (20). From there, we have,

$$F_{\pi_\theta} w_*^{\pi_\theta} = \mathbb{E}^{d^{\pi_\theta}} \left[ \nabla_\theta \log \pi_\theta(A|Z) A(Z, A) \right] \tag{69}$$

$$F_{\pi_\theta} w_*^{\pi_\theta} = \mathbb{E}^{d^{\pi_\theta}} \left[ \nabla_\theta \log \pi_\theta(A|Z) \mathbb{E}^{d^{\pi_\theta}} [\mathcal{A}(S, Z, A)|Z, A] \right] \tag{70}$$

$$F_{\pi_\theta} w_*^{\pi_\theta} = \mathbb{E}^{d^{\pi_\theta}} \left[ \mathbb{E}^{d^{\pi_\theta}} [\nabla_\theta \log \pi_\theta(A|Z) \mathcal{A}(S, Z, A)|Z, A] \right] \tag{71}$$

$$F_{\pi_\theta} w_*^{\pi_\theta} = \mathbb{E}^{d^{\pi_\theta}} \left[ \nabla_\theta \log \pi_\theta(A|Z) \mathcal{A}(S, Z, A) \right], \tag{72}$$

which follows from the law of total probability. From there, by following the same steps as in the asymmetric case (see Subsection B.1), we obtain,

$$F_{\pi_\theta} w_*^{\pi_\theta} = (1 - \gamma) \nabla_\theta J(\pi_\theta). \tag{73}$$

This concludes the proof. $\qquad\square$

## C. Proof of the Finite-Time Bound for the Asymmetric Critic

In this section, we prove Theorem 3, that is recalled below.

**Theorem 3** (Finite-time bound for asymmetric $m$-step temporal difference learning). For any agent-state policy $\pi \in \Pi_\mathcal{M}$, and any $m \in \mathbb{N}$, we have for Algorithm 1 with $\alpha = \frac{1}{\sqrt{K}}$ and arbitrary $B > 0$,

$$\sqrt{\mathbb{E}\left[\left\|\mathcal{Q}^\pi - \overline{\mathcal{Q}}^\pi\right\|_d^2\right]} \leq \varepsilon_{\text{td}} + \varepsilon_{\text{app}} + \varepsilon_{\text{shift}}, \tag{29}$$

where the temporal difference learning, function approximation, and distribution shift terms are given by,

$$\varepsilon_{\text{td}} = \sqrt{\frac{4B^2 + \left(\frac{1}{1-\gamma} + 2B\right)^2}{2\sqrt{K}(1 - \gamma^m)}} \tag{30}$$

$$\varepsilon_{\text{app}} = \frac{1 + \gamma^m}{1 - \gamma^m} \min_{f \in \mathcal{F}_\phi^B} \|f - \mathcal{Q}^\pi\|_d \tag{31}$$

$$\varepsilon_{\text{shift}} = \left(B + \frac{1}{1 - \gamma}\right) \sqrt{\frac{2\gamma^m}{1 - \gamma^m}} \sqrt{\|d_m - d\|_{\text{TV}}}, \tag{32}$$

with $d(s, z, a) = d^\pi(s, z) \pi(a|z)$ the sampling distribution, and $d_m(s, z, a) = d_m^\pi(s, z) \pi(a|z)$ the bootstrapping distribution.

*Proof.* To simplify notation, we drop the dependence on $\pi$ and $\beta$ and use $\mathcal{Q}$ as a shorthand for $\mathcal{Q}^\pi$, $\widehat{Q}^*$ as a shorthand for $\widehat{Q}_*^\pi$, $\overline{\mathcal{Q}}$ as a shorthand for $\overline{\mathcal{Q}}^\pi$ and $\widehat{\mathcal{Q}}_k$ as a shorthand for $\widehat{\mathcal{Q}}_{\beta_k}^\pi$, where the subscripts and superscripts remain implicit but are assumed clear from context. When evaluating the Q-functions, we go one step further by using $\mathcal{Q}_{k,i}$ to denote $\mathcal{Q}(S_{k,i}, Z_{k,i}, A_{k,i})$, $\widehat{Q}_{k,i}^*$ to denote $\widehat{Q}^*(Z_{k,i}, A_{k,i})$ or $\widehat{\mathcal{Q}}_{k,i}$ to denote $\widehat{\mathcal{Q}}_k(S_{k,i}, Z_{k,i}, A_{k,i})$, and $\phi_{k,i}$ to denote $\phi(S_{k,i}, Z_{k,i}, A_{k,i})$. In addition, we define $d$ as a shorthand for $d^\pi \otimes \pi$, such that $d(s, z, a) = d^\pi(s, z) \pi(a|z)$, and $d_m$ as a shorthand for $d_m^\pi \otimes \pi$, such that $d_m(s, z, a) = d_m^\pi(s, z) \pi(a|z)$.

First, let us define $\Delta_k$ as,

$$\Delta_k = \sqrt{\mathbb{E}\left[\left\|\mathcal{Q} - \widehat{\mathcal{Q}}_k\right\|_d^2\right]} = \sqrt{\mathbb{E}\left[\left\|\mathcal{Q}(\cdot) - \langle \beta_k, \phi(\cdot)\rangle\right\|_d^2\right]}. \tag{74}$$

Using the linearity of $\overline{\mathcal{Q}}$ in $\beta_1, \ldots, \beta_{K-1}$, the triangle inequality, the subadditivity of the square root, and Jensen's inequality, we have,

$$\sqrt{\mathbb{E}\left[\left\|\mathcal{Q} - \overline{\mathcal{Q}}\right\|_d^2\right]} = \sqrt{\mathbb{E}\left[\left\|\mathcal{Q}(\cdot) - \left\langle \frac{1}{K} \sum_{k=0}^{K-1} \beta_k, \phi(\cdot)\right\rangle\right\|_d^2\right]} \tag{75}$$

$$= \sqrt{\mathbb{E}\left[\left\|\frac{1}{K}\sum_{k=0}^{K-1}\left(\mathcal{Q}(\cdot)-\langle\beta_k,\phi(\cdot)\rangle\right)\right\|_d^2\right]} \tag{76}$$

$$= \sqrt{\mathbb{E}\left[\left\|\sum_{k=0}^{K-1}\frac{1}{K}\left(\mathcal{Q}(\cdot)-\langle\beta_k,\phi(\cdot)\rangle\right)\right\|_d^2\right]} \tag{77}$$

$$\leq \sqrt{\mathbb{E}\left[\sum_{k=0}^{K-1}\frac{1}{K^2}\left\|\mathcal{Q}(\cdot)-\langle\beta_k,\phi(\cdot)\rangle\right\|_d^2\right]} \tag{78}$$

$$= \sqrt{\frac{1}{K^2}\sum_{k=0}^{K-1}\mathbb{E}\left[\left\|\mathcal{Q}(\cdot)-\langle\beta_k,\phi(\cdot)\rangle\right\|_d^2\right]} \tag{79}$$

$$= \frac{1}{K}\sqrt{\sum_{k=0}^{K-1}\Delta_k^2} \tag{80}$$

$$\leq \frac{1}{K}\sum_{k=0}^{K-1}\sqrt{\Delta_k^2} \tag{81}$$

$$= \frac{1}{K}\sum_{k=0}^{K-1}\Delta_k \tag{82}$$

$$= \frac{1}{K}\sum_{k=0}^{K-1}(\Delta_k-l)+l \tag{83}$$

$$\leq \sqrt{\left(\frac{1}{K}\sum_{k=0}^{K-1}(\Delta_k-l)\right)^2}+l \tag{84}$$

$$\leq \sqrt{\frac{1}{K}\sum_{k=0}^{K-1}(\Delta_k-l)^2}+l, \tag{85}$$

where $l$ is arbitrary.

Now, we consider the Lyapounov function $\mathcal{L}(\beta)=\|\beta_*-\beta\|_2^2$ in order to find a bound on $\frac{1}{K}\sum_{k=0}^{K-1}(\Delta_k-l)^2$. Since $\beta_*\in\mathcal{B}_2(0,B)$, with $\mathcal{B}_2(0,B)$ a convex subset of $\mathbb{R}^{d_\phi}$, and the projection $\Gamma_\mathcal{C}$ is non-expansive for closed and convex $\mathcal{C}$, we have for all $k\geq 0$,

$$\mathcal{L}(\beta_{k+1})=\|\beta_*-\beta_{k+1}\|_2^2 \tag{86}$$

$$\leq \|\beta_*-\beta_{k+1}^-\|_2^2 \tag{87}$$

$$= \|\beta_*-(\beta_k+\alpha g_k)\|_2^2 \tag{88}$$

$$= \|(\beta_*-\beta_k)-\alpha g_k\|_2^2 \tag{89}$$

$$= \langle(\beta_*-\beta_k)-\alpha g_k,(\beta_*-\beta_k)-\alpha g_k\rangle \tag{90}$$

$$= \langle\beta_*-\beta_k,\beta_*-\beta_k\rangle-2\alpha\langle\beta_*-\beta_k,g_k\rangle+\alpha^2\langle g_k,g_k\rangle \tag{91}$$

$$= \mathcal{L}(\beta_k)-2\alpha\langle\beta_*-\beta_k,g_k\rangle+\alpha^2\|g_k\|_2^2 \tag{92}$$

$$= \mathcal{L}(\beta_k)+2\alpha\langle\beta_k-\beta_*,g_k\rangle+\alpha^2\|g_k\|_2^2. \tag{93}$$

Let us consider the Lyapounov drift $\mathbb{E}\left[\mathcal{L}(\beta_{k+1})-\mathcal{L}(\beta_k)\right]$, and exploit the fact that environments samples used to compute $g_k$ are independent and identically distributed. Formally, we define $\mathfrak{G}_k=\sigma(S_{i,j},Z_{i,j},A_{i,j},i\leq k,j\leq m)$ and $\mathfrak{F}_k=\sigma(S_{k,0},Z_{k,0},A_{k,0})$, where $\sigma(X_i:i\in\mathcal{I})$ denotes the $\sigma$-algebra generated by a collection $\{X_i:i\in\mathcal{I}\}$ of random

variables. We can write, using to the law of total expectation,

$$\mathbb{E}\left[\mathcal{L}(\beta_{k+1}) - \mathcal{L}(\beta_k)\right] = \mathbb{E}\left[\mathbb{E}\left[\mathcal{L}(\beta_{k+1}) - \mathcal{L}(\beta_k)|\mathfrak{G}_{k-1}\right]\right] \tag{94}$$

$$\leq 2\alpha\mathbb{E}\left[\mathbb{E}\left[\langle\beta_k - \beta_*, g_k\rangle|\mathfrak{G}_{k-1}\right]\right] + \alpha^2\mathbb{E}\left[\mathbb{E}\left[\|g_k\|_2^2\Big|\mathfrak{G}_{k-1}\right]\right]. \tag{95}$$

Let us focus on the first term of equation (95) with $\mathbb{E}\left[\langle g_k, \beta_k - \beta_*\rangle|\mathfrak{G}_{k-1}\right]$. First, since $\nabla_\beta\widehat{\mathcal{Q}}_{k,0} = \phi_{k,0}$, the semi-gradient $g_k$ is given by (see equation (13)),

$$g_k = \left(\sum_{t=0}^{m-1}\gamma^t R_{k,t} + \gamma^m\widehat{\mathcal{Q}}_{k,m} - \widehat{\mathcal{Q}}_{k,0}\right)\phi_{k,0}. \tag{96}$$

By conditioning on the sigma-fields $\mathfrak{G}_{k-1}$ and $\mathfrak{F}_k$, we have,

$$\mathbb{E}\left[\langle\beta_k - \beta_*, g_k\rangle|\mathfrak{F}_k, \mathfrak{G}_{k-1}\right] = \left(\mathbb{E}\left[\sum_{t=0}^{m-1}\gamma^t R_{k,t} + \gamma^m\widehat{\mathcal{Q}}_{k,m}\Big|\mathfrak{F}_k, \mathfrak{G}_{k-1}\right] - \widehat{\mathcal{Q}}_{k,0}\right)\langle\beta_k - \beta_*, \phi_{k,0}\rangle \tag{97}$$

$$= \left(\mathbb{E}\left[\sum_{t=0}^{m-1}\gamma^t R_{k,t} + \gamma^m\widehat{\mathcal{Q}}_{k,m}\Big|\mathfrak{F}_k, \mathfrak{G}_{k-1}\right] - \widehat{\mathcal{Q}}_{k,0}\right)\left(\widehat{\mathcal{Q}}_{k,0} - \widehat{\mathcal{Q}}_{k,0}^*\right). \tag{98}$$

Note that according to the Bellman operator (5) we have,

$$\mathbb{E}\left[\sum_{t=0}^{m-1}\gamma^t R_{k,t}\Big|\mathfrak{F}_k, \mathfrak{G}_{k-1}\right] = \mathcal{Q}_{k,0} - \gamma^m\mathbb{E}\left[\mathcal{Q}_{k,m}|\mathfrak{F}_k, \mathfrak{G}_{k-1}\right]. \tag{99}$$

By substituting equation (99) in equation (98), we obtain,

$$\mathbb{E}\left[\langle\beta_k - \beta_*, g_k\rangle|\mathfrak{F}_k, \mathfrak{G}_{k-1}\right]$$

$$= \left(\mathbb{E}\left[\sum_{t=0}^{m-1}\gamma^t R_{k,t}\Big|\mathfrak{F}_k, \mathfrak{G}_{k-1}\right] + \gamma^m\mathbb{E}\left[\widehat{\mathcal{Q}}_{k,m}\Big|\mathfrak{F}_k, \mathfrak{G}_{k-1}\right] - \widehat{\mathcal{Q}}_{k,0}\right)\left(\widehat{\mathcal{Q}}_{k,0} - \widehat{\mathcal{Q}}_{k,0}^*\right) \tag{100}$$

$$= \left(\mathcal{Q}_{k,0} - \gamma^m\mathbb{E}\left[\mathcal{Q}_{k,m}|\mathfrak{F}_k, \mathfrak{G}_{k-1}\right] + \gamma^m\mathbb{E}\left[\widehat{\mathcal{Q}}_{k,m}\Big|\mathfrak{F}_k, \mathfrak{G}_{k-1}\right] - \widehat{\mathcal{Q}}_{k,0}\right)\left(\widehat{\mathcal{Q}}_{k,0} - \widehat{\mathcal{Q}}_{k,0}^*\right) \tag{101}$$

$$= \left((\mathcal{Q}_{k,0} - \widehat{\mathcal{Q}}_{k,0}) - \gamma^m\mathbb{E}\left[\mathcal{Q}_{k,m} - \widehat{\mathcal{Q}}_{k,m}\Big|\mathfrak{F}_k, \mathfrak{G}_{k-1}\right]\right)\left((\widehat{\mathcal{Q}}_{k,0} - \mathcal{Q}_{k,0}) + (\mathcal{Q}_{k,0} - \widehat{\mathcal{Q}}_{k,0}^*)\right) \tag{102}$$

$$= -(\mathcal{Q}_{k,0} - \widehat{\mathcal{Q}}_{k,0})^2 + (\mathcal{Q}_{k,0} - \widehat{\mathcal{Q}}_{k,0})(\mathcal{Q}_{k,0} - \widehat{\mathcal{Q}}_{k,0}^*)$$

$$+ \gamma^m\mathbb{E}\left[\widehat{\mathcal{Q}}_{k,m} - \mathcal{Q}_{k,m}\Big|\mathfrak{F}_k, \mathfrak{G}_{k-1}\right](\widehat{\mathcal{Q}}_{k,0} - \mathcal{Q}_{k,0}) + \gamma^m\mathbb{E}\left[\widehat{\mathcal{Q}}_{k,m} - \mathcal{Q}_{k,m}\Big|\mathfrak{F}_k, \mathfrak{G}_{k-1}\right](\mathcal{Q}_{k,0} - \widehat{\mathcal{Q}}_{k,0}^*). \tag{103}$$

Let us now take the expectation of (103) over $\mathfrak{F}_k$ given $\mathfrak{G}_{k-1}$, for each term separately,

- For the first term, we have,

$$\mathbb{E}\left[-(\mathcal{Q}_{k,0} - \widehat{\mathcal{Q}}_{k,0})^2\Big|\mathfrak{G}_{k-1}\right] = -\left\|\mathcal{Q} - \widehat{\mathcal{Q}}_k\right\|_d^2. \tag{104}$$

- For the second term, we have, using the Cauchy-Schwarz inequality,

$$\mathbb{E}\left[(\mathcal{Q}_{k,0} - \widehat{\mathcal{Q}}_{k,0})(\mathcal{Q}_{k,0} - \widehat{\mathcal{Q}}_{k,0}^*)\Big|\mathfrak{G}_{k-1}\right] = \left\|(\mathcal{Q} - \widehat{\mathcal{Q}}_k)(\mathcal{Q} - \widehat{\mathcal{Q}}^*)\right\|_d \tag{105}$$

$$\leq \left\|\mathcal{Q} - \widehat{\mathcal{Q}}_k\right\|_d\left\|\mathcal{Q} - \widehat{\mathcal{Q}}^*\right\|_d. \tag{106}$$

Before proceeding to the third and fourth terms, let us notice that,

$$\mathbb{E}\left[\widehat{\mathcal{Q}}_{k,m} - \mathcal{Q}_{k,m}\Big|\mathfrak{G}_{k-1}\right] = \sum_{s,z,a}d_m(s,z,a)\left(\widehat{\mathcal{Q}}_k(s,z,a) - \mathcal{Q}(s,z,a)\right) \tag{107}$$

$$= \sum_{s,z,a} (d(s,z,a) + d_m(s,z,a) - d(s,z,a)) \left( \widehat{\mathcal{Q}}_k(s,z,a) - \mathcal{Q}(s,z,a) \right). \tag{108}$$

Remembering that $\sup_{s,z,a} \widehat{\mathcal{Q}}_k(s,z,a) \leq B$ and $\sup_{s,z,a} \mathcal{Q}(s,z,a) \leq \frac{1}{1-\gamma}$, we have,

$$\mathbb{E}\left[ \left( \widehat{\mathcal{Q}}_{k,m} - \mathcal{Q}_{k,m} \right)^2 \Big| \mathfrak{G}_{k-1} \right] = \sum_{s,z,a} (d(s,z,a) + d_m(s,z,a) - d(s,z,a)) \left( \widehat{\mathcal{Q}}_k(s,z,a) - \mathcal{Q}(s,z,a) \right)^2 \tag{109}$$

$$= \left\| \widehat{\mathcal{Q}}_k - \mathcal{Q} \right\|_d^2 + \sum_{s,z,a} (d_m(s,z,a) - d(s,z,a)) \left( \widehat{\mathcal{Q}}_k(s,z,a) - \mathcal{Q}(s,z,a) \right)^2 \tag{110}$$

$$\leq \left\| \widehat{\mathcal{Q}}_k - \mathcal{Q} \right\|_d^2 + \|d_m - d\|_{\mathrm{TV}} \sup_{s,z,a} \left( \widehat{\mathcal{Q}}_k(s,z,a) - \mathcal{Q}(s,z,a) \right)^2 \tag{111}$$

$$\leq \left\| \widehat{\mathcal{Q}}_k - \mathcal{Q} \right\|_d^2 + \|d_m - d\|_{\mathrm{TV}} \left( B + \frac{1}{1-\gamma} \right)^2, \tag{112}$$

where $\left( B + \frac{1}{1-\gamma} \right)$ is an upper bound on $\sup_{s,z,a} \left| \widehat{\mathcal{Q}}_k(s,z,a) - \mathcal{Q}(s,z,a) \right|$. Now, using Jensen's inequality and the subadditivity of the square root, we have,

$$\mathbb{E}\left[ \widehat{\mathcal{Q}}_{k,m} - \mathcal{Q}_{k,m} \Big| \mathfrak{G}_{k-1} \right] \leq \mathbb{E}\left[ \sqrt{(\widehat{\mathcal{Q}}_{k,m} - \mathcal{Q}_{k,m})^2} \Big| \mathfrak{G}_{k-1} \right] \tag{113}$$

$$\leq \sqrt{ \mathbb{E}\left[ \left( \widehat{\mathcal{Q}}_{k,m} - \mathcal{Q}_{k,m} \right)^2 \Big| \mathfrak{G}_{k-1} \right] } \tag{114}$$

$$\leq \left\| \widehat{\mathcal{Q}}_k - \mathcal{Q} \right\|_d + \left( B + \frac{1}{1-\gamma} \right) \sqrt{\|d_m - d\|_{\mathrm{TV}}}. \tag{115}$$

With this, we proceed to the third and fourth terms (without the multiplier $\gamma^m$) and show the following.

- For the third term, we have by upper bounding $|\widehat{\mathcal{Q}}_{k,0} - \mathcal{Q}_{k,0}|$ by $B + \frac{1}{1-\gamma}$,

$$\mathbb{E}\left[ (\widehat{\mathcal{Q}}_{k,m} - \mathcal{Q}_{k,m})(\widehat{\mathcal{Q}}_{k,0} - \mathcal{Q}_{k,0}) \Big| \mathfrak{G}_{k-1} \right] \leq \left\| \widehat{\mathcal{Q}}_k - \mathcal{Q} \right\|_d^2 + \left( B + \frac{1}{1-\gamma} \right)^2 \sqrt{\|d_m - d\|_{\mathrm{TV}}}. \tag{116}$$

- For the fourth term, we have by upper bounding $|\mathcal{Q}_{k,0} - \widehat{\mathcal{Q}}_{k,0}^*|$ by $\frac{1}{1-\gamma} + B$,

$$\mathbb{E}\left[ (\widehat{\mathcal{Q}}_{k,m} - \mathcal{Q}_{k,m})(\mathcal{Q}_{k,0} - \widehat{\mathcal{Q}}_{k,0}^*) \Big| \mathfrak{G}_{k-1} \right] \leq \left\| \widehat{\mathcal{Q}}_k - \mathcal{Q} \right\|_d \left\| \mathcal{Q} - \widehat{\mathcal{Q}}^* \right\|_d + \left( B + \frac{1}{1-\gamma} \right)^2 \sqrt{\|d_m - d\|_{\mathrm{TV}}}. \tag{117}$$

By taking expectation over $\mathfrak{G}_{k-1}$ of the four terms and using the previous upper bounds, we obtain,

$$\mathbb{E}\left[ \langle \beta_k - \beta_*, g_k \rangle \right] = \mathbb{E}\left[ \mathbb{E}\left[ \langle \beta_k - \beta_*, g_k \rangle | \mathfrak{G}_{k-1} \right] \right] \tag{118}$$

$$\leq -(1 - \gamma^m) \mathbb{E}\left[ \left\| \widehat{\mathcal{Q}}_k - \mathcal{Q} \right\|_d^2 \right] + (1 + \gamma^m) \mathbb{E}\left[ \left\| \widehat{\mathcal{Q}}_k - \mathcal{Q} \right\|_d \right] \left\| \widehat{\mathcal{Q}}^* - \mathcal{Q} \right\|_d$$

$$+ 2\gamma^m \left( B + \frac{1}{1-\gamma} \right)^2 \sqrt{\|d_m - d\|_{\mathrm{TV}}} \tag{119}$$

$$= -(1 - \gamma^m) \Delta_k^2 + (1 + \gamma^m) \Delta_k \left\| \widehat{\mathcal{Q}}^* - \mathcal{Q} \right\|_d + 2\gamma^m \left( B + \frac{1}{1-\gamma} \right)^2 \sqrt{\|d_m - d\|_{\mathrm{TV}}}. \tag{120}$$

Let us now focus on the second term of equation (95) with $\mathbb{E}\left[ \|g_k\|_2^2 \Big| \mathfrak{G}_{k-1} \right]$. Since $\sup_{s,z,a} \|\phi(s,z,a)\|_2 \leq 1$ and $\|\beta_k\|_2 \leq B$ for all $k \geq 0$, and $r_{k,i} \leq 1$ for all $k \geq 0$ and for all $i < m - 1$, the norm of the gradient (96) is bounded as follows,

$$\sup_{k \geq 0} \|g_k\|_2 \leq \frac{1 - \gamma^m}{1 - \gamma} + (1 + \gamma^m) B \leq \frac{1}{1-\gamma} + 2B. \tag{121}$$

We obtain, for the second term of equation (95),

$$\mathbb{E}\left[\|g_k\|_2^2\right] = \mathbb{E}\left[\mathbb{E}\left[\|g_k\|_2^2\Big|\mathfrak{G}_{k-1}\right]\right] \tag{122}$$

$$\leq \left(\frac{1}{1-\gamma} + 2B\right)^2. \tag{123}$$

By substituting equations (120) and (123) into the Lyapounov drift of equation (95), we obtain,

$$\mathbb{E}\left[\mathcal{L}(\beta_{k+1}) - \mathcal{L}(\beta_k)\right] \leq -2\alpha(1-\gamma^m)\Delta_k^2 + 2\alpha(1+\gamma^m)\Delta_k\left\|\widehat{\mathcal{Q}}^* - \mathcal{Q}\right\|_d + \alpha^2\left(\frac{1}{1-\gamma} + 2B\right)^2$$
$$+ 4\alpha\gamma^m\left(B + \frac{1}{1-\gamma}\right)^2\sqrt{\|d_m - d\|_{\mathrm{TV}}}. \tag{124}$$

By setting $l = \frac{1+\gamma^m}{2(1-\gamma^m)}\min_{f\in\mathcal{F}_\phi^B}\|f - \mathcal{Q}\|_d$, we can write,

$$\mathbb{E}\left[\mathcal{L}(\beta_{k+1}) - \mathcal{L}(\beta_k)\right] \leq -2\alpha(1-\gamma^m)\left(\Delta_k^2 - 2l\Delta_k\right) + \alpha^2\left(\frac{1}{1-\gamma} + 2B\right)^2$$
$$+ 4\alpha\gamma^m\left(B + \frac{1}{1-\gamma}\right)^2\sqrt{\|d_m - d\|_{\mathrm{TV}}} \tag{125}$$

$$= -2\alpha(1-\gamma^m)\left(\Delta_k^2 - 2l\Delta_k + l^2\right) + 2\alpha(1-\gamma^m)l^2 + \alpha^2\left(\frac{1}{1-\gamma} + 2B\right)^2$$
$$+ 4\alpha\gamma^m\left(B + \frac{1}{1-\gamma}\right)^2\sqrt{\|d_m - d\|_{\mathrm{TV}}} \tag{126}$$

$$= -2\alpha(1-\gamma^m)\left(\Delta_k - l\right)^2 + 2\alpha(1-\gamma^m)l^2 + \alpha^2\left(\frac{1}{1-\gamma} + 2B\right)^2$$
$$+ 4\alpha\gamma^m\left(B + \frac{1}{1-\gamma}\right)^2\sqrt{\|d_m - d\|_{\mathrm{TV}}}. \tag{127}$$

By summing all Lyapounov drifts $\sum_{k=0}^{K-1}\mathbb{E}\left[\mathcal{L}(\beta_{k+1}) - \mathcal{L}(\beta_k)\right]$, we get,

$$\mathbb{E}\left[\mathcal{L}(\beta_K) - \mathcal{L}(\beta_0)\right] \leq -2\alpha(1-\gamma^m)\sum_{k=0}^{K-1}\left(\Delta_k - l\right)^2 + 2\alpha K(1-\gamma^m)l^2 + \alpha^2 K\left(\frac{1}{1-\gamma} + 2B\right)^2$$
$$+ 4\alpha K\gamma^m\left(B + \frac{1}{1-\gamma}\right)^2\sqrt{\|d_m - d\|_{\mathrm{TV}}}. \tag{128}$$

By rearranging and dividing by $2\alpha K(1-\gamma^m)$, we obtain after neglecting $\mathcal{L}(\beta_K) > 0$,

$$\frac{1}{K}\sum_{k=0}^{K-1}(\Delta_k - l)^2 \leq \frac{\mathbb{E}\left[\mathcal{L}(\beta_0) - \mathcal{L}(\beta_K)\right]}{2\alpha K(1-\gamma^m)} + l^2 + \frac{\alpha}{2(1-\gamma^m)}\left(\frac{1}{1-\gamma} + 2B\right)^2$$
$$+ \frac{2\gamma^m}{1-\gamma^m}\left(B + \frac{1}{1-\gamma}\right)^2\sqrt{\|d_m - d\|_{\mathrm{TV}}} \tag{129}$$

$$\leq \frac{\|\beta_0 - \beta_*\|_2^2}{2\alpha K(1-\gamma^m)} + l^2 + \frac{\alpha}{2(1-\gamma^m)}\left(\frac{1}{1-\gamma} + 2B\right)^2$$
$$+ \frac{2\gamma^m}{1-\gamma^m}\left(B + \frac{1}{1-\gamma}\right)^2\sqrt{\|d_m - d\|_{\mathrm{TV}}}. \tag{130}$$

The bound obtained through this Lyapounov drift summation can be used to further develop equation (85), using the subadditivity of the square root,

$$\sqrt{\mathbb{E}\left[\left\|\mathcal{Q}-\overline{\mathcal{Q}}\right\|_d^2\right]} \le \sqrt{\frac{1}{K}\sum_{k=0}^{K-1}(\Delta_k - l)^2 + l} \tag{131}$$

$$\le \frac{\|\beta_0 - \beta_*\|_2}{\sqrt{2\alpha K(1-\gamma^m)}} + 2l + \sqrt{\frac{\alpha}{2(1-\gamma^m)}}\left(\frac{1}{1-\gamma} + 2B\right)$$

$$+ \left(B + \frac{1}{1-\gamma}\right)\sqrt{\frac{2\gamma^m}{1-\gamma^m}}\sqrt{\|d_m - d\|_{\text{TV}}} \tag{132}$$

$$= \frac{\|\beta_0 - \beta_*\|_2}{\sqrt{2\alpha K(1-\gamma^m)}} + \frac{1+\gamma^m}{1-\gamma^m}\min_{f\in\mathcal{F}_\phi^B}\|f-\mathcal{Q}\|_d + \sqrt{\frac{\alpha}{2(1-\gamma^m)}}\left(\frac{1}{1-\gamma} + 2B\right)$$

$$+ \left(B + \frac{1}{1-\gamma}\right)\sqrt{\frac{2\gamma^m}{1-\gamma^m}}\sqrt{\|d_m - d\|_{\text{TV}}}. \tag{133}$$

By setting $\alpha = \frac{1}{\sqrt{K}}$ and upper bounding $\|\beta_0 - \beta_*\|$ by $2B$, we get,

$$\sqrt{\mathbb{E}\left[\left\|\mathcal{Q}-\overline{\mathcal{Q}}\right\|_d^2\right]} \le \frac{2B}{\sqrt{2\sqrt{K}(1-\gamma^m)}} + \frac{1+\gamma^m}{1-\gamma^m}\min_{f\in\mathcal{F}_\phi^B}\|f-\mathcal{Q}\|_d + \frac{1}{\sqrt{2\sqrt{K}(1-\gamma^m)}}\left(\frac{1}{1-\gamma} + 2B\right)$$

$$+ \left(B + \frac{1}{1-\gamma}\right)\sqrt{\frac{2\gamma^m}{1-\gamma^m}}\sqrt{\|d_m - d\|_{\text{TV}}} \tag{134}$$

$$= \sqrt{\frac{4B^2 + \left(\frac{1}{1-\gamma} + 2B\right)^2}{2\sqrt{K}(1-\gamma^m)}} + \frac{1+\gamma^m}{1-\gamma^m}\min_{f\in\mathcal{F}_\phi^B}\|f-\mathcal{Q}\|_d$$

$$+ \left(B + \frac{1}{1-\gamma}\right)\sqrt{\frac{2\gamma^m}{1-\gamma^m}}\sqrt{\|d_m - d\|_{\text{TV}}}. \tag{135}$$

This concludes the proof. □

## D. Proof of the Finite-Time Bound for the Symmetric Critic

Let us first find an upper bound on the distance $\left\|Q^\pi - \widetilde{Q}^\pi\right\|_d^2$ between the Q-function $Q^\pi$ and the fixed point $\widetilde{Q}^\pi$.

**Lemma D.1** (Upper bound on the aliasing bias (Cayci et al., 2024)). For any agent-state policy $\pi \in \Pi_\mathcal{M}$, and any $m \in \mathbb{N}$, we have,

$$\left\|Q^\pi - \widetilde{Q}^\pi\right\|_d \le \frac{1-\gamma^m}{1-\gamma}\left\|\mathbb{E}^\pi\left[\sum_{k=0}^\infty \gamma^{km}\left\|\hat{b}_{km} - b_{km}\right\|_{\text{TV}}\bigg| Z_0 = \cdot\right]\right\|_d. \tag{136}$$

*Proof.* The proof is similar to the one of Cayci et al. (2024). Let us first define the expected $m$-step return,

$$\bar{r}_m(s,z,a) = \mathbb{E}^\pi\left[\sum_{k=0}^{m-1}\gamma^k R_k\bigg| S_0 = s, Z_0 = s, A_0 = a\right]. \tag{137}$$

Using the expected $m$-step return and the definition of the belief $b$ in equation (33) and approximate belief $\hat{b}$ in equation (34), it can be noted that,

$$Q^\pi(z,a) = \mathbb{E}^\pi\left[\sum_{k=0}^\infty \gamma^{km}\sum_{s\in\mathcal{S}}b_{km}(s|H_{km})\bar{r}_m(s,Z_{km},A_{km})\bigg| Z_0 = z, A_0 = a\right] \tag{138}$$

$$\widetilde{Q}^{\pi}(z,a) = \mathbb{E}^{\pi}\left[\sum_{k=0}^{\infty} \gamma^{km} \sum_{s \in \mathcal{S}} \hat{b}_{km}(s|Z_{km}) \bar{r}_m(s, Z_{km}, A_{km}) \middle| Z_0 = z, A_0 = a\right]. \tag{139}$$

Indeed, bootstrapping at timestep $m$ based on the agent state only is equivalent to considering the distribution of future states to be $\hat{b}_m(\cdot|Z_m)$ instead of $b_m(\cdot|H_m)$. As a consequence, we have,

$$\left|Q^{\pi}(z,a) - \widetilde{Q}^{\pi}(z,a)\right| = \mathbb{E}^{\pi}\left[\sum_{k=0}^{\infty} \gamma^{km} \sum_{s \in \mathcal{S}} \left(b_{km}(s|H_{km}) - \hat{b}_{km}(s|Z_{km})\right) \bar{r}_m(s, Z_{km}, A_{km}) \middle| Z_0 = z, A_0 = a\right] \tag{140}$$

$$\leq \mathbb{E}^{\pi}\left[\sum_{k=0}^{\infty} \gamma^{km} \sup_{s \in \mathcal{S}}\left|b_{km}(s|H_{km}) - \hat{b}_{km}(s|Z_{km})\right| \sup_{s \in \mathcal{S}}\left|\bar{r}_m(s, Z_{km}, A_{km})\right| \middle| Z_0 = z, A_0 = a\right] \tag{141}$$

$$\leq \mathbb{E}^{\pi}\left[\sum_{k=0}^{\infty} \gamma^{km} \sup_{s \in \mathcal{S}}\left|b_{km}(s|H_{km}) - \hat{b}_{km}(s|Z_{km})\right| \frac{1-\gamma^m}{1-\gamma} \middle| Z_0 = z, A_0 = a\right] \tag{142}$$

$$= \frac{1-\gamma^m}{1-\gamma}\mathbb{E}^{\pi}\left[\sum_{k=0}^{\infty} \gamma^{km} \sup_{s \in \mathcal{S}}\left|b_{km}(s|H_{km}) - \hat{b}_{km}(s|Z_{km})\right| \middle| Z_0 = z, A_0 = a\right] \tag{143}$$

$$\leq \frac{1-\gamma^m}{1-\gamma}\mathbb{E}^{\pi}\left[\sum_{k=0}^{\infty} \gamma^{km} \left\|b_{km}(\cdot|H_{km}) - \hat{b}_{km}(\cdot|Z_{km})\right\|_{\mathrm{TV}} \middle| Z_0 = z, A_0 = a\right] \tag{144}$$

$$\leq \frac{1-\gamma^m}{1-\gamma}\mathbb{E}^{\pi}\left[\sum_{k=0}^{\infty} \gamma^{km} \left\|b_{km} - \hat{b}_{km}\right\|_{\mathrm{TV}} \middle| Z_0 = z, A_0 = a\right], \tag{145}$$

where we use $b_{km}$ and $\hat{b}_{km}$ to denote the random variables $b_{km}(\cdot|H_{km})$ and $\hat{b}_{km}(\cdot|Z_{km})$, respectively. It illustrates that the aliasing bias can be bounded proportionally to the distance between the true belief and the approximate belief at the bootstrapping timesteps. Then, we obtain,

$$\left\|Q^{\pi} - \widetilde{Q}^{\pi}\right\|_d \leq \frac{1-\gamma^m}{1-\gamma}\left\|\mathbb{E}^{\pi}\left[\sum_{k=0}^{\infty} \gamma^{km}\left\|\hat{b}_{km} - b_{km}\right\|_{\mathrm{TV}} \middle| Z_0 = \cdot\right]\right\|_d. \tag{146}$$

This concludes the proof. $\qquad\square$

Using Lemma D.1, we can prove Theorem 4, that is recalled below. Note that some notations used in Appendix C will be reused with another meaning.

**Theorem 4** (Finite-time bound for symmetric $m$-step temporal difference learning (Cayci et al., 2024)). For any agent-state policy $\pi \in \Pi_{\mathcal{M}}$, and any $m \in \mathbb{N}$, we have for Algorithm 1 with $\alpha = \frac{1}{\sqrt{K}}$, and arbitrary $B > 0$,

$$\sqrt{\mathbb{E}\left[\left\|Q^{\pi} - \overline{Q}^{\pi}\right\|_d^2\right]} \leq \varepsilon_{\mathrm{td}} + \varepsilon_{\mathrm{app}} + \varepsilon_{\mathrm{shift}} + \varepsilon_{\mathrm{alias}}, \tag{35}$$

where the temporal difference learning, function approximation, distribution shift, and aliasing terms are given by,

$$\varepsilon_{\mathrm{td}} = \sqrt{\frac{4B^2 + \left(\frac{1}{1-\gamma} + 2B\right)^2}{2\sqrt{K}(1-\gamma^m)}} \tag{36}$$

$$\varepsilon_{\mathrm{app}} = \frac{1+\gamma^m}{1-\gamma^m}\min_{f \in \mathcal{F}_{\chi}^B}\|f - Q^{\pi}\|_d \tag{37}$$

$$\varepsilon_{\mathrm{shift}} = \left(B + \frac{1}{1-\gamma}\right)\sqrt{\frac{2\gamma^m}{1-\gamma^m}}\sqrt{\|d_m - d\|_{\mathrm{TV}}} \tag{38}$$

$$\varepsilon_{\mathrm{alias}} = \frac{2}{1-\gamma}\left\|\mathbb{E}^{\pi}\left[\sum_{k=0}^{\infty} \gamma^{km}\left\|\hat{b}_{km} - b_{km}\right\|_{\mathrm{TV}} \middle| Z_0 = \cdot\right]\right\|_d, \tag{39}$$

with $d(z,a) = \sum_{s \in \mathcal{S}} d^{\pi}(s,z)\pi(a|z)$ the sampling distribution, and $d_m(z,a) = \sum_{s \in \mathcal{S}} d_m^{\pi}(s,z)\pi(a|z)$ the bootstrapping distribution.

*Proof.* To ease notation as for the proof of Theorem 3 in Appendix C, we use $Q$ as a shorthand for $Q^\pi$, $\widehat{Q}^*$ as a shorthand for $\widehat{Q}^\pi_*$, $\widetilde{Q}$ as a shorthand for $\widetilde{Q}^\pi$, $\overline{Q}$ as a shorthand for $\overline{Q}^\pi$ and $\widehat{Q}_k$ as a shorthand for $\widehat{Q}^\pi_{\beta_k}$, where the subscripts and superscripts remain implicit but are assumed clear from context. When evaluating the Q-functions, we go one step further by using $Q_{k,i}$ to denote $Q(Z_{k,i}, A_{k,i})$, $\widehat{Q}^*_{k,i}$ to denote $\widehat{Q}^*(Z_{k,i}, A_{k,i})$, $\widetilde{Q}_{k,i}$ to denote $\widetilde{Q}(Z_{k,i}, A_{k,i})$ and $\widehat{Q}_{k,i}$ to denote $\widehat{Q}_k(Z_{k,i}, A_{k,i})$, and $\chi_{k,i}$ to denote $\chi(Z_{k,i}, A_{k,i})$. In addition, we define $d$ as a shorthand for $d^\pi \otimes \pi$, such that $d(z,a) = d^\pi(z)\pi(a|z)$, and $d_m$ as a shorthand for $d^\pi_m \otimes \pi$, such that $d_m(z,a) = d^\pi_m(z)\pi(a|z)$. Using the triangle inequality and the subadditivity of the square root, we have,

$$\sqrt{\mathbb{E}\left[\left\|Q - \overline{Q}\right\|^2_d\right]} \leq \sqrt{\mathbb{E}\left[\left\|Q - \widetilde{Q}\right\|^2_d\right] + \mathbb{E}\left[\left\|\widetilde{Q} - \overline{Q}\right\|^2_d\right]} \tag{147}$$

$$\leq \sqrt{\mathbb{E}\left[\left\|Q - \widetilde{Q}\right\|^2_d\right]} + \sqrt{\mathbb{E}\left[\left\|\widetilde{Q} - \overline{Q}\right\|^2_d\right]} \tag{148}$$

$$\leq \left\|Q - \widetilde{Q}\right\|_d + \sqrt{\mathbb{E}\left[\left\|\widetilde{Q} - \overline{Q}\right\|^2_d\right]}. \tag{149}$$

We can bound the second term in equation (149) using similar steps as in the proof for the asymmetric finite-time bound (see Appendix C). We obtain,

$$\sqrt{\mathbb{E}\left[\left\|\widetilde{Q} - \overline{Q}\right\|^2_d\right]} \leq \sqrt{\frac{1}{K}\sum_{k=0}^{K-1}(\Delta_k - l)^2 + l}, \tag{150}$$

where $l$ is arbitrary, and $\Delta_k$ is defined as,

$$\Delta_k = \sqrt{\mathbb{E}\left[\left\|\widetilde{Q} - \widehat{Q}_k\right\|^2_d\right]} = \sqrt{\mathbb{E}\left[\left\|\widetilde{Q}(\cdot) - \langle\beta_k, \chi(\cdot)\rangle\right\|^2_d\right]}. \tag{151}$$

Similarly to the asymmetric case (see Appendix C), we consider the Lyapounov function $\mathcal{L}(\beta) = \|\beta_* - \beta\|^2_2$ in order to find a bound on $\frac{1}{K}\sum_{k=0}^{K-1}(\Delta_k - l)^2$. We define $\mathfrak{G}_k = \sigma(Z_{i,j}, A_{i,j}, i \leq k, j \leq m)$ and $\mathfrak{F}_k = \sigma(Z_{k,0}, A_{k,0})$. As in the asymmetric case (see Appendix C), we obtain, using to the law of total expectation,

$$\mathbb{E}\left[\mathcal{L}(\beta_{k+1}) - \mathcal{L}(\beta_k)\right] \leq 2\alpha\mathbb{E}\left[\mathbb{E}\left[\langle\beta_k - \beta_*, g_k\rangle | \mathfrak{G}_{k-1}\right]\right] + \alpha^2\mathbb{E}\left[\mathbb{E}\left[\|g_k\|^2_2 \Big| \mathfrak{G}_{k-1}\right]\right]. \tag{152}$$

Let us focus on the first term of equation (152) with $\mathbb{E}\left[\langle\beta_k - \beta_*, g_k\rangle | \mathfrak{G}_{k-1}\right]$. By conditioning on the sigma-fields $\mathfrak{G}_{k-1}$ and $\mathfrak{F}_k$, we have,

$$\mathbb{E}\left[\langle\beta_k - \beta_*, g_k\rangle | \mathfrak{F}_k, \mathfrak{G}_{k-1}\right] = \left(\mathbb{E}\left[\sum_{t=0}^{m-1}\gamma^t R_{k,t} + \gamma^m\widehat{Q}_{k,m}\Big| \mathfrak{F}_k, \mathfrak{G}_{k-1}\right] - \widehat{Q}_{k,0}\right)\left(\widehat{Q}_{k,0} - \widehat{Q}^*_{k,0}\right). \tag{153}$$

Note that, according to the Bellman operator (7), we have,

$$\mathbb{E}\left[\sum_{t=0}^{m-1}\gamma^t R_{k,t}\Big| \mathfrak{F}_k, \mathfrak{G}_{k-1}\right] = \widetilde{Q}_{k,0} - \gamma^m\mathbb{E}\left[\widetilde{Q}_{k,m}\Big| \mathfrak{F}_k, \mathfrak{G}_{k-1}\right]. \tag{154}$$

It differs from the asymmetric case (see Appendix C) in that we do not necessarily have $Q = \widetilde{Q}$ here. By substituting equation (154) in equation (153), we obtain,

$$\mathbb{E}\left[\langle\beta_k - \beta_*, g_k\rangle | \mathfrak{F}_k, \mathfrak{G}_{k-1}\right]$$
$$= \left(\mathbb{E}\left[\sum_{t=0}^{m-1}\gamma^t R_{k,t}\Big| \mathfrak{F}_k, \mathfrak{G}_{k-1}\right] + \gamma^m\mathbb{E}\left[\widehat{Q}_{k,m}\Big| \mathfrak{F}_k, \mathfrak{G}_{k-1}\right] - \widehat{Q}_{k,0}\right)\left(\widehat{Q}_{k,0} - \widehat{Q}^*_{k,0}\right) \tag{155}$$

$$= \left( \widetilde{Q}_{k,0} - \gamma^m \mathbb{E}\left[ \widetilde{Q}_{k,m} \middle| \mathfrak{F}_k, \mathfrak{G}_{k-1} \right] + \gamma^m \mathbb{E}\left[ \widehat{Q}_{k,m} \middle| \mathfrak{F}_k, \mathfrak{G}_{k-1} \right] - \widehat{Q}_{k,0} \right) \left( \widehat{Q}_{k,0} - \widehat{Q}_{k,0}^* \right) \tag{156}$$

$$= \left( \widetilde{Q}_{k,0} - \gamma^m \mathbb{E}\left[ \widetilde{Q}_{k,m} - \widehat{Q}_{k,m} \middle| \mathfrak{F}_k, \mathfrak{G}_{k-1} \right] - \widehat{Q}_{k,0} \right) \left( \widehat{Q}_{k,0} - \widetilde{Q}_{k,0} + \widetilde{Q}_{k,0} - \widehat{Q}_{k,0}^* \right) \tag{157}$$

$$= \left( (\widetilde{Q}_{k,0} - \widehat{Q}_{k,0}) - \gamma^m \mathbb{E}\left[ \widetilde{Q}_{k,m} - \widehat{Q}_{k,m} \middle| \mathfrak{F}_k, \mathfrak{G}_{k-1} \right] \right) \left( (\widehat{Q}_{k,0} - \widetilde{Q}_{k,0}) + (\widetilde{Q}_{k,0} - \widehat{Q}_{k,0}^*) \right) \tag{158}$$

$$= -(\widetilde{Q}_{k,0} - \widehat{Q}_{k,0})^2 + (\widetilde{Q}_{k,0} - \widehat{Q}_{k,0})(\widetilde{Q}_{k,0} - \widehat{Q}_{k,0}^*)$$
$$+ \gamma^m \mathbb{E}\left[ \widehat{Q}_{k,m} - \widetilde{Q}_{k,m} \middle| \mathfrak{F}_k, \mathfrak{G}_{k-1} \right] (\widehat{Q}_{k,0} - \widetilde{Q}_{k,0}) + \gamma^m \mathbb{E}\left[ \widehat{Q}_{k,m} - \widetilde{Q}_{k,m} \middle| \mathfrak{F}_k, \mathfrak{G}_{k-1} \right] (\widetilde{Q}_{k,0} - \widehat{Q}_{k,0}^*). \tag{159}$$

We now follow the same technique as in the asymmetric case (see Appendix C) for each of the four terms. By taking the expectation over $\mathfrak{F}_k$, we get the following.

- For the first term, we have,

$$\mathbb{E}\left[ -(\widetilde{Q}_{k,0} - \widehat{Q}_{k,0})^2 \middle| \mathfrak{G}_{k-1} \right] = -\left\| \widetilde{Q} - \widehat{Q}_k \right\|_d^2. \tag{160}$$

- For the second term, we have,

$$\mathbb{E}\left[ (\widetilde{Q}_{k,0} - \widehat{Q}_{k,0})(\widetilde{Q}_{k,0} - \widehat{Q}_{k,0}^*) \middle| \mathfrak{G}_{k-1} \right] \le \left\| \widetilde{Q} - \widehat{Q}_k \right\|_d \left\| \widetilde{Q} - \widehat{Q}^* \right\|_d. \tag{161}$$

- For the third term, we have,

$$\mathbb{E}\left[ (\widehat{Q}_{k,m} - \widetilde{Q}_{k,m})(\widehat{Q}_{k,0} - \widetilde{Q}_{k,0}) \middle| \mathfrak{G}_{k-1} \right] \le \left\| \widehat{Q}_k - \widetilde{Q} \right\|_d^2 + \left( B + \frac{1}{1-\gamma} \right)^2 \sqrt{\| d_m - d \|_{\mathrm{TV}}}. \tag{162}$$

- For the fourth term, we have,

$$\mathbb{E}\left[ (\widehat{Q}_{k,m} - \widetilde{Q}_{k,m})(\widetilde{Q}_{k,0} - \widehat{Q}_{k,0}^*) \middle| \mathfrak{G}_{k-1} \right] \le \left\| \widehat{Q}_k - \widetilde{Q} \right\|_d \left\| \widetilde{Q} - \widehat{Q}^* \right\|_d + \left( B + \frac{1}{1-\gamma} \right)^2 \sqrt{\| d_m - d \|_{\mathrm{TV}}}. \tag{163}$$

By taking expectation over $\mathfrak{G}_{k-1}$ of the four terms and using the previous upper bounds, we obtain,

$$\mathbb{E}[\langle \beta_k - \beta_*, g_k \rangle] \le -(1 - \gamma^m)\Delta_k^2 + (1 + \gamma^m)\Delta_k \left\| \widehat{Q}^* - \widetilde{Q} \right\|_d + 2\gamma^m \left( B + \frac{1}{1-\gamma} \right)^2 \sqrt{\| d_m - d \|_{\mathrm{TV}}}. \tag{164}$$

The second term in equation (152) is treated similarly to the asymmetric case (see Appendix C), which yields,

$$\mathbb{E}\left[ \| g_k \|_2^2 \right] \le \left( \frac{1}{1-\gamma} + 2B \right)^2. \tag{165}$$

By substituting equations (164) and (165) into the Lyapounov drift of equation (152), we obtain,

$$\mathbb{E}[\mathcal{L}(\beta_{k+1}) - \mathcal{L}(\beta_k)] \le -2\alpha(1 - \gamma^m)\Delta_k^2 + 2\alpha(1 + \gamma^m)\Delta_k \left\| \widehat{Q}^* - \widetilde{Q} \right\|_d + \alpha^2 \left( \frac{1}{1-\gamma} + 2B \right)^2$$
$$+ 4\alpha\gamma^m \left( B + \frac{1}{1-\gamma} \right)^2 \sqrt{\| d_m - d \|_{\mathrm{TV}}}. \tag{166}$$

We can upper bound $\left\| \widehat{Q}^* - \widetilde{Q} \right\|_d$ as follows,

$$\left\| \widehat{Q}^* - \widetilde{Q} \right\|_d \le \left\| \widehat{Q}^* - Q \right\|_d + \left\| Q - \widetilde{Q} \right\|_d. \tag{167}$$

By setting $l = \frac{1+\gamma^m}{2(1-\gamma^m)} \left( \left\| \widehat{Q}^* - Q \right\|_d + \left\| Q - \widetilde{Q} \right\|_d \right)$, we can write, following a similar strategy as in the asymmetric case (see Appendix C),

$$\mathbb{E}\left[\mathcal{L}(\beta_{k+1}) - \mathcal{L}(\beta_k)\right] \leq -2\alpha(1-\gamma^m)(\Delta_k - l)^2 + 2\alpha(1-\gamma^m)l^2 + \alpha^2 \left( \frac{1}{1-\gamma} + 2B \right)^2$$

$$+ 4\alpha\gamma^m \left( B + \frac{1}{1-\gamma} \right)^2 \sqrt{\|d_m - d\|_{\text{TV}}}. \tag{168}$$

By summing all drifts, rearranging, and dividing by $2\alpha K(1-\gamma^m)$, we obtain after neglecting $\mathcal{L}(\beta_K) > 0$,

$$\frac{1}{K}\sum_{k=0}^{K-1}(\Delta_k - l)^2 \leq \frac{\|\beta_0 - \beta_*\|_2^2}{2\alpha K(1-\gamma^m)} + l^2 + \frac{\alpha}{2(1-\gamma^m)}\left(\frac{1}{1-\gamma} + 2B\right)^2$$

$$+ \frac{2\gamma^m}{1-\gamma^m}\left( B + \frac{1}{1-\gamma} \right)^2 \sqrt{\|d_m - d\|_{\text{TV}}}. \tag{169}$$

The bound obtained through this Lyapounov drift summation can be used to further develop equation (150), using the subadditivity of the square root,

$$\sqrt{\mathbb{E}\left[\left\| \widetilde{Q} - \overline{Q} \right\|_d^2\right]} \leq \sqrt{\frac{1}{K}\sum_{k=0}^{K-1}(\Delta_k - l)^2} + l \tag{170}$$

$$\leq \frac{\|\beta_0 - \beta_*\|_2}{\sqrt{2\alpha K(1-\gamma^m)}} + 2l + \sqrt{\frac{\alpha}{2(1-\gamma^m)}}\left(\frac{1}{1-\gamma} + 2B\right)$$

$$+ \left( B + \frac{1}{1-\gamma} \right)\sqrt{\frac{2\gamma^m}{1-\gamma^m}}\sqrt{\|d_m - d\|_{\text{TV}}} \tag{171}$$

$$= \frac{\|\beta_0 - \beta_*\|_2}{\sqrt{2\alpha K(1-\gamma^m)}} + 2l + \sqrt{\frac{\alpha}{2(1-\gamma^m)}}\left(\frac{1}{1-\gamma} + 2B\right)$$

$$+ \left( B + \frac{1}{1-\gamma} \right)\sqrt{\frac{2\gamma^m}{1-\gamma^m}}\sqrt{\|d_m - d\|_{\text{TV}}}. \tag{172}$$

Plugging equation (172) into equation (149), and substituting back $l$, we finally have,

$$\sqrt{\mathbb{E}\left[\left\| Q - \overline{Q} \right\|_d^2\right]} \leq \frac{\|\beta_0 - \beta_*\|_2}{\sqrt{2\alpha K(1-\gamma^m)}} + \frac{1+\gamma^m}{1-\gamma^m}\left( \left\| \widehat{Q}^* - Q \right\|_d + \left\| Q - \widetilde{Q} \right\|_d \right) + \sqrt{\frac{\alpha}{2(1-\gamma^m)}}\left(\frac{1}{1-\gamma} + 2B\right)$$

$$+ \left( B + \frac{1}{1-\gamma} \right)\sqrt{\frac{2\gamma^m}{1-\gamma^m}}\sqrt{\|d_m - d\|_{\text{TV}}} + \left\| Q - \widetilde{Q} \right\|_d \tag{173}$$

$$\leq \frac{\|\beta_0 - \beta_*\|_2}{\sqrt{2\alpha K(1-\gamma^m)}} + \frac{1+\gamma^m}{1-\gamma^m}\left\| \widehat{Q}^* - Q \right\|_d + \sqrt{\frac{\alpha}{2(1-\gamma^m)}}\left(\frac{1}{1-\gamma} + 2B\right)$$

$$+ \left( B + \frac{1}{1-\gamma} \right)\sqrt{\frac{2\gamma^m}{1-\gamma^m}}\sqrt{\|d_m - d\|_{\text{TV}}} + \frac{2}{1-\gamma^m}\left\| Q - \widetilde{Q} \right\|_d \tag{174}$$

Using Lemma D.1, we finally obtain,

$$\sqrt{\mathbb{E}\left[\left\| Q - \overline{Q} \right\|_d^2\right]} \leq \frac{\|\beta_0 - \beta_*\|_2}{\sqrt{2\alpha K(1-\gamma^m)}} + \frac{1+\gamma^m}{1-\gamma^m}\left\| \widehat{Q}^* - Q \right\|_d + \sqrt{\frac{\alpha}{2(1-\gamma^m)}}\left(\frac{1}{1-\gamma} + 2B\right)$$

$$+ \left( B + \frac{1}{1-\gamma} \right)\sqrt{\frac{2\gamma^m}{1-\gamma^m}}\sqrt{\|d_m - d\|_{\text{TV}}}$$

$$+ \left( \frac{2}{1-\gamma^m} \right)\frac{1-\gamma^m}{1-\gamma}\left\| \mathbb{E}\left[ \sum_{k=0}^{\infty} \gamma^{km} \left\| \hat{b}_{km} - b_{km} \right\|_{\text{TV}} \middle| Z_0 = \cdot \right] \right\|_d \tag{175}$$

$$\leq \frac{\|\beta_0 - \beta_*\|_2}{\sqrt{2\alpha K(1-\gamma^m)}} + \frac{1+\gamma^m}{1-\gamma^m} \min_{f \in \mathcal{F}_\phi^B} \|f - \mathcal{Q}\|_d + \sqrt{\frac{\alpha}{2(1-\gamma^m)}} \left(\frac{1}{1-\gamma} + 2B\right)$$

$$+ \left(B + \frac{1}{1-\gamma}\right) \sqrt{\frac{2\gamma^m}{1-\gamma^m}} \sqrt{\|d_m - d\|_{\mathrm{TV}}}$$

$$+ \frac{2}{1-\gamma} \left\| \mathbb{E}\left[\sum_{k=0}^{\infty} \gamma^{km} \left\|\hat{b}_{km} - b_{km}\right\|_{\mathrm{TV}} \Big| Z_0 = \cdot\right] \right\|_d. \tag{176}$$

By setting $\alpha = \frac{1}{\sqrt{K}}$ and upper bounding $\|\beta_0 - \beta_*\|$ by $2B$, we get,

$$\sqrt{\mathbb{E}\left[\|Q - \overline{Q}\|_d^2\right]} \leq \sqrt{\frac{4B^2 + \left(\frac{1}{1-\gamma} + 2B\right)^2}{2\sqrt{K}(1-\gamma^m)}} + \frac{1+\gamma^m}{1-\gamma^m} \min_{f \in \mathcal{F}_\phi^B} \|f - \mathcal{Q}\|_d$$

$$+ \left(B + \frac{1}{1-\gamma}\right) \sqrt{\frac{2\gamma^m}{1-\gamma^m}} \sqrt{\|d_m - d\|_{\mathrm{TV}}}$$

$$+ \frac{2}{1-\gamma} \left\| \mathbb{E}\left[\sum_{k=0}^{\infty} \gamma^{km} \left\|\hat{b}_{km} - b_{km}\right\|_{\mathrm{TV}} \Big| Z_0 = \cdot\right] \right\|_d. \tag{177}$$

This concludes the proof. $\qquad\square$

## E. Proof of the Finite-Time Bound for the Natural Actor-Critic

Let us first give the performance difference lemma for POMDP proved by Cayci et al. (2024). Note that this proof is completely agnostic about the critic used to compute $\pi_1, \pi_2 \in \Pi_\mathcal{M}$ and is thus applicable both to the asymmetric setting and the symmetric setting.

**Lemma E.1** (Performance difference (Cayci et al., 2024)). For any two agent-state polices $\pi_1, \pi_2 \in \Pi_\mathcal{M}$,

$$V^{\pi_2}(z_0) - V^{\pi_1}(z_0) \leq \frac{1}{1-\gamma}\mathbb{E}^{d^{\pi_2}}\left[A^{\pi_1}(Z, A)|Z_0 = z_0\right] + \frac{2}{1-\gamma}\varepsilon_{\inf}^{\pi_2}(z_0), \tag{178}$$

where,

$$\varepsilon_{\inf}^{\pi_2}(z_0) = \mathbb{E}^{\pi_2}\left[\sum_{k=0}^{\infty} \gamma^k \left\|\hat{b}_k - b_k\right\|_{\mathrm{TV}} \Big| Z_0 = z_0\right]. \tag{179}$$

*Proof.* The proof is similar to the one of Cayci et al. (2024). First, let us decompose the performance difference in the following terms,

$$V^{\pi_2}(z_0) - V^{\pi_1}(z_0) = \mathbb{E}^{\pi_2}\left[\sum_{t=0}^{\infty} \gamma^t R_t \Big| Z_0 = z_0\right] - V^{\pi_1}(z_0) \tag{180}$$

$$= \mathbb{E}^{\pi_2}\left[\sum_{t=0}^{\infty} \gamma^t \left(R_t - V^{\pi_1}(Z_t) + V^{\pi_1}(Z_t)\right) \Big| Z_0 = z_0\right] - V^{\pi_1}(z_0) \tag{181}$$

$$= \mathbb{E}^{\pi_2}\left[\sum_{t=0}^{\infty} \gamma^t \left(R_t - V^{\pi_1}(Z_t) + \gamma V^{\pi_1}(Z_{t+1})\right) \Big| Z_0 = z_0\right] \tag{182}$$

$$= \mathbb{E}^{\pi_2}\left[\sum_{t=0}^{\infty} \gamma^t \left(R_t + \gamma \mathcal{V}^{\pi_1}(S_{t+1}, Z_{t+1}) - V^{\pi_1}(Z_t)\right) \Big| Z_0 = z_0\right]$$

$$+ \mathbb{E}^{\pi_2}\left[\sum_{t=0}^{\infty} \gamma^t \left(\gamma V^{\pi_1}(Z_{t+1}) - \gamma \mathcal{V}^{\pi_1}(S_{t+1}, Z_{t+1})\right) \Big| Z_0 = z_0\right] \tag{183}$$

$$= \mathbb{E}^{\pi_2} \left[ \sum_{t=0}^{\infty} \gamma^t \left( R_t + \gamma \mathcal{V}^{\pi_1}(S_{t+1}, Z_{t+1}) - V^{\pi_1}(Z_t) \right) \Big| Z_0 = z_0 \right]$$

$$+ \mathbb{E}^{\pi_2} \left[ \sum_{t=0}^{\infty} \gamma^{t+1} \left( V^{\pi_1}(Z_{t+1}) - \mathcal{V}^{\pi_1}(S_{t+1}, Z_{t+1}) \right) \Big| Z_0 = z_0 \right]. \tag{184}$$

Let us focus on bounding the first term in equation (184). We have, for any $T > 0$,

$$\left| \sum_{t=0}^{T} \gamma^t \left( R_t + \gamma \mathcal{V}^{\pi_1}(S_{t+1}, Z_{t+1}) - V^{\pi_1}(Z_t) \right) \right| \leq \frac{2}{(1-\gamma)^2} < \infty. \tag{185}$$

By Lebesgue's dominated convergence, we have,

$$\mathbb{E}^{\pi_2} \left[ \sum_{t=0}^{\infty} \gamma^t \left( R_t + \gamma \mathcal{V}^{\pi_1}(S_{t+1}, Z_{t+1}) - V^{\pi_1}(Z_t) \right) \Big| Z_0 = z_0 \right]$$

$$= \sum_{t=0}^{\infty} \gamma^t \mathbb{E}^{\pi_2} \left[ R_t + \gamma \mathcal{V}^{\pi_1}(S_{t+1}, Z_{t+1}) - V^{\pi_1}(Z_t) | Z_0 = z_0 \right]. \tag{186}$$

Then, by the law of total expectation, we have at any timestep $t \geq 0$,

$$\mathbb{E}^{\pi_2} \left[ R_t + \gamma \mathcal{V}^{\pi_1}(S_{t+1}, Z_{t+1}) - V^{\pi_1}(Z_t) | Z_0 = z_0 \right]$$

$$= \mathbb{E} \left[ \mathbb{E}^{\pi_2} \left[ R_t + \gamma \mathcal{V}^{\pi_1}(S_{t+1}, Z_{t+1}) | H_t, Z_t \right] - V^{\pi_1}(Z_t) \Big| Z_0 = z_0 \right]. \tag{187}$$

And, we have,

$$\mathbb{E}^{\pi_2} \left[ R_t + \gamma \mathcal{V}^{\pi_1}(S_{t+1}, Z_{t+1}) | H_t = h_t, Z_t = z_t \right]$$

$$= \sum_{s_t, a_t} b_t(s_t | h_t) \pi_2(a_t | z_t) \mathcal{Q}^{\pi_1}(s_t, z_t, a_t) \tag{188}$$

$$= \sum_{a_t} \pi_2(a_t | z_t) Q^{\pi_1}(z_t, a_t) + \sum_{s_t, a_t} b_t(s_t | h_t) \pi_2(a_t | z_t) \mathcal{Q}^{\pi_1}(s_t, z_t, a_t) - \sum_{a_t} \pi_2(a_t | z_t) Q^{\pi_1}(z_t, a_t) \tag{189}$$

$$= \sum_{a_t} \pi_2(a_t | z_t) Q^{\pi_1}(z_t, a_t) + \sum_{s_t, a_t} b_t(s_t | h_t) \pi_2(a_t | z_t) \mathcal{Q}^{\pi_1}(s_t, z_t, a_t)$$

$$- \sum_{s_t, a_t} \hat{b}_t(s_t | z_t) \pi_2(a_t | z_t) \mathcal{Q}^{\pi_1}(s_t, z_t, a_t) \tag{190}$$

$$= \sum_{a_t} \pi_2(a_t | z_t) Q^{\pi_1}(z_t, a_t) + \sum_{s_t, a_t} \left( b_t(s_t | h_t) - \hat{b}_t(s_t | z_t) \right) \pi_2(a_t | z_t) \mathcal{Q}^{\pi_1}(s_t, z_t, a_t). \tag{191}$$

By noting that $\sup_{s,z} |\sum_a \pi_2(a|z) \mathcal{Q}^{\pi_1}(s, z, a)| \leq \sup_{s,z,a} |\mathcal{Q}^{\pi_1}(s, z, a)| \leq \frac{1}{1-\gamma}$, we obtain,

$$\mathbb{E}^{\pi_2} \left[ R_t + \gamma \mathcal{V}^{\pi_1}(S_{t+1}, Z_{t+1}) | H_t = h_t, Z_t = z_t \right]$$

$$\leq \sum_{a_t} \pi_2(a_t | z_t) Q^{\pi_1}(z_t, a_t) + \frac{1}{1-\gamma} \left\| b_t(\cdot | h_t) - \hat{b}_t(\cdot | z_t) \right\|_{\mathrm{TV}}. \tag{192}$$

Finally, the expectation at time $t \geq 0$ can be written as,

$$\mathbb{E}^{\pi_2} \left[ R_t + \gamma \mathcal{V}^{\pi_1}(S_{t+1}, Z_{t+1}) - V^{\pi_1}(Z_t) | Z_0 = z_0 \right]$$

$$= \mathbb{E} \left[ \mathbb{E}^{\pi_2} \left[ R_t + \gamma \mathcal{V}^{\pi_1}(S_{t+1}, Z_{t+1}) | H_t, Z_t \right] - V^{\pi_1}(Z_t) \Big| Z_0 = z_0 \right] \tag{193}$$

$$\leq \mathbb{E}^{\pi_2} \left[ Q^{\pi_1}(Z_t, A_t) + \frac{1}{1-\gamma} \left\| b_t(\cdot | H_t) - \hat{b}_t(\cdot | Z_t) \right\|_{\mathrm{TV}} - V^{\pi_1}(Z_t) \Big| Z_0 = z_0 \right] \tag{194}$$

$$= \mathbb{E}^{\pi_2} \left[ A^{\pi_1}(Z_t, A_t) - \frac{1}{1-\gamma} \left\| b_t(\cdot | H_t) - \hat{b}_t(\cdot | Z_t) \right\|_{\mathrm{TV}} \Big| Z_0 = z_0 \right] \tag{195}$$

Now, by using Lebesgue's dominated theorem in the reverse direction, we have,

$$\mathbb{E}^{\pi_2}\left[\sum_{t=0}^{\infty}\gamma^t\left(R_t+\gamma\mathcal{V}^{\pi_1}(S_{t+1},Z_{t+1})-V^{\pi_1}(Z_t)\right)\bigg|Z_0=z_0\right]$$

$$\leq\mathbb{E}^{\pi_2}\left[\sum_{t=0}^{\infty}\gamma^t A^{\pi_1}(Z_t,A_t)\bigg|Z_0=z_0\right]+\frac{1}{1-\gamma}\mathbb{E}^{\pi_2}\left[\sum_{t=0}^{\infty}\gamma^t\left\|\hat{b}_t-b_t\right\|_{\mathrm{TV}}\bigg|Z_0=z_0\right] \tag{196}$$

$$=\mathbb{E}^{\pi_2}\left[\sum_{t=0}^{\infty}\gamma^t A^{\pi_1}(Z_t,A_t)\bigg|Z_0=z_0\right]+\frac{1}{1-\gamma}\varepsilon_{\inf}^{\pi_2}(z_0) \tag{197}$$

Now, let us focus on bounding the second term in equation (184). We have, for any $T>0$,

$$\left|\sum_{t=0}^{T}\gamma^{t+1}\left(V^{\pi_1}(Z_{t+1})-\mathcal{V}^{\pi_1}(S_{t+1},Z_{t+1})\right)\right|\leq\frac{2}{(1-\gamma)^2}<\infty. \tag{198}$$

Using Lebesgue dominated convergence theorem, we can write,

$$\mathbb{E}^{\pi_2}\left[\sum_{t=0}^{\infty}\gamma^{t+1}\left(V^{\pi_1}(Z_{t+1})-\mathcal{V}^{\pi_1}(S_{t+1},Z_{t+1})\right)\bigg|Z_0=z_0\right]$$

$$=\sum_{t=0}^{\infty}\gamma^{t+1}\mathbb{E}^{\pi_2}\left[V^{\pi_1}(Z_{t+1})-\mathcal{V}^{\pi_1}(S_{t+1},Z_{t+1})|Z_0=z_0\right]. \tag{199}$$

By the law of total expectation, we have at any timestep $t\geq 0$,

$$\mathbb{E}^{\pi_2}\left[V^{\pi_1}(Z_{t+1})-\mathcal{V}^{\pi_1}(S_{t+1},Z_{t+1})|Z_0=z_0\right]$$

$$=\mathbb{E}\left[V^{\pi_1}(Z_{t+1})-\mathbb{E}^{\pi_2}\left[\mathcal{V}^{\pi_1}(S_{t+1},Z_{t+1})|H_{t+1},Z_{t+1}\right]\big|Z_0=z_0\right]. \tag{200}$$

And, we have,

$$\mathbb{E}^{\pi_2}\left[\mathcal{V}^{\pi_1}(S_{t+1},z_{t+1})|H_{t+1}=h_{t+1},Z_{t+1}=z_{t+1},\right]$$

$$=\sum_{s_{t+1}}b_{t+1}(s_{t+1}|h_{t+1})\mathcal{V}^{\pi_1}(s_{t+1},z_{t+1}) \tag{201}$$

$$=V^{\pi_1}(z_{t+1})+\sum_{s_{t+1}}b_{t+1}(s_{t+1}|h_{t+1})\mathcal{V}^{\pi_1}(s_{t+1},z_{t+1})-V^{\pi_1}(z_{t+1}) \tag{202}$$

$$=V^{\pi_1}(z_{t+1})+\sum_{s_{t+1}}b_{t+1}(s_{t+1}|h_{t+1})\mathcal{V}^{\pi_1}(s_{t+1},z_{t+1})-\sum_{s_{t+1}}\hat{b}_{t+1}(s_{t+1}|z_{t+1})\mathcal{V}^{\pi_1}(s_{t+1},z_{t+1}) \tag{203}$$

$$=V^{\pi_1}(z_{t+1})+\sum_{s_{t+1}}\left(b_{t+1}(s_{t+1}|h_{t+1})-\hat{b}_{t+1}(s_{t+1}|z_{t+1})\right)\mathcal{V}^{\pi_1}(s_{t+1},z_{t+1}). \tag{204}$$

From there, by noting that $\sup_{s,z}|\mathcal{V}^{\pi_1}(s,z)|\leq\frac{1}{1-\gamma}$, we obtain,

$$\mathbb{E}^{\pi_2}\left[\mathcal{V}^{\pi_1}(S_{t+1},z_{t+1})|H_{t+1}=h_{t+1},Z_{t+1}=z_{t+1},\right]$$

$$\geq V^{\pi_1}(z_{t+1})-\frac{1}{1-\gamma}\left\|b_{t+1}(\cdot|h_{t+1})-\hat{b}_{t+1}(\cdot|z_{t+1})\right\|_{\mathrm{TV}}. \tag{205}$$

Finally, the expectation at time $t\geq 0$ can be written as,

$$\mathbb{E}^{\pi_2}\left[V^{\pi_1}(Z_{t+1})-\mathcal{V}^{\pi_1}(S_{t+1},Z_{t+1})|Z_0=z_0\right]$$

$$=\mathbb{E}\left[V^{\pi_1}(Z_{t+1})-\mathbb{E}^{\pi_2}\left[\mathcal{V}^{\pi_1}(S_{t+1},Z_{t+1})|H_{t+1},Z_{t+1}\right]\big|Z_0=z_0\right] \tag{206}$$

$$\leq\mathbb{E}\left[V^{\pi_1}(Z_{t+1})-V^{\pi_1}(Z_{t+1})+\frac{1}{1-\gamma}\left\|b_{t+1}(\cdot|H_{t+1})-\hat{b}_{t+1}(\cdot|Z_{t+1})\right\|_{\mathrm{TV}}\bigg|Z_0=z_0\right] \tag{207}$$

$$\leq \mathbb{E}\left[\frac{1}{1-\gamma}\left\|b_{t+1}(\cdot|H_{t+1}) - \hat{b}_{t+1}(\cdot|Z_{t+1})\right\|_{\text{TV}}\middle|Z_0 = z_0\right]. \tag{208}$$

Now, by using Lebesgue's dominated theorem in the reverse direction, we have,

$$\mathbb{E}^{\pi_2}\left[\sum_{t=0}^{\infty}\gamma^{t+1}\left(V^{\pi_1}(Z_{t+1}) - \mathcal{V}^{\pi_1}(S_{t+1}, Z_{t+1})\right)\middle|Z_0 = z_0\right]$$

$$\leq \frac{1}{1-\gamma}\mathbb{E}^{\pi_2}\left[\sum_{t=0}^{\infty}\gamma^{t+1}\left\|b_{t+1}(\cdot|H_{t+1}) - \hat{b}_{t+1}(\cdot|Z_{t+1})\right\|_{\text{TV}}\middle|Z_0 = z_0\right] \tag{209}$$

$$= \frac{1}{1-\gamma}\mathbb{E}^{\pi_2}\left[\sum_{t=0}^{\infty}\gamma^{t}\left\|b_t(\cdot|H_t) - \hat{b}_t(\cdot|Z_t)\right\|_{\text{TV}} - \left\|b_0(\cdot|H_0) - \hat{b}_0(\cdot|Z_0)\right\|_{\text{TV}}\middle|Z_0 = z_0\right] \tag{210}$$

$$= \frac{1}{1-\gamma}\mathbb{E}^{\pi_2}\left[\sum_{t=0}^{\infty}\gamma^{t}\left\|b_t(\cdot|H_t) - \hat{b}_t(\cdot|Z_t)\right\|_{\text{TV}}\middle|Z_0 = z_0\right]$$

$$\qquad - \mathbb{E}^{\pi_2}\left[\left\|b_0(\cdot|H_0) - \hat{b}_0(\cdot|Z_0)\right\|_{\text{TV}}\middle|Z_0 = z_0\right] \tag{211}$$

$$= \frac{1}{1-\gamma}\varepsilon_{\text{inf}}^{\pi_2}(z_0) - \mathbb{E}^{\pi_2}\left[\left\|b_0(\cdot|H_0) - \hat{b}_0(\cdot|Z_0)\right\|_{\text{TV}}\middle|Z_0 = z_0\right] \tag{212}$$

$$\leq \frac{1}{1-\gamma}\varepsilon_{\text{inf}}^{\pi_2}(z_0). \tag{213}$$

Finally, by substituting the upper bound (197) on the first term and the upper bound (213) on the second term into equation (184), we obtain,

$$V^{\pi_2}(z_0) - V^{\pi_1}(z_0) \leq \mathbb{E}^{\pi_2}\left[\sum_{t=0}^{\infty}\gamma^{t}A^{\pi_1}(Z_t, A_t)\middle|Z_0 = z_0\right] + \frac{2}{1-\gamma}\varepsilon_{\text{inf}}^{\pi_2}(z_0) \tag{214}$$

$$= \frac{1}{1-\gamma}\mathbb{E}^{d^{\pi_2}}\left[A^{\pi_1}(Z, A)|Z_0 = z_0\right] + \frac{2}{1-\gamma}\varepsilon_{\text{inf}}^{\pi_2}(z_0). \tag{215}$$

This concludes the proof. $\qquad\square$

Using Lemma E.1, we can prove Theorem 5, that is recalled below. The proof from Cayci et al. (2024) is generalized to the asymmetric setting.

**Theorem 5** (Finite-time bound for asymmetric and symmetric natural actor-critic algorithm). For any agent-state process $\mathcal{M} = (\mathcal{Z}, U)$, we have for Algorithm 2 with $\alpha = \frac{1}{\sqrt{K}}$, $\zeta = \frac{B\sqrt{1-\gamma}}{\sqrt{2N}}$, $\eta = \frac{1}{\sqrt{T}}$ and arbitrary $B > 0$,

$$(1-\gamma)\min_{0\leq t<T}\mathbb{E}\left[J(\pi^*) - J(\pi_t)\right] \leq \varepsilon_{\text{nac}} + 2\varepsilon_{\text{inf}} + \overline{C}_\infty\left(\varepsilon_{\text{actor}} + 2\varepsilon_{\text{grad}} + 2\sqrt{6}\frac{1}{T}\sum_{t=0}^{T-1}\varepsilon_{\text{critic}}^{\pi_t}\right), \tag{41}$$

where the different terms may differ for asymmetric and symmetric critics,

$$\varepsilon_{\text{nac}} = \frac{B^2 + 2\log|\mathcal{A}|}{2\sqrt{T}} \tag{42}$$

$$\varepsilon_{\text{actor}} = \sqrt{\frac{(2-\gamma)B}{(1-\gamma)\sqrt{N}}} \tag{43}$$

$$\varepsilon_{\text{inf,asym}} = 0 \tag{44}$$

$$\varepsilon_{\text{inf,sym}} = \mathbb{E}^{\pi^*}\left[\sum_{k=0}^{\infty}\gamma^{k}\left\|\hat{b}_k - b_k\right\|_{\text{TV}}\right] \tag{45}$$

$$\varepsilon_{\text{grad,asym}} = \sup_{0\leq t<T}\sqrt{\min_{w}\mathcal{L}_t(w)} \tag{46}$$

$$\varepsilon_{\text{grad,sym}} = \sup_{0 \le t < T} \sqrt{\min_w L_t(w)}, \tag{47}$$

and $\varepsilon_{\text{critic}}^{\pi_t}$ is given in Theorem 3 and Theorem 4.

*Proof.* The proof is based on a Lyapounov drift result using the following Lyapounov function,

$$\Lambda(\pi) = \sum_{z \in \mathcal{Z}} d^{\pi^*}(z) \text{KL}(\pi^*(\cdot|z) \| \pi(\cdot|z)). \tag{216}$$

The Lyapounov drift is given by,

$$\Lambda(\pi_{t+1}) - \Lambda(\pi_t) = \sum_{z \in \mathcal{Z}} d^{\pi^*}(z) \sum_{a \in \mathcal{A}} \pi^*(a|z) \log \frac{\pi_t(a|z)}{\pi_{t+1}(a|z)} \tag{217}$$

$$= \sum_{z,a} d^{\pi^*}(z,a) \log \frac{\pi_t(a|z)}{\pi_{t+1}(a|z)}. \tag{218}$$

Since $\sup_{z,a} \|\psi(z,a)\|_2 \le 1$, we have that $\log \pi_\theta(a|z)$ is 1-smooth (Agarwal et al., 2021), which implies,

$$\log \pi_{\theta_2}(a|z) \le \log \pi_{\theta_1}(a|z) + \langle \nabla_\theta \log \pi_{\theta_1}(a|z), \theta_2 - \theta_1 \rangle + \frac{1}{2} \|\theta_2 - \theta_1\|_2^2. \tag{219}$$

By selecting $\theta_2 = \theta_t$ and $\theta_1 = \theta_{t+1}$ and noting that $\theta_{t+1} - \theta_t = \eta \bar{w}_t = \eta \frac{1}{N} \sum_{n=0}^{N-1} w_{t,n}$ we obtain,

$$\log \frac{\pi_t(a|z)}{\pi_{t+1}(a|z)} \le \frac{\eta^2}{2} \|\bar{w}_t\|_2^2 - \eta \langle \nabla_\theta \log \pi_t(a|z), \bar{w}_t \rangle. \tag{220}$$

Now, we separately bound the Lyapounov drift for the asymmetric and symmetric settings. In the following, some notations are overloaded across both setting when their meaning is clear from context. For the asymmetric setting, we have,

$$\Lambda(\pi_{t+1}) - \Lambda(\pi_t) = \sum_{z,a} d^{\pi^*}(z,a) \log \frac{\pi_t(a|z)}{\pi_{t+1}(a|z)} \tag{221}$$

$$\le \frac{\eta^2}{2} \|\bar{w}_t\|_2^2 - \eta \sum_{z,a} d^{\pi^*}(z,a) \langle \nabla_\theta \log \pi_t(a|z), \bar{w}_t \rangle \tag{222}$$

$$= \frac{\eta^2}{2} B^2 - \eta \sum_{s,z,a} d^{\pi^*}(s,z,a) \mathcal{A}^{\pi_t}(s,z,a) - \eta \sum_{s,z,a} d^{\pi^*}(s,z,a) \left( \langle \nabla_\theta \log \pi_t(a|z), \bar{w}_t \rangle - \mathcal{A}^{\pi_t}(s,z,a) \right) \tag{223}$$

$$\le \frac{\eta^2}{2} B^2 - \eta \sum_{s,z,a} d^{\pi^*}(s,z,a) \mathcal{A}^{\pi_t}(s,z,a) + \eta \sum_{z,a} d^{\pi^*}(s,z,a) \sqrt{\left( \langle \nabla_\theta \log \pi_t(a|z), \bar{w}_t \rangle - \mathcal{A}^{\pi_t}(s,z,a) \right)^2}. \tag{224}$$

For the symmetric setting, we observe instead,

$$\Lambda(\pi_{t+1}) - \Lambda(\pi_t) = \sum_{z,a} d^{\pi^*}(z,a) \log \frac{\pi_t(a|z)}{\pi_{t+1}(a|z)} \tag{225}$$

$$\le \frac{\eta^2}{2} B^2 - \eta \sum_{z,a} d^{\pi^*}(z,a) A^{\pi_t}(z,a) + \eta \sum_{z,a} d^{\pi^*}(z,a) \sqrt{\left( \langle \nabla_\theta \log \pi_t(a|z), \bar{w}_t \rangle - A^{\pi_t}(z,a) \right)^2}. \tag{226}$$

Now, let $\mathfrak{H}_t$ denote the sigma field of all samples used in the computation of $\pi_t$ (which excludes the samples used for computing $\bar{w}_t$), along with all the samples used in the computation of $\overline{Q}^{\pi_t}$. We define the ideal and approximate loss functions, both in the asymmetric and the symmetric setting,

$$\mathcal{L}_t(w) = \mathbb{E}\left[ \left( \langle \nabla_\theta \log \pi_t(A|Z), w \rangle - \mathcal{A}^{\pi_t}(S,Z,A) \right)^2 \Big| \mathfrak{H}_t \right] \tag{227}$$

$$\bar{\mathcal{L}}_t(w) = \mathbb{E}\left[\left(\langle\nabla_\theta\log\pi_t(A|Z), w\rangle - \bar{\mathcal{A}}^{\pi_t}(S,Z,A)\right)^2\Big|\mathfrak{H}_t\right] \tag{228}$$

$$L_t(w) = \mathbb{E}\left[\left(\langle\nabla_\theta\log\pi_t(A|Z), w\rangle - A^{\pi_t}(Z,A)\right)^2\Big|\mathfrak{H}_t\right] \tag{229}$$

$$\bar{L}_t(w) = \mathbb{E}\left[\left(\langle\nabla_\theta\log\pi_t(A|Z), w\rangle - \bar{A}^{\pi_t}(Z,A)\right)^2\Big|\mathfrak{H}_t\right]. \tag{230}$$

Because $\mathbb{E}\left[\left\|\mathcal{V}^{\pi_t} - \bar{\mathcal{V}}^{\pi_t}\right\|_{d^{\pi_t}}^2\Big|\mathfrak{H}_t\right] \leq \mathbb{E}\left[\left\|\bar{\mathcal{Q}}^{\pi_t} - \mathcal{Q}^{\pi_t}\right\|_{d^{\pi_t}}^2\Big|\mathfrak{H}_t\right]$, the error between the asymmetric advantage $\mathcal{A}$ and its approximation $\bar{\mathcal{A}}$ is upper bounded by,

$$\sqrt{\mathbb{E}\left[\left(\bar{\mathcal{A}}^{\pi_t}(S,Z,A) - \mathcal{A}^{\pi_t}(S,Z,A)\right)^2\Big|\mathfrak{H}_t\right]} = \sqrt{\mathbb{E}\left[\left\|\bar{\mathcal{A}}^{\pi_t} - \mathcal{A}^{\pi_t}\right\|_{d^{\pi_t}}^2\Big|\mathfrak{H}_t\right]} \tag{231}$$

$$= \sqrt{\mathbb{E}\left[\left\|\bar{\mathcal{Q}}^{\pi_t} - \bar{\mathcal{V}}^{\pi_t} - \mathcal{Q}^{\pi_t} + \mathcal{V}^{\pi_t}\right\|_{d^{\pi_t}}^2\Big|\mathfrak{H}_t\right]} \tag{232}$$

$$= \sqrt{\mathbb{E}\left[\left\|\bar{\mathcal{Q}}^{\pi_t} - \mathcal{Q}^{\pi_t} + \mathcal{V}^{\pi_t} - \bar{\mathcal{V}}^{\pi_t}\right\|_{d^{\pi_t}}^2\Big|\mathfrak{H}_t\right]} \tag{233}$$

$$\leq \sqrt{\mathbb{E}\left[\left\|\bar{\mathcal{Q}}^{\pi_t} - \mathcal{Q}^{\pi_t}\right\|_{d^{\pi_t}}^2 + \left\|\mathcal{V}^{\pi_t} - \bar{\mathcal{V}}^{\pi_t}\right\|_{d^{\pi_t}}^2\Big|\mathfrak{H}_t\right]} \tag{234}$$

$$\leq \sqrt{\mathbb{E}\left[\left\|\bar{\mathcal{Q}}^{\pi_t} - \mathcal{Q}^{\pi_t}\right\|_{d^{\pi_t}}^2\Big|\mathfrak{H}_t\right]} + \sqrt{\mathbb{E}\left[\left\|\mathcal{V}^{\pi_t} - \bar{\mathcal{V}}^{\pi_t}\right\|_{d^{\pi_t}}^2\Big|\mathfrak{H}_t\right]} \tag{235}$$

$$\leq 2\varepsilon_{\text{critic,asym}}^{\pi_t}, \tag{236}$$

where $\varepsilon_{\text{critic,asym}}^{\pi_t} = \varepsilon_{\text{td,asym}}^{\pi_t} + \varepsilon_{\text{app,asym}}^{\pi_t} + \varepsilon_{\text{shift,asym}}^{\pi_t}$ is given by the upper bound (29) in Theorem 3. Similarly, the error between the symmetric advantage $A$ and its approximation $\bar{A}$ is upper bounded by,

$$\sqrt{\mathbb{E}\left[\left(\bar{A}^{\pi_t}(Z,A) - A^{\pi_t}(Z,A)\right)^2\Big|\mathfrak{H}_t\right]} \leq 2\varepsilon_{\text{critic,sym}}^{\pi_t}, \tag{237}$$

where $\varepsilon_{\text{critic,sym}}^{\pi_t} = \varepsilon_{\text{td,sym}}^{\pi_t} + \varepsilon_{\text{app,sym}}^{\pi_t} + \varepsilon_{\text{shift,sym}}^{\pi_t} + \varepsilon_{\text{alias,sym}}^{\pi_t}$ is given by the upper bound (35) in Theorem 4. By using the inequality $(x + y)^2 \leq 2x^2 + 2y^2$,

$$\bar{\mathcal{L}}_t(w) = \mathbb{E}\left[\left(\langle\nabla_\theta\log\pi_t(A|Z), w\rangle - \bar{\mathcal{A}}^{\pi_t}(S,Z,A)\right)^2\Big|\mathfrak{H}_t\right] \tag{238}$$

$$= \mathbb{E}\left[\left(\langle\nabla_\theta\log\pi_t(A|Z), w\rangle - \mathcal{A}^{\pi_t}(S,Z,A) + \mathcal{A}^{\pi_t}(S,Z,A) - \bar{\mathcal{A}}^{\pi_t}(S,Z,A)\right)^2\Big|\mathfrak{H}_t\right] \tag{239}$$

$$\leq 2\mathbb{E}\left[\left(\langle\nabla_\theta\log\pi_t(A|Z), w\rangle - \mathcal{A}^{\pi_t}(S,Z,A)\right)^2\Big|\mathfrak{H}_t\right] + 2\mathbb{E}\left[\left(\mathcal{A}^{\pi_t}(S,Z,A) - \bar{\mathcal{A}}^{\pi_t}(S,Z,A)\right)^2\Big|\mathfrak{H}_t\right] \tag{240}$$

$$\leq 2\mathcal{L}_t(w) + 2(2\varepsilon_{\text{critic,asym}}^{\pi_t})^2. \tag{241}$$

Similarly, we obtain in the symmetric case,

$$\bar{L}_t(w) \leq 2L_t(w) + 2(2\varepsilon_{\text{critic,sym}}^{\pi_t})^2. \tag{242}$$

Starting from the ideal objective and following a similar technique, we also obtain,

$$\mathcal{L}_t(w) \leq 2\bar{\mathcal{L}}_t(w) + 2(2\varepsilon_{\text{critic,asym}}^{\pi_t})^2 \tag{243}$$

$$L_t(w) \leq 2\bar{L}_t(w) + 2(2\varepsilon_{\text{critic,sym}}^{\pi_t})^2. \tag{244}$$

By using Theorem 14.8 in (Shalev-Shwartz & Ben-David, 2014) with step size $\zeta = \frac{B\sqrt{1-\gamma}}{\sqrt{2N}}$, we obtain for the average iterate $\bar{w}_t$ under the asymmetric loss and symmetric loss, respectively,

$$\bar{\mathcal{L}}_t(\bar{w}_t) \leq \varepsilon_{\text{actor}}^2 + \min_{\|w\|_2 \leq B}\bar{\mathcal{L}}_t(w) \tag{245}$$

$$\bar{L}_t(\bar{w}_t) \leq \varepsilon_{\text{actor}}^2 + \min_{\|w\|_2 \leq B}\bar{L}_t(w), \tag{246}$$

where $\varepsilon_{\text{actor}}^2 = \frac{(2-\gamma)B}{2(1-\gamma)\sqrt{N}}$. On expectation, for the ideal asymmetric objective $\mathcal{L}_t$, we obtain,

$$\mathbb{E}\left[\mathcal{L}_t(\bar{w}_t)\right] \leq 2\mathbb{E}\left[\bar{\mathcal{L}}_t(\bar{w}_t)\right] + 2(2\varepsilon_{\text{critic,asym}}^{\pi_t})^2 \tag{247}$$

$$\leq 2\varepsilon_{\text{actor}}^2 + 2\min_{\|w\|_2 \leq B} \bar{\mathcal{L}}_t(w) + 2(2\varepsilon_{\text{critic,asym}}^{\pi_t})^2 \tag{248}$$

$$\leq 2\varepsilon_{\text{actor}}^2 + 2\left(2\min_{\|w\|_2 \leq B} \mathcal{L}_t(w) + 2(2\varepsilon_{\text{critic,asym}}^{\pi_t})^2\right) + 2(2\varepsilon_{\text{critic,asym}}^{\pi_t})^2 \tag{249}$$

$$= 2\varepsilon_{\text{actor}}^2 + 4\min_{\|w\|_2 \leq B} \mathcal{L}_t(w) + 6(2\varepsilon_{\text{critic,asym}}^{\pi_t})^2 \tag{250}$$

$$= 2\varepsilon_{\text{actor}}^2 + 4\left(\varepsilon_{\text{grad,asym}}^{\pi_t}\right)^2 + 6(2\varepsilon_{\text{critic,asym}}^{\pi_t})^2, \tag{251}$$

where we define the actor gradient function approximation error as,

$$\left(\varepsilon_{\text{grad,asym}}^{\pi_t}\right)^2 = \min_{\|w\|_2 \leq B} \mathcal{L}_t(w). \tag{252}$$

Similarly, we obtain on expectation for the ideal symmetric objective $L_t$,

$$\mathbb{E}\left[L_t(\bar{w}_t)\right] \leq 2\varepsilon_{\text{actor}}^2 + 4\left(\varepsilon_{\text{grad,sym}}^{\pi_t}\right)^2 + 6(2\varepsilon_{\text{critic,sym}}^{\pi_t})^2, \tag{253}$$

where we define the actor gradient function approximation error as,

$$\left(\varepsilon_{\text{grad,sym}}^{\pi_t}\right)^2 = \min_{\|w\|_2 \leq B} L_t(w). \tag{254}$$

Now, let us go back to the asymmetric and symmetric Lyapounov drift functions of equation (224) and (226). First, we assume that there exists $\overline{C}_\infty < \infty$ such that $\sup_{t \geq 0} \mathbb{E}[C_t] \leq \overline{C}_\infty$ with,

$$C_t = \sup_{s,z,a} \left| \frac{d^{\pi^*}(s,z)\pi^*(a|z)}{d^{\pi_{\theta_t}}(s,z)\pi_{\theta_t}(a|z)} \right|. \tag{255}$$

Second, we leverage the performance difference lemma to bound the advantage. For the asymmetric setting, the performance difference lemma for MDP (Kakade & Langford, 2002) holds because of the Markovianity of $(S_t, Z_t)$,

$$(1-\gamma)\left(V^{\pi^*}(s_0,z_0) - V^{\pi_t}(s_0,z_0)\right) = \mathbb{E}^{d^{\pi^*}}[\mathcal{A}^{\pi_t}(S,Z,A)|S_0 = s_0, Z_0 = z_0]. \tag{256}$$

We note that $\mathbb{E}\left[V^{\pi^*}(S_0, Z_0) - V^{\pi_t}(S_0, Z_0)\right] = \mathbb{E}\left[J(\pi^*) - J(\pi_t)\right]$, such that,

$$-\mathbb{E}^{d^{\pi^*}}[\mathcal{A}^{\pi_t}(S,Z,A)] = -(1-\gamma)\left(J(\pi^*) - J(\pi_t)\right). \tag{257}$$

$$= -(1-\gamma)\left(J(\pi^*) - J(\pi_t)\right) + \varepsilon_{\text{inf,asym}}, \tag{258}$$

where $\varepsilon_{\text{inf,asym}} = 0$. For the symmetric setting, using Lemma E.1 with $\pi_2 = \pi^*$ and $\pi_1 = \pi_t$, we note that,

$$(1-\gamma)\left(V^{\pi^*}(z_0) - V^{\pi_t}(z_0)\right) \leq \mathbb{E}^{d^{\pi^*}}[A^{\pi_t}(Z,A)|Z_0 = z_0] + 2\varepsilon_{\text{inf}}^{\pi^*}(z_0), \tag{259}$$

which implies,

$$-\mathbb{E}^{d^{\pi^*}}[A^{\pi_t}(Z,A)|Z_0 = z_0] \leq -(1-\gamma)\left(V^{\pi^*}(z_0) - V^{\pi_t}(z_0)\right) + 2\varepsilon_{\text{inf}}^{\pi^*}(z_0). \tag{260}$$

We note that $\mathbb{E}\left[V^{\pi^*}(Z_0) - V^{\pi_t}(Z_0)\right] = \mathbb{E}\left[J(\pi^*) - J(\pi_t)\right]$ and we denote $\mathbb{E}\left[\varepsilon_{\text{inf}}^{\pi^*}(Z_0)\right]$ with $\varepsilon_{\text{inf,sym}}$, so that,

$$\varepsilon_{\text{inf,sym}} = \mathbb{E}\left[\mathbb{E}^{\pi^*}\left[\sum_{k=0}^{\infty} \gamma^k \left\|\hat{b}_k - b_k\right\|_{\text{TV}} \middle| Z_0 = Z_0\right]\right] \tag{261}$$

$$= \mathbb{E}^{\pi^*}\left[\sum_{k=0}^{\infty}\gamma^k\left\|\hat{b}_k - b_k\right\|_{\mathrm{TV}}\right]. \tag{262}$$

By rearranging, we have,

$$-\mathbb{E}^{d^{\pi^*}}\left[A^{\pi_t}(Z,A)\right] \leq -(1-\gamma)\mathbb{E}\left[J(\pi^*) - J(\pi_t)\right] + 2\varepsilon_{\mathrm{inf,sym}}. \tag{263}$$

Note that $\sum_{s,z,a} d^{\pi^*}(s,z,a)f(s,z,a) = \sum_{s,z,a}\frac{d^{\pi^*}(s,z,a)}{d^{\pi_t}(s,z,a)}d^{\pi_t}(s,z,a)f(s,z,a) \leq C_t\sum_{s,z,a}d^{\pi_t}(s,z,a)f(s,z,a)$ for positive $f$. Taking expectation over the asymmetric Lyapounov drift of equation (224), we obtain using equation (255),

$$\mathbb{E}\left[\Lambda(\pi_{t+1}) - \Lambda(\pi_t)\right] \leq \frac{\eta^2}{2}B^2 - \eta\sum_{z,a}d^{\pi^*}(z,a)A^{\pi_t}(z,a)$$
$$+ \eta\sum_{s,z,a}d^{\pi^*}(s,z,a)\sqrt{\left(\langle\nabla_\theta\log\pi_t(a|z),\bar{w}_t\rangle - \mathcal{A}^{\pi_t}(s,z,a)\right)^2} \tag{264}$$

$$\leq \frac{\eta^2}{2}B^2 - \eta(1-\gamma)\mathbb{E}\left[J(\pi^*) - J(\pi_t)\right] + 2\eta\varepsilon_{\mathrm{inf,asym}}$$
$$+ \eta\overline{C}_\infty\sqrt{2\varepsilon_{\mathrm{actor}}^2 + 4\left(\varepsilon_{\mathrm{grad,asym}}^{\pi_t}\right)^2 + 6(2\varepsilon_{\mathrm{critic,asym}}^{\pi_t})^2} \tag{265}$$

$$\leq \frac{\eta^2}{2}B^2 - \eta(1-\gamma)\mathbb{E}\left[J(\pi^*) - J(\pi_t)\right] + 2\eta\varepsilon_{\mathrm{inf,asym}}$$
$$+ \eta\overline{C}_\infty\left(\sqrt{2}\varepsilon_{\mathrm{actor}} + 2\varepsilon_{\mathrm{grad,asym}}^{\pi_t} + 2\sqrt{6}\varepsilon_{\mathrm{critic,asym}}^{\pi_t}\right). \tag{266}$$

Similarly, taking expectation over the symmetric drift of equation (226), we obtain a similar expression,

$$\mathbb{E}\left[\Lambda(\pi_{t+1}) - \Lambda(\pi_t)\right] \leq \frac{\eta^2}{2}B^2 - \eta\sum_{z,a}d^{\pi^*}(z,a)A^{\pi_t}(z,a)$$
$$+ \eta\sum_{z,a}d^{\pi^*}(z,a)\sqrt{\left(\langle\nabla_\theta\log\pi_t(a|z),\bar{w}_t\rangle - A^{\pi_t}(z,a)\right)^2} \tag{267}$$

$$\leq \frac{\eta^2}{2}B^2 - \eta(1-\gamma)\mathbb{E}\left[J(\pi^*) - J(\pi_t)\right] + 2\eta\varepsilon_{\mathrm{inf,sym}}$$
$$+ \eta\overline{C}_\infty\left(\sqrt{2}\varepsilon_{\mathrm{actor}} + 2\varepsilon_{\mathrm{grad,sym}}^{\pi_t} + 2\sqrt{6}\varepsilon_{\mathrm{critic,sym}}^{\pi_t}\right). \tag{268}$$

Given the similarity of equation (266) and equation (268), in the following we denote the denote the upper bounds using $\varepsilon_{\mathrm{inf}}$, $\varepsilon_{\mathrm{grad}}^{\pi_t}$ and $\varepsilon_{\mathrm{critic}}^{\pi_t}$, irrespectively of the setting (i.e., asymmetric or symmetric).

By summing all Laypounov drifts, we obtain,

$$\mathbb{E}\left[\Lambda(\pi_T) - \Lambda(\pi_0)\right] \leq T\frac{\eta^2}{2}B^2 - \eta(1-\gamma)\sum_{t=0}^{T-1}\mathbb{E}\left[J(\pi^*) - J(\pi_t)\right] + 2\eta T\varepsilon_{\mathrm{inf}}$$
$$+ \eta\sum_{t=0}^{T-1}\overline{C}_\infty\left(\sqrt{2}\varepsilon_{\mathrm{actor}} + 2\varepsilon_{\mathrm{grad}}^{\pi_t} + 2\sqrt{6}\varepsilon_{\mathrm{critic}}^{\pi_t}\right) \tag{269}$$

$$\leq T\frac{\eta^2}{2}B^2 - \eta(1-\gamma)\sum_{t=0}^{T-1}\mathbb{E}\left[J(\pi^*) - J(\pi_t)\right] + 2\eta T\varepsilon_{\mathrm{inf}}$$
$$+ \eta\overline{C}_\infty\left(\sqrt{2}T\varepsilon_{\mathrm{actor}} + 2\sum_{t=0}^{T-1}\varepsilon_{\mathrm{grad}}^{\pi_t} + 2\sqrt{6}\sum_{t=0}^{T-1}\varepsilon_{\mathrm{critic}}^{\pi_t}\right). \tag{270}$$

Since $\pi_0$ is initialized at the uniform policy with $\theta_0 := 0$, we have,

$$\Lambda(\pi_0) = \sum_{z\in\mathcal{Z}}d^{\pi^*}(z)\mathrm{KL}(\pi^*(\cdot|z)\,\|\,\pi_0(\cdot|z)) \tag{271}$$

$$= \sum_{z \in \mathcal{Z}} d^{\pi^*}(z) \left( \sum_{a \in \mathcal{A}} \pi^*(a|z) \log \pi^*(a|z) - \sum_{a \in \mathcal{A}} \pi^*(a|z) \log \pi_0(a|z) \right) \tag{272}$$

$$= \sum_{z \in \mathcal{Z}} d^{\pi^*}(z) \left( \sum_{a \in \mathcal{A}} \pi^*(a|z) \log \pi^*(a|z) - \sum_{a \in \mathcal{A}} \pi^*(a|z) \log \frac{1}{|\mathcal{A}|} \right) \tag{273}$$

$$= \sum_{z \in \mathcal{Z}} d^{\pi^*}(z) \left( \sum_{a \in \mathcal{A}} \pi^*(a|z) \log \pi^*(a|z) + \log |\mathcal{A}| \right) \tag{274}$$

$$= \sum_{z \in \mathcal{Z}} d^{\pi^*}(z) \left( \log |\mathcal{A}| - H(\pi^*(\cdot|z)) \right) \tag{275}$$

$$\leq \sum_{z \in \mathcal{Z}} d^{\pi^*}(z) \log |\mathcal{A}| \tag{276}$$

$$\leq \log |\mathcal{A}|, \tag{277}$$

where $H$ denotes the Shannon entropy. Rearranging and dividing by $\eta T$, we obtain after neglecting $\mathcal{L}(\pi_T) > 0$,

$$(1 - \gamma) \frac{1}{T} \sum_{t=0}^{T-1} \mathbb{E}\left[ J(\pi^*) - J(\pi_t) \right] \leq \frac{\log |\mathcal{A}|}{\eta T} + \frac{\eta}{2} B^2 + 2\varepsilon_{\inf}$$
$$+ \overline{C}_\infty \left( \sqrt{2} \varepsilon_{\text{actor}} + 2 \frac{1}{T} \sum_{t=0}^{T-1} \varepsilon_{\text{grad}}^{\pi_t} + 2\sqrt{6} \frac{1}{T} \sum_{t=0}^{T-1} \varepsilon_{\text{critic}}^{\pi_t} \right). \tag{278}$$

It can also be noted that $\min_{0 \leq t < T} [x_t] \leq \frac{1}{T} \sum_{t=0}^{T} x_t$, which implies that,

$$(1 - \gamma) \min_{0 \leq t < T} \mathbb{E}\left[ J(\pi^*) - J(\pi_t) \right] \leq \frac{\log |\mathcal{A}|}{\eta T} + \frac{\eta}{2} B^2 + 2\varepsilon_{\inf}$$
$$+ \overline{C}_\infty \left( \sqrt{2} \varepsilon_{\text{actor}} + 2 \frac{1}{T} \sum_{t=0}^{T-1} \varepsilon_{\text{grad}}^{\pi_t} + 2\sqrt{6} \frac{1}{T} \sum_{t=0}^{T-1} \varepsilon_{\text{critic}}^{\pi_t} \right). \tag{279}$$

Let us define the worse actor gradient function approximation error,

$$\varepsilon_{\text{grad}} = \sup_{0 \leq t < T} \varepsilon_{\text{grad}}^{\pi_t} \tag{280}$$

$$= \sup_{0 \leq t < T} \sqrt{\min_{\|w\|_2 \leq B} L_t(w)}, \tag{281}$$

and let us note that,

$$\frac{1}{T} \sum_{t=0}^{T-1} \varepsilon_{\text{grad}}^{\pi_t} \leq \varepsilon_{\text{grad}}. \tag{282}$$

By setting $\eta = \frac{1}{\sqrt{T}}$, we obtain,

$$(1 - \gamma) \min_{0 \leq t < T} \mathbb{E}\left[ J(\pi^*) - J(\pi_t) \right] \leq \frac{\log |\mathcal{A}|}{\sqrt{T}} + \frac{B^2}{2\sqrt{T}} + 2\varepsilon_{\inf}$$
$$+ \overline{C}_\infty \left( \sqrt{2} \varepsilon_{\text{actor}} + 2 \frac{1}{T} \sum_{t=0}^{T-1} \varepsilon_{\text{grad}}^{\pi_t} + 2\sqrt{6} \frac{1}{T} \sum_{t=0}^{T-1} \varepsilon_{\text{critic}}^{\pi_t} \right) \tag{283}$$

$$= \frac{B^2 + 2\log |\mathcal{A}|}{2\sqrt{T}} + 2\mathbb{E}^{\pi^*} \left[ \sum_{k=0}^{\infty} \gamma^k \left\| \hat{b}_k - b_k \right\|_{\text{TV}} \right]$$
$$+ \overline{C}_\infty \left( \sqrt{\frac{(2 - \gamma)B}{(1 - \gamma)\sqrt{N}}} + 2\varepsilon_{\text{grad}} + 2\sqrt{6} \frac{1}{T} \sum_{t=0}^{T-1} \varepsilon_{\text{critic}}^{\pi_t} \right). \tag{284}$$

This concludes the proof. $\qquad \square$

