# OpenReview forum: "A Theoretical Justification for Asymmetric Actor-Critic Algorithms"
_ICML.cc/2025/Conference — ICML 2025 poster_

### Official Review · Reviewer_rwhu · 2025-02-20

**Overall Recommendation:** 4

**Summary:**

This paper investigates reinforcement learning in partially observable environments. The authors present finite sample analysis for neural actor-critic methods, comparing both asymmetric and symmetric critics. Their findings elucidate the relationship between the use of an asymmetric critic and the reduction of aliasing errors in the agent state.

**Claims And Evidence:**

Yes

**Essential References Not Discussed:**

No

**Experimental Designs Or Analyses:**

N/A

**Methods And Evaluation Criteria:**

Yes

**Other Comments Or Suggestions:**

Please refer to Question section.

**Other Strengths And Weaknesses:**

Strengths:
The paper presents a finite sample analysis for neural actor-critic methods, comparing both asymmetric and symmetric critics. The convergence rate achieved, $O(1/\sqrt{T} + 1/\sqrt{N} + 1/\sqrt{K})$, aligns well with state-of-the-art analyses in the field. The authors demonstrate that the actor-critic method using a symmetric critic may introduce aliasing errors, which does not arise with asymmetric critics.

Weakness:
Please refer to Question section.

**Questions For Authors:**

1. As I am not an expert in reinforcement learning within the POMDP setting, I found the background section somewhat confusing. Specifically, I would like clarification on the state space and agent state space: which one represents the state space accessible to the agent, and which one corresponds to the underlying true state space? Additionally, what is the distinction between $U$ and the policy $\pi$? Clearer definitions would greatly aid readers who may not have extensive knowledge of POMDPs.

2. In the definition of POMDPs, do both $|S|$ and $|A|$ need to be finite? I believe both are assumed to be finite in [1].

3. In Algorithms 1 and 2, the initial state-action pairs are required to be sampled from a discounted visitation distribution. Why can’t the starting point be chosen arbitrarily from the state-action space? Additionally, how is the starting point sampled from the discounted visitation distribution?

Reference:

[1] Cayci, Semih, Niao He, and R. Srikant. "Finite-time analysis of natural actor-critic for pomdps." SIAM Journal on Mathematics of Data Science 6.4 (2024): 869-896.

**Relation To Broader Scientific Literature:**

N/A

**Theoretical Claims:**

Yes

---

> ### Author Rebuttal · Authors · 2025-03-31
>
> Dear Reviewer,
>
> Thank you for your review and for hihglighting that we attain state-of-the-art convergence rate. Below, we answer to your remarks.
>
> **Environment and agent states.** We agree that some other key concepts could be better introduced in the background. The environment state is typically inaccessible to the agent, which is provided with a stream of observations instead. From these observations, the agent maintain a state, updated recurrently after each action and new observation, that we call the agent state. This is the state accessible to the policy. In the asymmetric learning paradigm, we assume that in addition to the agent state, the agent is provided with the environment state *at training time only*. **We propose to improve the background to better explain and illustrate with examples what could be typical observations, environment states, agent states, update functions, and what would be aliasing in this context.** We could notably use the example discussed in our response to Reviewer UL7j.
>
> **Update function and policy.** In a sense, the update function simply implements the processing of the history that is done before passing the (stochastic) feature $z$ of the history $h$ to the policy $\pi$. Together, the update function $U$ and the policy $\pi$ thus implement the history-dependent function that controls the POMDP. Only the policy $\pi$ is considered learnable, and common examples for the update function $U$ are a window of past observations, the belief/Kalman filter (if the latter is known), etc. **We propose to clarify this distinction and to give some examples to help readers outside of the field to better understand this.** Please note that we also discuss in the conclusion the fact that the proof straightforwardly adapts to an agent state process $U$ that is learned through RL.
>
> **Finite states.** All state spaces ($\mathcal{S}$, $\mathcal{A}$, $\mathcal{O}$ and $\mathcal{Z}$) are considered finite in this analysis. **It is indicated in Subsection 2.1, but if you feel that we do not insist enough on that in the introduction or anywhere else in the main text, please let us know and we would be happy to address this.**
>
> **Initial sampling distribution.** Concerning the temporal difference learning algorithm (Algorithm~1) and the finite-time bound for the critics (Theorem 3 and Theorem 4), any initial distribution $p(s,z,a)$ could have been chosen, and the bound would have been obtained with norm $\lVert\cdot\rVert_p$ instead of $\lVert\cdot\rVert_d$. However, since we need to take samples from the discounted visitation measure $d = d^\pi \otimes \pi$ for the policy-gradient (as is standard for on-policy policy-gradient), the performance bound is expressed in terms of expectation under that distribution. As a consequence, we thus need to know a bound on the critic error under that norm $\lVert\cdot\rVert_d$, which explains why the temporal difference algorithm was studied under that sampling distribution. **We propose to add a note in our paper to clearly state that the analysis of Theorem 3 and Theorem 4 can be straightforwardly adapted to any sampling distribution $p(s,z,a)$.**
>
> **Sampling algorithm.** The algorithm implicitly assumes that we have an oracle allowing to sample from this distribution $d = d^\pi \otimes \pi$, as is standard in prior works. **We will clarify this in the algorithm.** In any case, there exists a tractable algorithm for sampling from this distribution. This algorithm samples an initial timestep $t_0 \sim \text{Geom}(1-\gamma)$ from a geometric distribution with success rate $1 - \gamma$, and then the current policy would be used to take $t_0 - 1$ actions in the environments, and would then sample a last action. The environment state $s_{t_0}$, agent state $z_{t_0}$ and action $a_{t_0}$ constitute a sample from this distribution $d$.
>
> We hope that you will find these adaptations pertinent to improve the overall accessibility of the document, and please do not hesitate if you need us to clarify something further.

---

> > ### Comment · Reviewer_rwhu · 2025-04-06
> >
> > Thank you to the authors for their detailed response. My concerns have been fully addressed. I suggest clarifying the background further, as this would be beneficial for readers who may be less familiar with POMDPs. In light of the improvements made, I am increasing my score to 4.

---

> > > ### Author Response · Authors · 2025-04-06
> > >
> > > We thank the reviewer for increasing their score. We will expand the background to clarify environment and agent states and also make the other changes discussed in our reply.

---

### Official Review · Reviewer_UL7j · 2025-03-13

**Overall Recommendation:** 4

**Summary:**

This paper analyzes two different versions of a natural actor-critic method for learning policies in finite POMDPs. The first version is "symmetric": both actor and critic depend only on the internal agent state. The second version is "asymmetric". Here, the critic depends also on the underlying environment state, a privileged information that is often available during training. The authors prove convergence bounds for both methods, and show that the asymmetric method has an edge over the symmetric one.

**Claims And Evidence:**

All claims are supported by clear and convincing evidence.

**Essential References Not Discussed:**

None that I am aware of.

**Experimental Designs Or Analyses:**

This is a purely theoretical paper.

**Methods And Evaluation Criteria:**

This is a purely theoretical paper.

**Other Comments Or Suggestions:**

### Minor comments
- Section 3 (nitpicking): I assume that the set $\mathcal C$ closed and convex, such that the projection is properly defined? Also, do you assume that $\mu$ has full support? Otherwise $\lVert\cdot\rVert_\mu$ is technically a seminorm (not sure if this is relevant).
- Section 3.1: I don't think you defined the distribution $d$ in the expressions $\lVert\cdot\rVert_d$. Is it simply $d(s, z, a) = d^\pi(s, z)\pi(a | z)$?
- Section 3.2: I suppose now $d(z, a) = \sum_{s \in \mathcal S} d^\pi(s, z)\pi(a | z)$?
- Eq. 16: Typo: $\mathcal F^B_\chi$
- Eq. 19: Why is $\pi_\theta$ "log-linear"? Specifically, is $\log \pi_\theta$ really a linear function of $\theta$, even though it includes a log-sum-exp term?
- Section 4.1/4.2: Typo: the errors of the Q-function should be measured by a seminorm weighted over not just states and agent states, but also actions, (e.g. the norm $d^\pi(s, z)\pi(a | z)$). The equations here (e.g. Eq. 28) just use $d^\pi$.
- Regarding the aliasing error (Eq. 38), does the expected distance between the beliefs depend on $k$? If not, the expression inside the weighted norm could be simplified as $\frac{1}{1 - \gamma^m}\mathbb E^\pi[\lVert\hat b_{0, m} - b_{0, m}\rVert]$
- Eq. 39: I assume that $\pi_t \doteq \pi_{\theta_t}$?
- Theorem 5: What is $R$ in the definition of $\zeta$? (Is it the maximum reward?)
- In $\varepsilon_{\mathrm{inf}}$, what are $\hat b_k$ and $b_k$? (Does it refer to the subroutine Algorithm 1 inside Algorithm 2?)

**Other Strengths And Weaknesses:**

The paper is very well written. I had no problem following despite not being very familiar with the prior works. Additionally, the result is interesting and seems very relevant.

**Questions For Authors:**

- Line 163 (left column): This is an interesting point. Are you saying that the symmetric Q-function $Q^\pi$ generally does not satisfy the $m$-step symmetric Bellman equation? Is there a counterexample?
- Algorithms 1 and 2 assume i.i.d. sampling from $d^\pi$. In a more realistic setting, the samples are generated from online interaction with the environment. Did you consider this setting as well?
- What is the motivation for the "asymmetric obective", Eq. 22? If I understand correctly, you want to justify why $w_*^{\pi_\theta}$ is a good parameter update ("it minimizes this loss"). Or are you introducing the loss function $\mathcal L$ for another reason, namely that $w_*^{\pi_\theta}$ is a minimizer of $\mathcal L$ and $\mathcal L$ can be more easily optimized than solving equation (20)?
- Can you give some intuition about the assumption Eq. 39?
- Line 356 (left column). What do you mean by "asymmetric learning is insensitive to aliasing in the agent state"? Clearly also asymmetric learning is impacted by the error that comes from incomplete state information. (For example, $\varepsilon_{\mathrm{inf}}$ captures some of this error). Additionally, this sentence sounds a bit as though you prove that asymmetric learning is better than symmetric learning. However, the paper presents only upper bounds on the risk, meaning that no such conclusion can be drawn (this would require lower bounds). Could you comment on this, and how much you think this conclusion ("asymmetric is better than symmetric") to depend on the specific algorithm?

**Relation To Broader Scientific Literature:**

The authors clearly position their work within the context of the existing literature.

**Theoretical Claims:**

I checked the results for obvious flaws, but did not read the proofs in detail.

---

> ### Author Rebuttal · Authors · 2025-03-31
>
> Dear Reviewer,
>
> Thank you for you review. We are happy that you found the paper convincing and accessible. Please find our answers below.
>
> **Minor remarks.** All your remarks and typos were valid. The set needs to be closed, we can assume it strictly convex to ensure the uniqueness of the projection. We consider distributions with full support, which is reasonable regarding the policy parametrization (no action has a zero probability). Log-linear policies indeed denote policies whose logarithm is not linear in $\theta$. In Thm. 5, $R$ should be $B$ in the definition of $\zeta$. Finally, $b_k$ and $\hat{b}_k$ are not related to the subroutine, but are well-defined random variables. **We will clarify all this.**
>
> **Aliasing Bias.** An example where $Q^\pi(z,a)$ is not equal to $\tilde{Q}^\pi(z,a)$ is the following POMDP with $z_t=o_t$. There are two starting states (L or R), that are observed perfectly. Any action taken from these states always yields a zero reward. The agent can switch to the other starting state, or go to the corresponding absorbing state (G or P). In the absorbing states, he stays there forever with the corresponding reward (±1). The policy is $\pi(a|z)=go,\forall z\in\mathcal{Z}$. It gives $Q^{\pi}(o=L,a=down)=\frac{\gamma}{1-\gamma}$ and $\tilde{Q}^\pi(o=L,a=down)=0$ because $\tilde{Q}^\pi(o=D,a=\cdot)=0$. **If you think this example would be worth being added to the paper, we would be happy to do so.**
> - $O(L|L)=O(R|R)=1$
> - $O(D|G)=O(D|P)=1$
> - $R(+1|G,:)=R(-1|P,:)=1$
> - $R(0|L, :)=R(0|R,:)=1$
> - $A=\\{swap, go\\}$
>
> **IID samples.** We tried deriving the proof for correlated samples (with the usual assumption that we have reached a stationary distribution) following similar steps as [Bhandari et al. (2018)](https://arxiv.org/abs/1806.02450). While it is feasible to derive this proof for the asymmetric critic, the symmetric setting seemed more subtle and we did not succeed. It would thus have prevented a direct comparison. We can cite a concurrent work [(Cai et al., 2024)](https://arxiv.org/abs/2412.00985) that does not make the IID assumption. However, it uses an explicit belief approximation, which makes the comparison with standard asymmetric AC difficult. **Despite not required by the concurrent work policy of ICML, we propose to discuss this reference, because we believe it is an interesting related work.** Please note that Thm. 3-4 could be straightforwardly adapted to any distribution instead of $d$, provided that we still sample IID.
>
> **NPG Objective.** The NPG $w_\*^{\pi_\theta}$ is considered a good choice, motivated in the literature from a long time [(Kakade, 2001)](https://papers.nips.cc/paper/2073-a-natural-policy-gradient). It can be seen as a standard PG, but where we select an appropriate metric for the parameter space. However, since computing the Fisher information matrix is not trivial, we prefer to minimize these simple convex losses ($\mathcal{L}$ and $L$), which are both proven to be minimized by the NPG $w_\*^{\pi_\theta}$. **We propose to better explain the NPG in the paper, to make things more clear for the reader.**
>
> **Concentrability coefficient.** Eq. (39) shows a concentrability coefficient, assumed to upper bound the ratio of probabilities between the optimal policy and any policy $\pi_t$. It roughly states that the policy should be stochastic enough so as to visit all state-action pairs from the optimal policy with nonzero probability throughout the learning process. Thm. 5 states that the lower this concentrability coefficient, the lower the upper bound on the suboptimality. It notably motivates the initialization of the policy to the max-entropy policy. **We propose to add a better explanation of this hypothesis in the paper.**
>
> **Insensitivity to aliasing.** Given the current analysis, you are totally right, we overlooked the $\epsilon_\text{inf}$ term and the claim was too strong. However, after double checking, you pointed out something interesting: this term can be removed in the asymmetric case. Indeed, we could stop at equation (222) instead of (223) so that we keep $A^\pi(S,Z,A)$ instead of using $A^\pi(Z,A)$. Then, in (256-258) we can use the MDP version of Lemma D.1 with $A^{\pi^+}(S,Z,A)$ instead of $A^\pi(Z,A)$, where $\pi^+$ is defined as in Appendix A.1, which does not have the $\epsilon_\text{inf}$ term [(Kakade & Langford, 2002)](https://homes.cs.washington.edu/~sham/papers/rl/aoarl.pdf). Finally, when substituting back (258) in (260), it would make this term disappear from the final result. **While the previous proof was valid, this result is even stronger, and we could add this improvement to the paper.** We also agree that we should not claim that asymmetric learning is insensitive to aliasing, but that we have an *upper bound* that is not dependent on the level of aliasing. **We will fix that in the paper.**
>
> We hope that these discussions will have answered your questions. Please do not hesitate to request additional explanations.

---

> > ### Comment · Reviewer_UL7j · 2025-04-04
> >
> > I thank the authors for their clarifying comments, and I will keep my score as is.
> >
> > I think it would be beneficial to include the example that you provide here (in the appendix, for curious readers). However, I don't quite understand it yet. If possible, could you please elaborate on the dynamics of this POMDP: how does the state change given the action, and what is the action "down" (is it "swap" or "go")? Thank you!

---

> > > ### Author Response · Authors · 2025-04-06
> > >
> > > Dear Reviewer,
> > >
> > > Thank you for your response. We will add the example in the appendix. There was indeed a typo in which “down” should have been understood as the action “go”, sorry for the inconvenience. Here is a more detailed explanation.
> > >
> > > We consider a POMDP $\mathcal{P} = (\mathcal{S}, \mathcal{A}, \mathcal{O}, T, R, O, P, \gamma)$ where $\mathcal{S} = \\{ L, R, G, P \\}$ (left, right, goal, pit), $\mathcal{A} = \\{ \text{swap}, \text{go} \\}$ and $\mathcal{O} = \\{ L, R, D \\}$ (left, right, down). The agent state $z_t \in \mathcal{Z} = \mathcal{O}$ is considered to be the last observation $o_t$ only, i.e. $U(z_t | z_{t-1}, a_{t-1}, o_t) = \delta_{o_t}(z_t)$. The states and corresponding observations are depicted below, along with the rewards obtained when taking any actions from a given state. We also detail the probability functions $T$, $R$, $O$, and $P$.
> > > ```
> > >  states       observations     rewards
> > > +---+---+      +---+---+      +---+---+
> > > | L | R |      | L | R |      | 0 | 0 |
> > > +---+---+      +---+---+      +---+---+
> > > | G | P |      | D | D |      |+1 |-1 |
> > > +---+---+      +---+---+      +---+---+
> > >
> > > We can swap observable states:
> > > - T(L | R, swap) = T(R | L, swap) = 1
> > > We can go to absorbing states:
> > > - T(G | L, go) = T(P | R, go ) = 1
> > > We stay in absorbing states
> > > - T(G | G, :) = T(P | P, :) = 1
> > >
> > > We observe the observable state:
> > > - O(L | L) = O(R | R) = 1
> > > We do not observe the absorbing state (aliasing):
> > > - O(D | G) = O(D | P) = 1
> > >
> > > No reward from observable states:
> > > - R(0 | L, :) = R(0 | R, :) = 1
> > > Rewards from absorbing states:
> > > - R(+1 | G, :) = 1
> > > - R(-1 | P, :) = 1
> > >
> > > Uniform distribution over initial states:
> > > - P(L) = P(R) = P(G) = P(P) = 0.25
> > > ```
> > >
> > > Let us now consider the policy $\pi(\text{go} | z) = 1, \forall z \in \mathcal{Z}$. Since $Pr(s = G | z = D) = \Pr(s = P | z = D) = 0.5$, we have $Q^\pi(z = D, a) = \tilde{Q}^\pi(z = D, a) = 0, \forall a \in \mathcal{A}$.
> > > It is also easy to see that $Q^{\pi}(z = L,a = \text{go})=\frac{\gamma}{1-\gamma}$.
> > > However, we obtain $\tilde{Q}^\pi(z = L,a=\text{go})=0$ because $\tilde{Q}^\pi(o=D,a=a)=0, \forall a \in \mathcal{A}$, which indeed results in $Q^\pi \neq \tilde{Q}^\pi$.
> > >
> > > In addition, it is worth noting that the choice of agent state was reasonable for this POMDP. Indeed, the agent state $z_t = o_t$ is sufficient for optimal control (i.e., there exists an optimal policy conditioned on $z_t$ that is as good as the optimal policy conditioned on the complete history $h_t$).

---

### Official Review · Reviewer_Y2Go · 2025-03-14

**Overall Recommendation:** 4

**Summary:**

This paper provides a theoretical justification for the empirical success of asymmetric actor-critic algorithms in partially observable environments. Using finite-time convergence analysis, the authors prove that asymmetric critic methods (which leverage additional state information during training) eliminate an "aliasing error" term that appears in symmetric methods. This error term arises from the difference between the true belief distribution (given history) and the approximate belief distribution (given agent state). The mathematical analysis demonstrates that this is the only difference in the error bounds between asymmetric and symmetric approaches, providing a clear theoretical explanation for why asymmetric methods often converge faster and perform better in practice.

**Claims And Evidence:**

The paper claims that asymmetric actor-critic algorithms eliminate an error term due to aliasing, leading to improved convergence compared to symmetric algorithms. It also claims that the theoretical bounds hold for finite state policies and linear function approximators.

The claims are supported by mathematical analysis:
1. Formalize both symmetric and asymmetric actor-critic algorithms
2. Derive finite-time bounds for both approaches
3. Identify the specific error term (aliasing error) that exists in symmetric but not asymmetric methods
4. Show that this is the only difference between the error bounds

The paper lacks empirical validation.

**Essential References Not Discussed:**

The paper covers the most relevant literature, but could benefit from discussing:

1. **Information-theoretic approaches to POMDPs**: The paper by Eysenbach et al. (2023) "Information Asymmetry in KL-Regularized RL" (NeurIPS 2023) directly addresses quantifying and mitigating observation aliasing through an information-theoretic lens, showing how information asymmetry creates similar benefits to the asymmetric actor-critic approach. This work provides complementary theoretical insights about the same underlying issue.

2. Chen et al. (2020) "Learning with Privileged Information for Efficient Image Super-Resolution" (ECCV 2020) employs a teacher-student framework with privileged information during training that closely parallels the asymmetric actor-critic approach. Their theoretical analysis of knowledge distillation in asymmetric settings could provide useful insights for reinforcement learning.

Both works approach the problem of leveraging privileged information during training from different angles and could strengthen the contextual understanding of asymmetric learning approaches.

**Experimental Designs Or Analyses:**

The paper does not include experimental validation, which is a significant limitation.

**Methods And Evaluation Criteria:**

**Methods:** The theoretical approach is sound for analyzing the problem. The authors adapt existing convergence analysis techniques to both symmetric and asymmetric settings, enabling direct comparison. The mathematical framework correctly models partially observable environments and agent behavior.

**Evaluation Criteria:** The evaluation is purely theoretical, with no empirical experiments. While the theoretical bounds are well-justified, the lack of empirical evaluation limits the practical relevance of the results.

**Other Comments Or Suggestions:**

### Suggestions for Empirical Validation
The theory could be empirically validated with a simple, specific POMDP grid world environment:

#### **Setup**:
- **Environment**: 5×5 grid with deterministic movements.
- **True State**: Full (x, y) position of the agent.
- **Observation**: Limited 3×3 visual field centered on the agent, showing wall/path/goal.
- **Controlled Aliasing**: Multiple visually identical room patterns are deliberately placed in different positions (e.g., hallway segments that look identical but lead to different rewards).
- **Precise Aliasing Mechanism**: The environment uses a fixed set of 3×3 templates that repeat in different locations, creating true state aliasing.

#### **Control Parameter**:
- Vary the template repetition frequency (e.g., 10%, 30%, 50% of states share identical observations with at least one other state).

#### **Experiment**:
1. Compare symmetric critic (using only observations) vs. asymmetric critic (with access to true x,y position)
2. Measure learning curves, final performance, and sample efficiency
3. Calculate belief state entropy at each position to quantify the actual impact of aliasing

This controlled environment would directly test whether the performance gap between methods increases proportionally with aliasing severity, as predicted by the theoretical bounds in **Theorems 3 and 4**.

---

### Suggestions for Improving Accessibility

Provide more intuitive explanations of key concepts, such as aliasing, to make the paper more accessible to a broader audience.

**Other Strengths And Weaknesses:**

**Strengths:**
- Clear identification of the specific mathematical advantage of asymmetric methods
- Rigorous theoretical analysis with well-defined bounds
- Addresses an important gap between empirical success and theoretical understanding

**Weaknesses:**
- Restricted to linear function approximation and finite state policies, limiting applicability to deep RL
- Lacks empirical validation even in simple environments

**Questions For Authors:**

1. Could you provide empirical validation of your theoretical findings in a simple environment with controllable degrees of aliasing? This would help verify that the performance gap between symmetric and asymmetric methods increases with aliasing severity as predicted by the theory.

2. How do your theoretical results extend to non-linear function approximation as commonly used in deep RL? The current analysis is limited to linear approximators.

**Relation To Broader Scientific Literature:**

This work connects to several research directions:

1. **Asymmetric learning paradigms**: Builds on prior work by Pinto et al. (2018) and Baisero & Amato (2022), providing theoretical justification for previously heuristic methods.
2. **POMDP solving**: Relates to approaches for partially observable environments, complementing techniques like recurrent neural networks and belief state modeling.
3. **Convergence analysis**: Extends finite-time bounds from Cayci et al. (2024) to the asymmetric setting.
4. **State representations**: Has implications for representation learning in POMDPs and the benefits of leveraging privileged information.

**Theoretical Claims:**

I verified the key theoretical claims, particularly:

1. **Theorems 1 and 2 (Natural Policy Gradients)**: The proofs correctly show that both symmetric and asymmetric approaches optimize similar objectives.
2. **Theorem 3 and 4 (Finite-time bounds)**
3. **Lemma C.1 (Upper bound on aliasing)**: The derivation connecting aliasing error to total variation distance between belief distributions is mathematically sound.
The mathematical analysis is rigorous, though some proof steps could benefit from additional explanation for clarity.

---

> ### Author Rebuttal · Authors · 2025-03-31
>
> Dear Reviewer,
>
> Thank you for your review. We are happy to read that you appreciated our manuscript and that you considered that we addressed an important gap in the theoretical understanding of these methods. Please find our answers below.
>
> **Toy experiment.** We agree that an empirical validation of the findings of this paper could be interesting. However, given the short rebuttal period, it will probably not be possible to obtain these results on time. We hope that you will still consider the results interesting and worthy of publication.
>
> **Clarity.** After proofreading Lemma C.1, we completely agree that some things needed to be improved to help the reader understand the philosophy of the proof, and the origin of aliasing. In particular, we think it would be useful to justify the initial expression of $Q$ and $\tilde{Q}$, which would help understanding that the bias comes from the difference in distribution at the bootstrapping timestep $m$, when we condition on $z_m$ versus $h_m$. **We propose to add this discussion in the proof of the Lemma.**
>
> **Related Works.** We are not sure what papers you wanted to refer exactly. We found papers with the same titles but different bibliographic data. Could it be that you wanted to refer the papers "Information Asymmetric in KL-Regularized RL" (Galashov et al., ICLR 2019) and the paper "Learning with Privileged Information for Eﬃcient Image Super-Resolution" (Lee et al., ECCV 2020)? **We definitely think that the first paper would be worth mentioning in the introduction.** Regarding the second one, we feel like discussing this vision paper may not be that much relevant. Please note that we already mentioned seminal "teacher-student" approaches in RL in our introduction (Choudhury et al., 2018; Warrington et al., 2019).
>
> **Nonlinear approximators.** This is an interesting future work. Moreover, as discussed above, this analysis could also embed the analysis of a learnable RNN, by considering a fixed window as the fixed agent state process. This follow-up work could perhaps build on the analysis proposed by Cayci & Eryilmaz (2024a, 2024b). **We propose to adapt the discussion about future works to make that more clear in the paper.**
>
> **Aliasing.** We acknowledge a confusion in our paper about the definition of aliasing. We think that (perceptual) aliasing should be understood as the fact that an observation $o$ (or an agent state $z$) corresponds to different  $h$. This is a direct consequence of $o$ (or $z$) being non Markovian. As a consequence of aliasing, the fixed point $\tilde{Q}(z,a)$ is not equal to $Q(z,a)$. We also referred to this as aliasing, but we agree that we should call this the aliasing bias. Then, Lemma C.1 highlights in (139) that the aliasing bias $\lVert\tilde{Q}-Q\rVert$ is given by the difference in distribution of $s$ at the bootstrapping timestep $m$, depending on whether we condition on $z_m$ or $h_m$. This bias is then upper bounded by the TV between these distributions times the maximum rewards. We call this term the aliasing term. **We propose to revise the paper by fixing the naming convention for these three aspects (aliasing, aliasing bias, aliasing term), and by better discussing the origin of aliasing.** In addition, we agree that some other key concepts could be better introduced in the background. **We propose to improve the background to better explain and illustrate with examples what could be typical observations, agent states, environment states update functions, and what would be aliasing in this context.** We could notably use the example discussed in our response to Reviewer UL7j.
>
> We hope that these explanations will have answered all your questions. Please do not hesitate if you want us to further elaborate on something that was unclear.

---

### Official Review · Reviewer_VgVc · 2025-03-19

**Overall Recommendation:** 3

**Summary:**

This paper provides a finite-time analysis of asymmetric AC approach in POMDPs, showing that leveraging privileged state information during training eliminates uncertainty errors inherent in symmetric critics. Theoretically, it compares convergence bounds for asymmetric/symmetric critics, proving that asymmetric critics remove the so called aliasing term which comes form the uncertainty inherent in symmetric critics.

**Claims And Evidence:**

The claim of uncertainty error of the symmetric critic is well supported and provided with formal proof.

**Essential References Not Discussed:**

It seems that the paper is citing all the prominent related works.

**Experimental Designs Or Analyses:**

N/A

**Methods And Evaluation Criteria:**

This paper do not propose methods or included evalution.

**Other Comments Or Suggestions:**

How would the bounds change if the agent state process M is learned (e.g., via RNNs) rather than fixed?
I would suggest including toy domains as a sanity check, could simple synthetic POMDPs (e.g., darkroom) illustrate the impact of the uncertainly error?

**Other Strengths And Weaknesses:**

I appreciate that fact that the paper is highly well-written, the presentation is logically structured, with well-defined notation with a very focused narrative. The finite-time bounds (Theorems 3–5) are derived meticulously, extending prior NAC analyses (e.g., Cayci et al.) to asymmetric settings loosely following the style of Baisero et al.. The connection between aliasing and critic bias is clearly formalized.

Nevertheless, I would like to comment that aliasing in POMDPs may be usually understood as aliasing multiple histories into one state, emphasizing on a many-to-one relationship as suppose to differences in belief distributions; hence I consider the term to be a bit abused in this paper. The core idea—using privileged state information to mitigate aliasing—echoes Baisero & Amato, who used h instead of z; while the analysis is novel, the conceptual leap is modest. Now, moving to the theoretical results, although we understand that we are short of one error term, one is to be uncertain whether the difference in error can be made up by less variance in learning targets, and there are no further discussion or tests on such fronts.

**Questions For Authors:**

Could the analysis be extended to non-linear function approximators (which is common), where aliasing might interact with approximation errors differently, at least include a few sentence of discussion?

**Relation To Broader Scientific Literature:**

This work is a relatively small patch on the popular AC method, turning what's previously floating intuition into solid terms, this work have the potential for summoning further theoretical works on asymmetric AC.

**Theoretical Claims:**

I checked for the correctness of the theoretical claims, the proofs seem well-established, well-organized and error-free.

---

> ### Author Rebuttal · Authors · 2025-03-31
>
> Dear Reviewer,
>
> Thank you for your review. We are glad that you found the results interesting and the paper clear. Please find our answers below.
>
> **Aliasing.** We acknowledge a confusion in our paper about the definition of aliasing. We agree that aliasing could refer to the fact that an history $h$ corresponds to the different environment states $s$. However, it would not pose any problem since $h$ is the Markovian state of an equivalent MDP: the "history MDP". We instead think that (perceptual) aliasing should be understood as the fact that an observation $o$ (or an agent state $z$) corresponds to different $h$. This is a direct consequence of $o$ (or $z$) being non Markovian. As a consequence of aliasing, the fixed point $\tilde{Q}(z,a)$ is not equal to $Q(z,a)$. We also referred to this as aliasing, but we agree that we should call this the aliasing bias. Then, Lemma C.1 highlights in (139) that the aliasing bias $\lVert\tilde{Q}-Q\rVert$ is given by the difference in distribution of $s$ at the bootstrapping timestep $m$, depending on whether we condition on $z_m$ or $h_m$. This bias is then upper bounded by the TV between these distributions times the maximum rewards. We call this term the aliasing term.  Cayci et al. (2024) try to mitigate this bias using multistep TD learning, while we show that the problem disappear when considering asymmetric learning. **We propose to revise the paper by fixing the naming convention for these three aspects (aliasing, aliasing bias, aliasing term), and by better discussing the origin of aliasing.**
>
> **Variance effect.** We agree that, with the law of total variance, for any $\Pr(S,Z)$, any $z\in\mathcal{Z}$ and any $a\in\mathcal{A}$, we have $\mathbb{E}[\mathbb{V}[\sum_t\gamma^tR_t|S_0=S,Z_0=z,A_0=a]]\leq\mathbb{V}[\sum_t\gamma^tR_t|Z_0=z,A_0=a]$. It could suggest that using $Q(s,z,a)$ instead of $Q(z,a)$ in the PG will provide less variance. However, this neglects the variance of the estimator $\hat{Q}(s,z,a)$ that may have a higher variance than $\hat{Q}(z,a)$, since it is learned on fewer samples. In conclusion, it not clear what will be the effect of the variance of the target value functions, and we consider this study out of the scope of this paper. We also highlight that the claims on the effect of variance from Baisero & Amato (2022) differ from these from Sinha & Mahajan (2023), which further motivate an empirical study. Anyway, our bound is given on average over the complete learning process, whatever the variance of the learning targets. It thus still gives an indication on the worse case average performance of asymmetric learning compared to symmetric learning, even if there is more/less variance in the targets. **We can expand the discussion in the paper if you find it will be useful.**
>
> **Learning the update.** The analysis perfectly holds for an agent state process $\mathcal{M}$ learned through RL. By (i) extending the action space $\mathcal{A}$ with the agent state space $\mathcal{Z}$: $a^+=(a,z')$, (ii) extending the agent state $\mathcal{Z}$ with the observation space $\mathcal{O}$: $z^+=(z,o)$, and (iii) considering an update $U$ that transition to the selected agent state: $z^{+'}=(z',o')=U(z^+,a^+,o')$ where $a^+ = (a,z')$, we can explicitly learn an update by selecting both the environment action and the next agent state $a^+=(a,z')\sim \pi^+(·|z,o)$. In the conclusion, we discuss a more general version of this where the agent can learn a stochastic update. In both cases, the analysis goes through, we are just selecting a particular POMDP and update $U$. **If you feel that this is not clear, we can discuss this further in the paper.** We acknowledge that this is different from the usual way of learning $U_\psi$ with an RNN and BPTT. However, even when using an RNN, we always consider a fixed $\mathcal{M}$ to learn from (usually a window, on which we do BPTT). The analysis for a learned $\mathcal{M}$ can thus be cast as the problem of learning nonlinear features of the window $z$, where the nonlinear feature $\psi(z,a)$ is an RNN. **We propose to add this discussion in the paper.**
>
> **Toy experiment.** We agree that an empirical validation of the findings of this paper could be interesting. However, given the short rebuttal period, it will probably not be possible to obtain these results on time. We hope that you will still consider the results interesting and worthy of publication.
>
> **Nonlinear approximators.** This is an interesting future work. Moreover, as discussed above, this analysis could also embed the analysis of a learnable RNN, by considering a fixed window as the fixed agent state process. This follow-up work could perhaps build on the analysis proposed by Cayci & Eryilmaz (2024a, 2024b). **We propose to adapt the discussion about future works to make that more clear in the paper.**
>
> We hope that these discussions will have answered your questions. Please do not hesitate to ask us to develop further if something was not clear.

---

### Decision · Program_Chairs · 2025-05-01

**Decision:**

Accept (poster)

**Comment:**

This paper provides a theoretical analysis of asymmetric actor-critic in POMDPs, where the critic, Q-function, uses the state information during learning. The key result is that, based on the analysis in [1], this asymmetric critic can remove one "aliasing" error term due to the use of agent state in the finite-time error bound, compared to the symmetric case. The problem being considered is important and of practical relevance, the insight regarding this aliasing term is new, and the manuscript is generally well written. However, there were some concerns regarding the assumptions being made: e.g., the use of i.i.d. sampling oracles, and boundedness of the concentrability coefficient, the proper use of the terminology "aliasing", etc.

During the discussion phase, some concern has come to my attention (based on the authors' responses), regarding whether $\epsilon_{grad}^{asym}$ would dominate $\epsilon_{grad}^{sym}$, especially in the tabular case, where the compatible error for the symmetric case could be made zero under tabular softmax parameterization, while the compatible error for the asymmetric one could not. This weakens the claims of "provable benefits" of asymmetric critic learning. One could argue, though, that when (linear) function approximation is used, these two terms might not be directly comparable. But again, this may weaken the overall claim of the provable benefit of asymmetric learning, by only looking at the removal of the “aliasing terms” (i.e., other error terms, since they are different, may matter as well). I suggest that the authors add these important remarks in the camera-ready version.

Also, during the discussion phase, one important related work [2] was brought to our attention, which was published in NeurIPS 2024 and appeared on OpenReview in Sept., 2024 (more than 4-months before ICML submission deadline), and also studied the provable benefit of asymmetric actor-critic with privileged state information, but was not mentioned in the submission. [2] addressed the more challenging setting with online exploration (and thus without the assumption on the boundedness of the concentrability coefficient, nor i.i.d. sampling), with finite-sample and computational complexity results, with benefits compared to the setting without privileged information. Some claims in the submission thus may need to be properly changed, e.g., those regarding the "lack of theoretical justification" on the "potential benefit of asymmetric learning".

That said, the focused settings and the types of results in these two works are different -- the submission focused on the in-expectation, finite-time-error bound, with linear function approximation, as in [1], and thus the analysis techniques are fundamentally different from [2] as well.

Overall, the submission still has some merit as a contribution to ICML, with new perspectives that may further our understanding of asymmetric actor-critic, especially through the explicit comparison with the symmetric counterpart. I suggest the authors incorporate the feedback from the reviews and discussions in preparing their camera-ready version.


[1] Cayci, S., He, N., and Srikant, R. Finite-Time Analysis of Natural Actor-Critic for POMDPs. SIAM Journal on Mathematics of Data Science, 2024.

[2] Cai, Y., Liu, X., Oikonomou, A., and Zhang, K. Provable partially observable reinforcement learning with privileged information. NeurIPS, 2024.